# In-situ and wavelength-dependent photocatalytic strain evolution of a single Au nanoparticle on a TiO$_2$ film

Sung Hyun Park [1], Sukyoung Kim [2,3], Jae Whan Park [4], Seunghee Kim [2,3], Wonsuk Cha [5] & Joonseok Lee [2,3,6] ✉

Photocatalysis is a promising technique due to its capacity to efficiently harvest solar energy and its potential to address the global energy crisis. However, the structure–activity relationships of photocatalyst during wavelength-dependent photocatalytic reactions remains largely unexplored because it is difficult to measure under operating conditions. Here we show the photocatalytic strain evolution of a single Au nanoparticle (AuNP) supported on a TiO$_2$ film by combining three-dimensional (3D) Bragg coherent X-ray diffraction imaging with an external light source. The wavelength-dependent generation of reactive oxygen species (ROS) has significant effects on the structural deformation of the AuNP, leading to its strain evolution. Density functional theory (DFT) calculations are employed to rationalize the induced strain caused by the adsorption of ROS on the AuNP surface. These observations provide insights of how the photocatalytic activity impacts on the structural deformation of AuNP, contributing to the general understanding of the atomic-level catalytic adsorption process.

Photocatalysis is a promising approach for converting solar energy into chemical energy and treating wastewater without relying on fossil fuels[1–4]. Among the various semiconductor photocatalysts, TiO$_2$ is the most promising candidate owing to its affordability, widespread commercial availability on a large scale, and improved catalytic stability[5–7]. However, the catalytic efficiency of TiO$_2$ is limited due to the high recombination rate of the photogenerated charge carriers and its limited light absorption ability owing to its large bandgap ($\approx 3.2$ eV)[8–10]. Therefore, plasmonic noble metal/TiO$_2$ catalysts, such as Au nanoparticle (AuNP)/TiO$_2$, have been extensively studied to enhance the photocatalytic efficiency of TiO$_2$, considering the tunability of AuNP absorption in the visible to near-infrared (NIR) range[11–14].

In particular, AuNPs in Au/TiO$_2$ heterostructures exhibit a distinctive ROS generation pathway depending on the excitation wavelength[15,16]. The specific excitation wavelengths can modulate the excited state of the active sites and influence the interaction between the adsorbates (i.e., ROS) and catalytic Au surface[17,18]. In this respect, understanding the catalytic reactions occurring at a single catalytic center is critical for the rational design of more efficient catalysts with improved selectivity, activity, and lifetime[19,20]. However, the structure–activity relationships of photocatalysts at the atomic level during photocatalytic reactions remain largely unexplored owing to difficulties in conducting measurements under the operating conditions[21]. To better comprehend the structure–activity relationships of catalytic materials, various conventional techniques have been

¹Department of HY-KIST Bio-Convergence, Hanyang University, Seoul 04763, Republic of Korea. ²Department of Chemistry, Hanyang University, Seoul 04763, Republic of Korea. ³Research Institute for Natural Sciences, Hanyang University, Seoul 04763, Republic of Korea. ⁴Center for Artificial Low Dimensional Electronic Systems, Institute for Basic Science, Pohang 37673, Republic of Korea. ⁵X-ray Science Division, Advanced Photon Source, Argonne National Laboratory, Argonne, IL 60439, USA. ⁶Research Institute for Convergence of Basic Sciences, Hanyang University, Seoul 04763, Republic of Korea. ✉e-mail: joonseoklee@hanyang.ac.kr

employed to study photocatalytic reactions. Microscopic techniques such as scanning tunneling microscopy have been used to identify low-coordinated atoms that act as active sites[22]. Infrared spectroscopy and X-ray absorption spectroscopy have been used to resolve the interactions between the catalyst and reactants[23,24]. Furthermore, in-situ high-resolution transmission electron microscopy (TEM) has been used to observe changes in the shape or size of individual photocatalysts during the catalytic process[25]. Nevertheless, a comprehensive understanding of the structural deformation and the catalytic activity of individual nanoparticles is not clearly understood.

Bragg coherent X-ray diffraction imaging (BCDI) has proven to be an effective technique for investigating the structural deformation of individual nanocrystals during catalytic reactions[26,27]. In this technique, coherent X-ray diffraction patterns of a single nanocrystal are obtained during a catalytic reaction[28]. The acquired structural data, represented by complex values, are reconstructed using phase-retrieval algorithms[26]. This approach enables the determination of the nanocrystal morphology with high spatial resolution and the characterization of the internal strain field with picometer sensitivity[26,29–31]. In recent years, BCDI has been successfully used to reveal the local strain field of catalysts based on noble metals in various in-situ environments, such as the methane oxidation-induced strain of Pt nanoparticles and the electrochemically induced strain of Pt-Ni alloy[32,33].

However, the influence of external light on the evolution of structural deformations of catalysts during photocatalytic reactions in aqueous environments has not yet been studied.

In this study, we investigated the strain evolution of a single AuNP on $TiO_2$ film using in-situ photocatalytic BCDI during photocatalytic reactions. We controlled the ROS generation activity of the Au/$TiO_2$ heterostructure by applying three different wavelengths (green, UV, and green/UV light). This observation presents wavelength-dependent strain evolutions of a single photocatalyst during photocatalytic reactions. Our results reveal that in-situ photocatalytic BCDI provides a detailed understanding of wavelength-dependent structure deformation (i.e., the intensity of Bragg peak, strain evolution, or d-band theory of catalysis) during the photocatalytic reaction and also gives insight into the rational design of photocatalysts.

## Results

### Preparation and characterization of Au/$TiO_2$ heterostructures

To investigate the strain evolution of a single AuNP during the photocatalytic reaction, we supported single AuNPs on a $TiO_2$ film (see Fig. 1). For this, a $TiO_2$ precursor solution was spin coated onto a Si substrate and subsequently annealed at 500 °C for 6 h to obtain a 100 nm thick $TiO_2$ film[34]. Then, AuNPs synthesized using the sol-gel method were drop-cast on the $TiO_2$ film[35]. The morphology of the

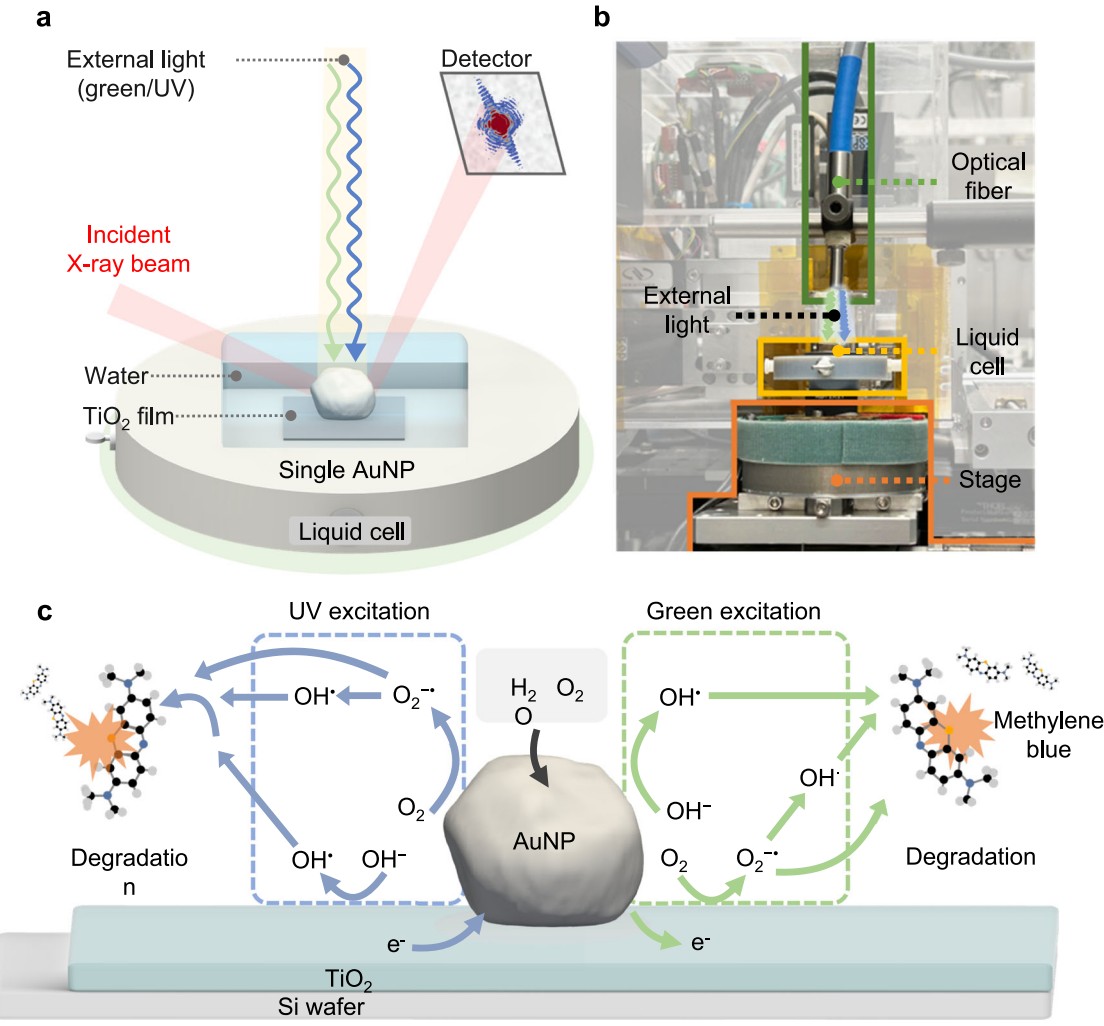

**Fig. 1 | Schematic illustration of the in-situ photocatalytic BCDI experiment.**
**a** Au/$TiO_2$ heterostructure was placed in the in-situ BCDI liquid cell and the excitation wavelength was controlled using a xenon lamp. This setup allowed us to systematically irradiate the green (532 nm), UV (365 nm), and green/UV (532 and 365 nm) to the Au/$TiO_2$ heterostructure during BCDI measurements. The incident-focused X-ray beam interacts with the single AuNP inside the reaction liquid cell. Diffraction patterns from the single AuNP were collected at the off-specular (111) Bragg angles. **b** Photograph of the in-situ photocatalytic BCDI setup at the 34-ID-C at APS. **c** Schematic illustration of the photocatalytic degradation mechanism of MB by Au/$TiO_2$ heterostructure under green and UV light irradiation.

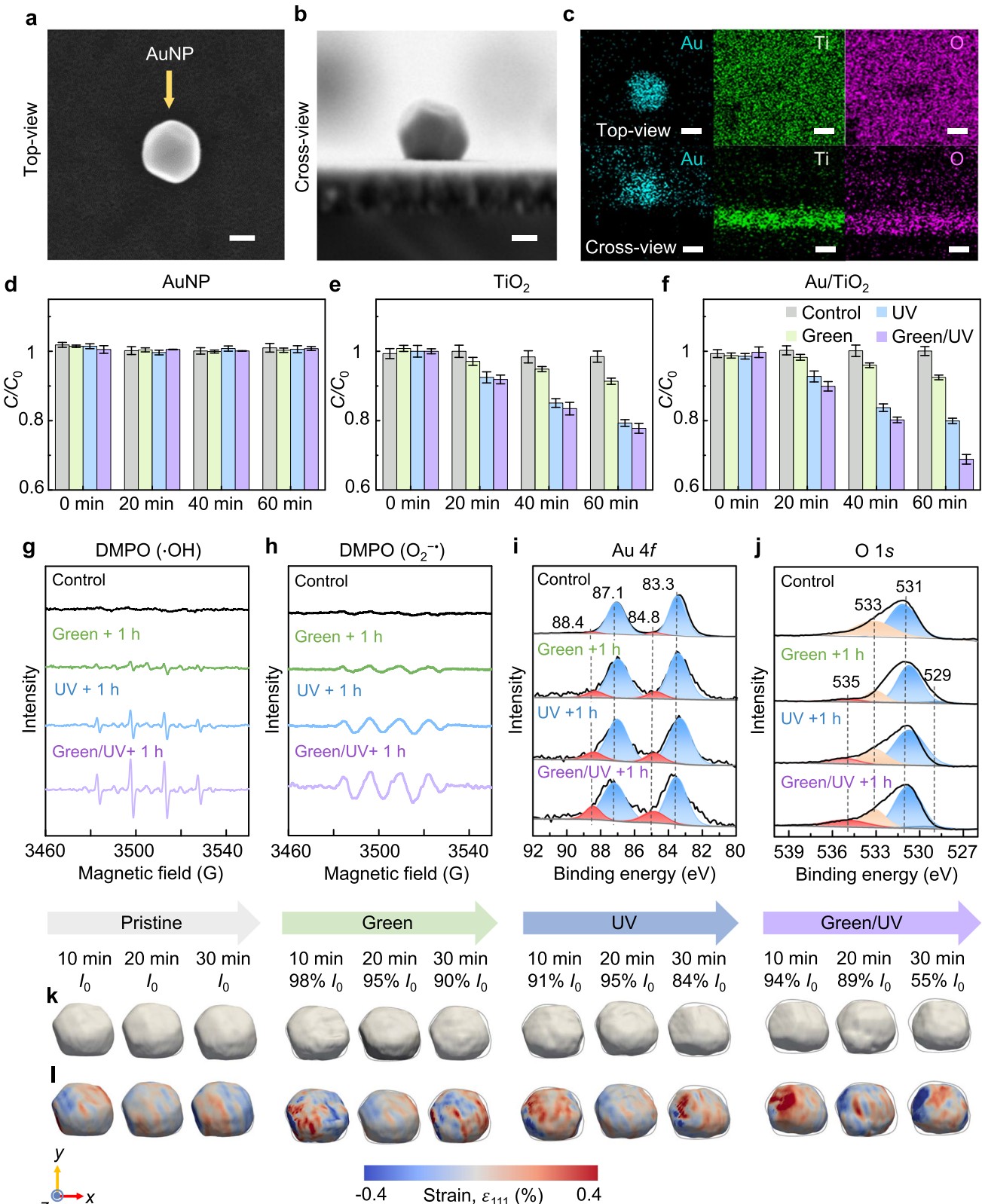

**Fig. 2 | Characterization of a single AuNP deposited on TiO₂ film.** SEM images of **a** top-view and **b** cross-view of the Au/TiO₂ heterostructure. **c** EDX mappings of Au, Ti, and O. Scale bar: 50 nm. Photocatalytic degradation of MB on **d** AuNP, **e** TiO₂, and **f** Au/TiO₂ heterostructure under green, UV, and green/UV irradiation for 60 min. Data are expressed as mean ± SD ($n$ = 3 independent experiments). ESR analysis of **g** •OH and **h** O₂⁻˙ generation under green, UV, and green/UV irradiation after 1 h. DMPO was used as the spin-trapping agent. High-resolution XPS spectra of **i** Au 4$f$ and **j** O 1$s$. **k** 3D reconstructed images of AuNP at a 30% amplitude threshold. The gray boundary illustrates the shape of the AuNP in the pristine condition at 10 min. **l** 3D strain images of AuNP associated with compressive (blue) and tensile (red) strains. The lattice strain along the [111] direction was projected onto the isosurfaces. Colors in gray, green, blue, and purple represent pristine, green, UV, and green/UV irradiation, respectively. Source data are provided as a Source Data file.

resulting Au/TiO$_2$ heterostructure was confirmed using scanning electron microscopy (SEM). Each AuNP was well separated from its neighbors on the TiO$_2$ film (Supplementary Fig. 1). This arrangement enabled selective irradiation of individual nanoparticles with an incident X-ray beam of ≈500 nm along both horizontal and vertical directions. As shown in Fig. 2a, b, we selected specific isolated AuNP with a size of ≈150 nm for BCDI. The presence of Au, Ti, and O elements in the Au/TiO$_2$ heterostructure was confirmed by energy-dispersive X-ray (EDX) mapping (Fig. 2c). The X-ray diffraction (XRD) pattern of the Au/TiO$_2$ heterostructure revealed that it consisted of pure face-centered cubic AuNPs and anatase TiO$_2$ (Supplementary Fig. 2). The anatase structure of TiO$_2$ has a large optical bandgap (3.2 eV) that allows the effective separation of electrons and holes under UV irradiation. The UV/visible spectra of the AuNP suspension exhibited a distinct absorption band at 532 nm due to the plasmonic resonance effect of the AuNP (Supplementary Fig. 3)[36].

To understand the wavelength-dependent photocatalytic activity, we conducted a methylene blue (MB) degradation test under green, UV, and green/UV irradiation of AuNP solution, TiO$_2$ film, and the Au/TiO$_2$ heterostructure[37]. MB was selected as the indicator to assess the catalytic activities of the different materials, because it is readily degraded by ROS through demethylation processes[38,39]. The absorbance of MB at 664 nm was measured to calculate its concentration (Supplementary Figs. 4–6). In the AuNP solution, MB degradation was not observed under the different irradiation conditions because of the high charge recombination rate (Fig. 2d)[40]. The TiO$_2$ film without AuNPs showed comparable photocatalytic degradation of MB under UV (16.3%) and green/UV (17.8%) irradiation for 60 min, but no photocatalytic reaction occurred under green irradiation because of its wide bandgap of 3.2 eV (Fig. 2e)[41,42]. On the other hand, the Au/TiO$_2$ heterostructure exhibited the highest degradation efficiency under green/UV (31.2%) irradiation, surpassing the efficiencies under UV (19.8%) and green (7.6%) irradiation conditions (Fig. 2f). The enhanced MB degradation under green/UV irradiation is attributed to the broad absorption of the Au/TiO$_2$ heterostructure and efficient charge separation process[11,41,42]. Thus, the Au/TiO$_2$ heterostructure exhibits distinct MB degradation characteristics under green, UV, and green/UV irradiation compared with those observed using AuNPs and TiO$_2$. These results suggest that the photocatalytic activity of the Au/TiO$_2$ heterostructure can be effectively controlled by varying the excitation wavelength.

We further investigated the ROS generation activity (ROS = •OH and O$_2^{-\bullet}$) of the Au/TiO$_2$ heterostructure to study the impact of the ROS on the AuNP structure[16,43]. The generation activities of •OH and O$_2^{-\bullet}$ were quantified by measuring the emission intensities of 7-OH-coumarin (455 nm) and dihydroethidium (DHE) (420 nm), respectively (Supplementary Figs. 7, 8)[32,44]. As shown in Supplementary Fig. 9a, the •OH generation activity under green and UV light irradiation was 2- and 4-fold higher compared to dark conditions. The Au/TiO$_2$ heterostructure exhibited the highest •OH generation activity with a 5.5-fold increase under green/UV irradiation. Similarly, the O$_2^{-\bullet}$ generation activity reached a maximum under green/UV irradiation, showing a 3.1-fold increase relative to dark conditions. This exceeded the increases observed under UV (1.9-fold) or green (1.3-fold) irradiation (Supplementary Fig. 9b). To verify the type and content of ROS generated, electron spin resonance (ESR) analysis was performed using DMPO as a spin trap agent for •OH and O$_2^{-\bullet}$ (Fig. 2g, h). After 1 h of reaction, the Au/TiO$_2$ heterostructure exhibited typical ESR signals of DMPO (•OH) and DMPO (O$_2^{-\bullet}$) under all illumination conditions. The intensities of both ESR signals progressively increased in the order of green, UV, and green/UV irradiation, reflecting the same increasing trend for the excitation wavelength. These findings align with those from the ROS generation activity experiments using coumarin and DHE in Supplementary Figs. 7, 8. The consistent rise in both the O$_2^{-\bullet}$ and •OH generation activities across different wavelengths indicates that the

excitation wavelength crucially influences photocatalytic reactions through an electron-transfer mechanism[45].

Supplementary Fig. 10 schematically illustrates the electron-transfer mechanism of the Au/TiO$_2$ heterostructure under various irradiation conditions. Under green-light irradiation, photo-induced electron-hole pairs are generated through the plasmonic resonance effect of the AuNP[11–13]. Because photoexcited electrons have a very short lifetime of the order of picoseconds, the electron-hole pairs on the AuNP cannot participate in the photocatalytic reaction[46]. However, as AuNPs are attached to the TiO$_2$ surface, the photoexcited electrons from the AuNPs can effectively migrate to the conduction band (CB) of TiO$_2$, inhibiting the relaxation and recombination of electrons in the Au/TiO$_2$ heterostructure[40,47,48]. This electron-transfer process facilitates the accumulation of electrons on the TiO$_2$ surface and promotes the reduction of O$_2$ to O$_2^{-\bullet}$ [10,48]. Furthermore, the remaining holes in the AuNPs can react with OH$^-$ to produce •OH[39,48,49]. However, under UV irradiation, electrons in the valence band (VB) of TiO$_2$ can be excited to the CB, resulting in holes in the VB[33,45]. Because the Fermi level of the AuNP is lower than that of the CB of TiO$_2$, the photoexcited electrons in the CB can migrate to the AuNP. This electron transfer generates O$_2^{-\bullet}$ on the Au surface through O$_2$ reduction, whereas the holes in TiO$_2$ react with OH$^-$ to generate •OH[33,45]. Under combined green/UV irradiation, these mechanisms operate simultaneously, leading to efficient charge transfer and resulting in the highest ROS generation activities. As shown in Supplementary Fig. 11, the plasmonic resonance effect induced by green irradiation generates electrons in AuNPs, which are then transferred to the CB of TiO$_2$. At the same time, UV irradiation generates electron-hole pairs in TiO$_2$. The excited electrons of TiO$_2$ accumulated in the CB of TiO$_2$ with the electrons transferred from AuNP, which could promote the generation of O$_2^{-\bullet}$ in the CB of TiO$_2$. Additionally, the holes in the AuNP and TiO$_2$ could promote the generation of •OH. This charge-transfer mechanism demonstrates that the Au/TiO$_2$ heterostructure is capable of optimal ROS generation through its efficient wavelength absorption ability under green/UV irradiation.

To confirm the adsorption of ROS on the Au surface, we conducted spectroscopic experiments including XPS, FT-IR, and Raman spectroscopy during the photocatalytic reaction. High-resolution XPS was performed on the Au samples before and after different wavelength irradiation. As shown in Fig. 2i, two peaks centered at 83.3 and 87.1 eV of XPS spectra in all samples represent the $4f_{7/2}$ and $4f_{5/2}$ signals of metallic Au (Au$^0$), respectively[50]. Before light irradiation, the relatively narrow and symmetrical Au $4f$ peaks indicate that it mainly exists in the state of Au$^0$. However, after different light irradiation, two shoulder peaks at 84.8 eV and 88.4 eV are observed, which are attributed to the oxide state of Au (Au$^+$)[50]. Furthermore, the O $1s$ XPS spectra show two additional shoulder peaks at 529.0 and 535.0 eV, corresponding to chemisorbed oxygen species from ROS generated during photocatalytic reactions (Fig. 2j)[51]. As shown in the Raman spectra in Supplementary Fig. 12a, a new peak at 480 cm$^{-1}$, which is attributed to the stretching vibrations of Au−O species, appears after different light irradiation[52]. The FT-IR spectra show bands at 3740 and 1250 cm$^{-1}$, which can be regarded as the generation of new Au−O species on the Au surface (Supplementary Fig. 12b, c)[53,54]. The intensity of each peak tends to increase in the order of green, UV, and green/UV irradiation. This means that the amount of ROS generated is proportional to the amount of adsorption on the Au surface.

## Photocatalytic strain in AuNP bulk and surface

To investigate the structural deformation of the AuNP in the Au/TiO$_2$ heterostructure, we performed in-situ photocatalytic BCDI experiments on a single AuNP under green, UV, and green/UV irradiation. The setup for the in-situ photocatalytic BCDI experiments is illustrated in Fig. 1b and Supplementary Fig. 13, and detailed information is provided in the Methods section. In this setup, a xenon lamp that

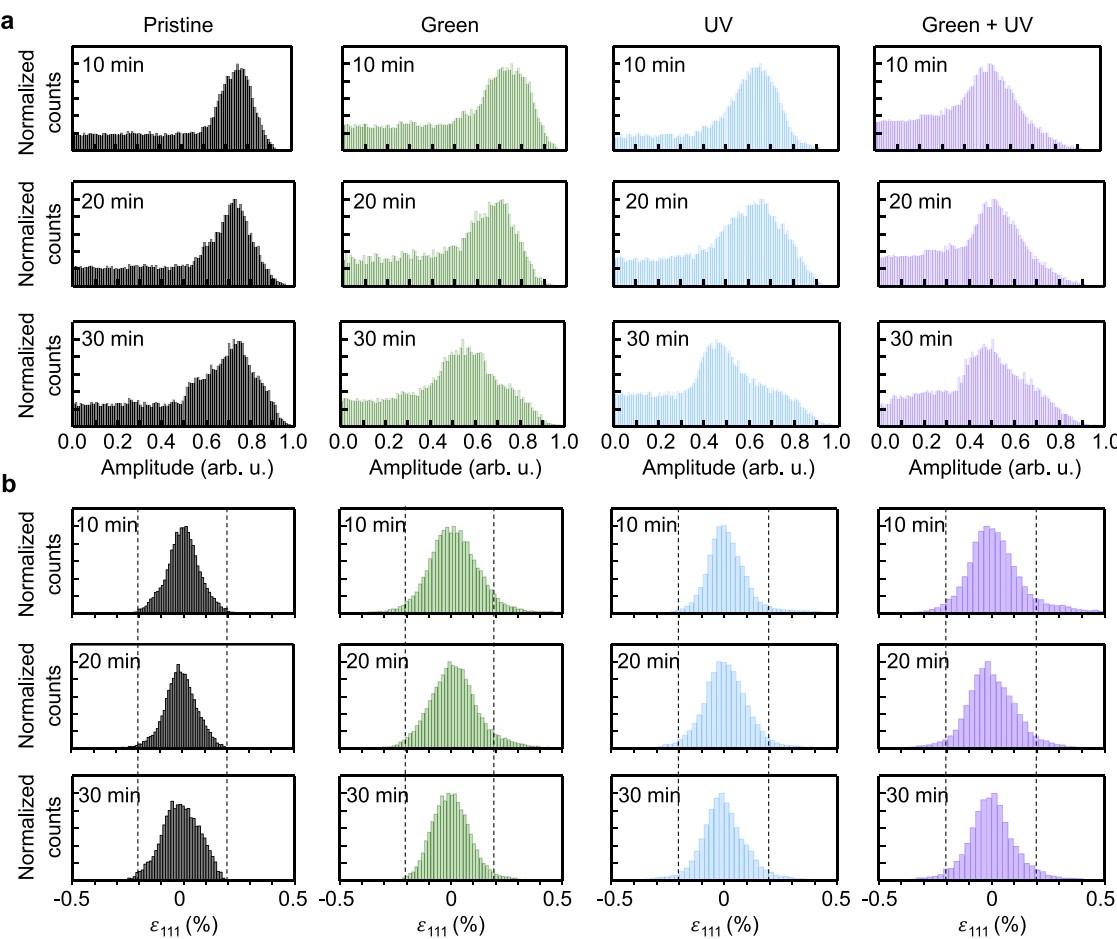

**Fig. 3 | Statistical distribution of the amplitude and strain of a single AuNP on TiO₂ film from Fig. 2.** The amplitude distributions between 0.0 and 1.0 are plotted in **a** with corresponding **b** bulk strain distributions. The lines on the strain plots indicate the threshold values (−0.2% and 0.2%). Colors in gray, green, blue, and purple represent pristine, green, UV, and green/UV irradiation, respectively. Source data are provided as a Source Data file.

emitted in the UV to NIR range served as the primary excitation source. The xenon lamp was equipped with optical filters to selectively filter specific wavelengths of light, enabling the irradiation of the sample with green (532 nm), UV (365 nm), or green/UV (532 and 365 nm) light. The excitation light was transmitted through an optical fiber and directly projected onto a liquid cell at a vertical angle of incidence. Because the Au/TiO₂ heterostructure was located in the water-based liquid cell, the photocatalytic reaction could be systematically controlled during the BCDI measurements. Coherent X-ray diffraction patterns were obtained around the Au (111) Bragg peak. Each 3D diffraction pattern was obtained by rotating the sample with respect to the (111) Bragg peak by 0.8° in steps of 0.02° (Supplementary Fig. 14). A phase-retrieval algorithm was employed to determine the 3D electron density and local strain field of a single AuNP (Supplementary Fig. 15). A measurement in air and water without irradiation was also performed to determine the beam stability and water radiolysis effect on AuNP (see Supplementary Note 1 and Supplementary Figs. 16–17)[55,56]. As shown in Fig. 2k, the AuNP structure under the four reaction conditions was reconstructed from the reconstructed electron density of the single AuNP. The reconstructed pristine AuNP (in water and non-irradiated) was ≈55 nm high, 95 nm wide, and 150 nm long, consistent with its morphology observed in the SEM images in Fig. 2a, b. Additionally, the pristine AuNP exhibited residual distortion of its surface due to the inherent strain signature of its morphology, and these distortions varied with the applied wavelength (Fig. 2l)[57].

To confirm the lattice distortion of the single AuNP, the integrated intensity of the Bragg peak (*I*) was calculated from the 3D diffraction pattern obtained under each irradiation condition (Fig. 2l). The initial integrated intensity (*I*₀) corresponds to the pristine AuNP. In the pristine condition, the integrated intensity of the AuNP remained constant over time. However, under green, UV, or green/UV irradiation, the Bragg peak intensity of the single AuNP decreased to 90%, 84%, and 55% of the *I*₀, respectively. Furthermore, the AuNP exhibited morphological changes such as volume reduction and surface roughening due to the photocatalytic reaction. As shown in Supplementary Fig. 18, under pristine conditions, the AuNPs did not exhibit significant volume changes over time. However, when the light source was irradiated for 30 min, the volume of AuNPs gradually decreased. The volume of the AuNPs decreased significantly in sequence with green (95%), UV (93%), and green/UV (90%) irradiation, indicating that lattice distortion in Au is proportional to the quantity of Au−O bonds generated. The adsorption of ROS on the surface of AuNPs forms Au−O bonds and causes lattice distortions, including partial phase transformations. In regions where these distortions are significant, the AuNPs may lose crystallinity or exhibit a crystallographic orientation that differs from the original state of the nanoparticle. These regions fail to contribute to the coherent X-ray diffraction pattern and could potentially result in large deformations, i.e., voids, in the shape of the reconstructed nanoparticles. Note that ROS exhibit high reactivity and possess a short lifetime, typically on the order of microseconds to nanoseconds[58]. Terminating light irradiation ceases the generation of

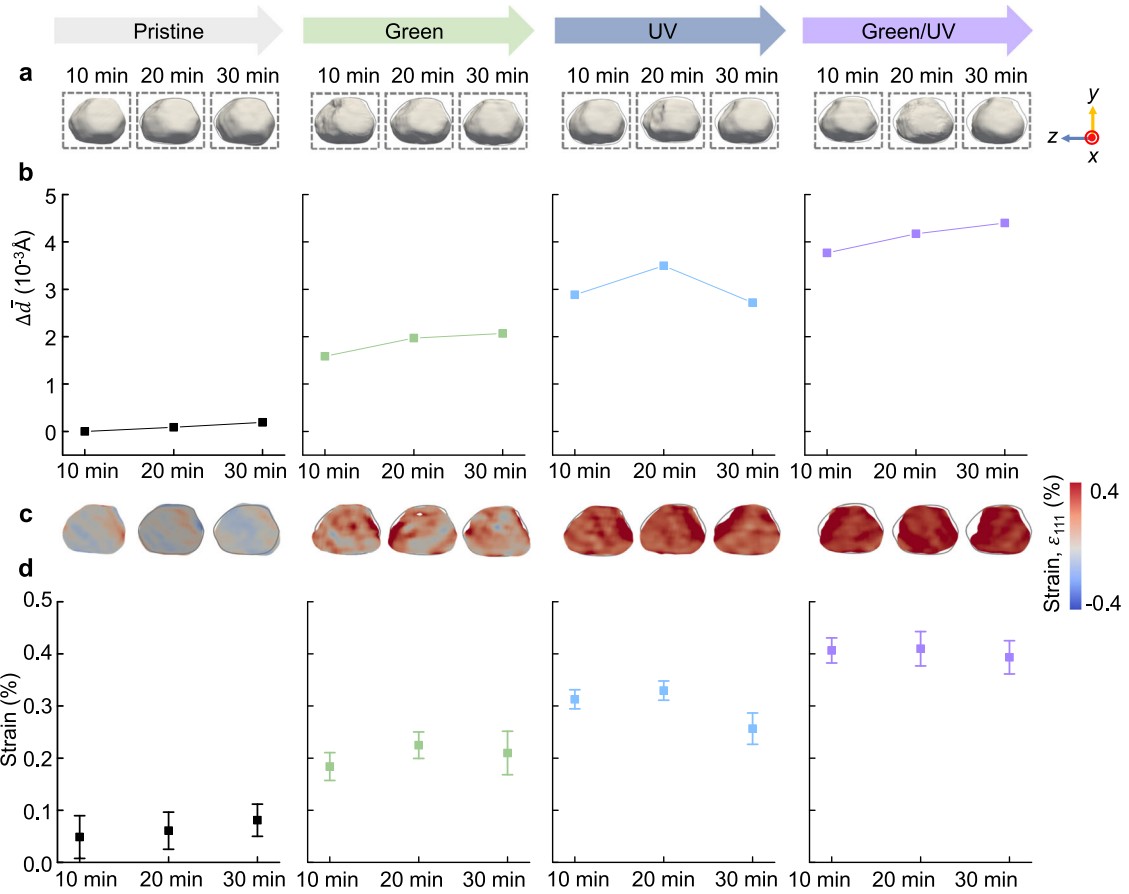

**Fig. 4 | Average strain field of a single AuNP. a** 3D reconstructed images at a 30% amplitude threshold with lattice displacement along the [111] direction projected on the isosurfaces. The gray boundary depicts the pristine cross-section in Fig. 4a. **b** Variation in the average lattice spacing ($\Delta \bar{d}$) under green, UV, and green/UV irradiation for 30 min. The change in $\Delta \bar{d}$ is calculated using the formula $\Delta \bar{d} = \bar{d}_{111} - \bar{d}^{*}_{111}$, where the $\bar{d}_{111}$ is the average lattice spacing measured across the AuNP and $\bar{d}^{*}_{111}$ is the average lattice spacing of the first measured pristine AuNP.

**c** Cross-sectional views of the internal strain field at the dashed line box in Fig. 4a. The gray boundary illustrates the shape of the AuNP in pristine condition at 10 min. **d** Average strain field for the internal plane in Fig. 4c under green, UV, and green/UV irradiation for 30 min. Error bars in **a** represent standard deviations ($n = 3$). Colors in gray, green, blue, and purple represent pristine, green, UV, and green/UV irradiation, respectively. Source data are provided as a Source Data file.

ROS and reduces adsorption to the AuNP surface, which reduces lattice distortions. This could potentially reduce voids, allowing the AuNPs to approach a state similar to their initial condition[59,60]. However, the fact that they do not fully revert to the original state implies irreversible structural changes (Supplementary Fig. 19 and Supplementary Table 1). Accumulative changes during repeated and long-term photocatalytic reactions could lead to significant irreversible structural changes. The single AuNP did not undergo significant changes in the air environment, in contrast to the water environment, irrespective of the wavelength of light (Supplementary Fig. 20). These results suggest that the structural deformation of the AuNP is due to the adsorption of the ROS, the products of photocatalytic reactions in water, onto the Au surface[46].

The photo-induced structural changes in the single AuNP were further investigated using the amplitude and strain histograms of the single AuNP. In the pristine condition, the amplitude histograms of the AuNP showed a Gaussian-like distribution with the peak value remaining constant for 30 min (Fig. 3a). However, under irradiation conditions, the amplitude histogram shifted to the left, and the counts of the high-amplitude region between 0.7 and 1.0 decreased as the reaction progressed. In particular, green/UV irradiation resulted in significantly lower counts in the high-amplitude region compared with those under the other irradiation conditions. This result implies that photo-induced ROS causes lattice distortions and reduces the

crystallinity of the AuNP. Based on the 3D strain values for AuNP, the quantitative strain distributions in the surface and bulk states are summarized in the histograms (Fig. 3b and Supplementary Fig. 22). The surface strain was calculated for all surface voxels, whereas the bulk strain was calculated for all voxels that were entirely surrounded by other voxels. As shown in Fig. 3b, the bulk strain distribution remained below the predefined threshold ($-0.2\%$ and $+0.2\%$) value under pristine conditions. In contrast, under all irradiation conditions, there was a significant increase in the high-strain region. Furthermore, this increasing trend was also confirmed in the surface strain distribution, indicating that the strain initiated at the surface propagates to the bulk of the AuNP (Supplementary Figs. 21, 22). Supplementary Table 2 presents the full width at half maximum (FWHM) values of surface versus bulk strain distributions as a function of the irradiated wavelength. Under pristine conditions, the FWHM values of surface and bulk strain distributions show similar values, with an average of 0.018% and 0.017%, respectively. However, during the photocatalytic reactions, the average FWHM value of the surface strain distribution showed an increase in the sequence of green (0.022%), UV (0.023%), and green/UV (0.026%) light irradiation. This trend indicates that the surface strain intensified in proportion to the generated ROS. In addition, the average FWHM value of the bulk strain distribution under photoreaction conditions also increased compared to the value of pristine conditions but showed almost the same value (0.021%)

regardless of the wavelength. This suggests that the strain originates at the AuNP surface and is accommodated across the entire bulk region of the AuNP.

### In-situ wavelength-dependent strain evolution of a AuNP

To investigate the propagation of surface changes inward to the bulk region of the AuNP, we calculated the absolute variation in average lattice spacing ($\Delta \bar{d} = \bar{d}_{111} - \bar{d}^{*}_{111}$), where $\bar{d}^{*}_{111}$ represents to the initially measured pristine AuNP (Fig. 4a, b)[30,61]. Under the pristine condition, no significant changes in $\Delta \bar{d}$ were observed during the BCDI measurement. However, subsequent to various light irradiations, $\Delta \bar{d}$ sequentially increased in the order of green (0.0017 to 0.0019 Å), UV (0.0027 to 0.0032 Å), and green/UV (0.0038 to 0.0043 Å). This increase in lattice spacing $\Delta \bar{d}$ correlates with the generation of ROS, which is attributed to ROS adsorption on the Au surface. Figure 4c shows the strain field of a single AuNP under different irradiation conditions. In contrast with the behavior in the pristine conditions, the strain distribution changed continuously during the reaction in the irradiated state. As shown in Fig. 4d, in the pristine conditions, slightly positive strain values were observed in water, ranging from 0.04% to 0.08%, suggesting that there was no significant water-induced effect on the isolated AuNP. In contrast, when irradiated with green light, the inner strain increased slightly to 0.18%–0.22%. Under UV irradiation, a significant expansion of the AuNP lattice was observed (0.28%–0.32%). The most significant expansion (0.39%–0.41%) occurred during green/UV irradiation, coinciding with the highest rate of ROS generation activity (see Fig. 2f). To increase the reliability of the results through BCDI measurement, we also analyzed a single ellipsoidal AuNP, which differs in shape from the AuNP shown in Fig. 2. As shown in Supplementary Figs. 23–25, we observed consistent trends in structural evolution across the AuNPs of different types. These results indicate that the structural deformation of the AuNP varies according to the extent of ROS generation.

DFT calculations were performed to predict which molecules preferentially adsorb to the Au surface and to understand the origin of the strain dynamics. We considered OH, $O_2$, and $H_2O$ as potential adsorbates on the Au (111) surface[62,63]. The DFT calculation with OH predicted that the bridge site is the most stable position, with an adsorption energy of −2.512 eV, compared to the top (−2.296 eV) and hollow (−2.398 eV) sites. Upon OH adsorption at the bridge site, the surface Au atoms shifted away from the adsorbate to stabilize the overall structure, resulting in surface expansion by 4.98 pm (Supplementary Fig. 26a, d). The $O_2$ molecule also exhibited stability at the Au bridge site, with an adsorption energy of −0.653 eV, causing the Au atoms to expand by 2.33 pm from their original positions (Supplementary Fig. 26b, e). In contrast, $H_2O$ adsorbed nearly parallel to the top site of the Au (111) surface via oxygen atoms with an adsorption energy of −0.112 eV, exhibiting the lowest adsorption energy among the considered molecules (Supplementary Fig. 26c). The surface expansion of the $H_2O$ adsorbed Au surface was only 0.170 pm (Supplementary Fig. 26f), which is 29.4 and 13.7 times smaller than those observed for OH and $O_2$ adsorption, respectively. These results suggest that the interactions of OH and $O_2$ with the Au surface can induce strain on the Au (111) surface, whereas the strain due to $H_2O$ adsorption is negligible.

### Discussion

In this study, in-situ photocatalytic BCDI was employed to investigate the wavelength-dependent strain evolution of a single AuNP supported on a semiconducting $TiO_2$ film during photocatalytic reactions (Fig. 1). The strain evolution of the AuNP varied according to the extent of ROS generation and was determined by the wavelength-dependent charge-transfer mechanism. Under green-light irradiation, plasmon-mediated electrons are transferred to the conduction band of $TiO_2$, resulting in holes on the Au surface[64,65]. These holes accumulate near the Au surface, serving as additional active sites for the generation of •OH. The resultant ROS induces ≈0.2% tensile strain in the Au lattice[66,67]. However, under UV irradiation, the AuNP acts as an electron sink, and these accumulated electrons on the surface trigger the generation of $O_2^{-}$ and •OH species[33,43,45]. During the photocatalytic reaction, the interaction between the Au surface and $O_2^{-}$ and •OH species leads to the formation of Au−O species on the Au surface. Considering that UV irradiation produces more ROS than green irradiation (Fig. 2f), it results in a larger tensile strain (≈0.3%) in the AuNP. Moreover, under green/UV irradiation, the Au/$TiO_2$ heterostructure efficiently absorbs both green and UV wavelengths, maximizing ROS generation through enhanced charge transfer, which is reflected in the increased formation of Au−O on the Au surface[45,68]. Consequently, the increased availability of electrons and holes under combined irradiation conditions provides more active sites for the generation of ROS compared to those available under either green or UV irradiation alone, leading to a significant dilation (≈0.4%) of the lattice due to the increased production of ROS. Even after the photocatalytic reaction, the reconstructed AuNPs did not completely revert to their initial state. These findings suggest that irreversible structural deformations at the single-particle level can contribute to macroscopic irreversibility including reductions in photocatalytic performance and catalyst stability, particularly during repeated and long-term reactions at a bulk scale. Based on the above, our results provide fundamental insights into the structure–activity relationship of single AuNPs, including the D-band theory of catalysis and strain evolution under different irradiation conditions, thereby facilitating strain and defect engineering of photocatalysts.

## Methods

### Materials

Titanium(IV) isopropoxide (97%), hydrochloric acid (HCl, 37%), isopropanol (99.5%), gold(III) chloride ($HAuCl_4$, 99.90%), cetyltrimethylammonium chloride solution (CTAC, 25 wt.% in water), sodium borohydride ($NaBH_4$, >98.0%), L-ascorbic acid (99%), L-glutathione (>98%), cetyltrimethylammonium bromide (CTAB, >98%), methylene blue (MB, >82%), coumarin (>99%), dihydroethidium (DHE, >95%), and 5,5-dimethyl-1-pyrroline N-oxide (DMPO, >98.0%) were purchased from Sigma-Aldrich.

### Sample preparation

The $TiO_2$ precursor solution for a thin film of $TiO_2$ was prepared by mixing 370 μL titanium (IV) isopropoxide, 35 μL of 2 M HCl, and 5 mL of isopropanol at room temperature for 10 min[34]. After depositing 20 μL of the solution onto a square silicon substrate with an area of 15 × 15 mm², spin coating was done at a speed of 4000 rpm for 10 s. Afterward, all the films were annealed at 500 °C in argon for 4 h.

For the synthesis of AuNP, spherical seeds with an approximate size of 2.5 nm were prepared by combining 250 μL of $HAuCl_4$ (10 mM) with 10 mL of CTAC (100 mM). Subsequently, 250 μL of a freshly prepared ice-cold $NaBH_4$ (10 mM) aqueous solution was added to the mixture solution. The solution was then incubated at 30 °C for 2 h. The growth solution for producing octahedral seeds consisted of a mixture containing 250 μL of $HAuCl_4$ (10 mM), 9.5 mL of CTAC (100 mM), and 220 μL of L-ascorbic acid (40 mM). To remove impurities, the solution underwent two rounds of centrifugation at 8600 × g for 150 s, followed by redispersion in CTAB solution (1 mM). The growth solution for AuNPs was prepared by combining 4.0 mL of water, 800 μL of CTAB (100 mM), 100 μL of $HAuCl_4$ (10 mM), 475 μL of L-ascorbic acid (100 mM), and 5 μL of L-glutathione (5 mM). Subsequently, the mixture was incubated at 30 °C for 2 h after spraying 50 μL of diluted seed solution. The resultant solution was centrifuged twice at 8600 × g and redispersed in CTAB (1 mM)[35]. The preparation of Au/$TiO_2$ heterostructure was conducted following the procedure. A droplet of the synthesized AuNP was dropped on a $TiO_2$ thin film. Then, it was placed under 3.8% $H_2$ gas condition at 400 °C for 1 h.

## Characterization of Au/TiO$_2$ heterostructures

The morphology, size, and elemental composition of the Au/TiO$_2$ heterostructures were characterized using extreme high-resolution scanning electron microscope (FE-SEM, Gemini 3, Zeiss) at the Center for Polymers and Composite Materials, Hanyang University, Korea. The crystal characteristics of the Au/TiO$_2$ heterostructures were characterized by a Bruker D8 ADVANCE diffractometer. The measurements utilized Cu K$\alpha$ radiation with a wavelength of 0.15405 nm and were performed at the Hanyang LINC 3.0 Center for Research Facilities located in Seoul. The samples were analyzed using FT-IR (Nicolet iS10, Thermo Fisher Scientific Co., USA), XPS (Theta, Thermo Fisher Scientific Co., Waltham, MA, USA), and Raman spectroscopy (DXR3xi, Thermo Fisher Scientific Co., Waltham, MA, USA).

## Photocatalytic degradation of MB

We monitored the bleaching of MB using a UV–vis spectrophotometer (Agilent Technologies, Cary 5000). MB aqueous solution (1 mM, pH 6.4, 25 °C) was freshly prepared using deionized water without O$_2$ bubbling prior to the photocatalytic reaction. The 1 cm × 1 cm film of Au/TiO$_2$ heterostructures was placed in a 24-well plate at a concentration and irradiated with green light (30 mW cm$^{-2}$ at $\lambda$ = 532 nm), UV light (30 mW cm$^{-2}$ at $\lambda$ = 365 nm), and green/UV light (30 mW cm$^{-2}$) for 1 h each. A xenon lamp (Thorlabs, SLS200) was used for green irradiation, coupled with 532 nm (FLH532-10, Thorlabs, 532 nm CWL, 25 mm diameter) bandpass filter. A UV lamp (Asone, LUV-4) was used for UV irradiation. For green/UV irradiation, both the xenon and UV lamps were simultaneously used to irradiate the Au/TiO$_2$ heterostructure. The change in absorption at 664 nm was recorded using the UV–vis spectrophotometer.

## Detection of ROS

To determine both the type and quantity of ROS generated by the photocatalytic reaction: DHE were employed as detection probes for hydroxyl radicals (•OH) and superoxide radicals (O$_2^-$), respectively. These probes were diluted to a concentration of 10 $\mu$M in the deionized water without O$_2$ bubbling and subsequently introduced to a 24-well plate with a 1 cm × 1 cm film of Au/TiO$_2$ heterostructure. Furthermore, for ESR analysis, DMPO at a concentration of 50 mM was introduced into the deionized water and a methanol solution, respectively, without O$_2$ bubbling.

The Au/TiO$_2$ heterostructures were exposed to green light (30 mW cm$^{-2}$ at $\lambda$ = 532 nm), UV light (30 mW cm$^{-2}$ at $\lambda$ = 365 nm), and green/UV light (30 mW cm$^{-2}$) for 1 h each. The same UV and green lamp for MB degradation test were used for ROS detection. A microplate reader (Thermo 520040) was used to measure fluorescence signals immediately before and after illumination. When coumarin was used, fluorescence emission at 455 nm was recorded upon excitation at 332 nm. When DHE was employed, fluorescence emission at 420 nm was recorded upon excitation at 580 nm.

## BCDI experimental setup

In-situ photocatalytic BCDI setup was installed in the 34-ID-C beamline at the Advanced Photon Source, Argonne National Laboratory, USA. The DI water (250 $\mu$L) without O$_2$ bubbling was dropped onto the sample mounted in a liquid cell for BCDI measurements. Additionally, the sample was measured in air to test the stability. An external xenon light source (Thorlabs, SLS200) was coupled with 532 nm (FLH532-10, Thorlabs, 532 nm CWL, 25 mm diameter), 365 nm (FBH365-10, Thorlabs, 365 nm CWL, 25 mm diameter) bandpass filters, and 405 nm (NF405-13, Thorlabs, 405 nm CWL, 25 mm diameter) notch filters for green, UV, and green/UV irradiation, respectively. To sequentially irradiate the excitation light during the BCDI measurements, the xenon lamp was connected to an optical fiber (Thorlabs, LG5-4T) placed vertically 5 cm above the liquid cell (see Supplementary Fig. 13). The

sample was sequentially exposed to green light (20 mW cm$^{-2}$ at $\lambda$ = 532 nm), UV light (40 mW cm$^{-2}$ at $\lambda$ = 365 nm), and green/UV light (70 mW cm$^{-2}$) for 30 min each, while BCDI was measured, with a 10-min rest period after each measurement.

## BCDI experiment

BCDI experiments were performed at 34-ID-C beamline at the APS in ANL, USA. A coherent X-ray beam, with an energy of 10 keV and a flux of $5 \times 10^9$ photons s$^{-1}$, was precisely focused to dimensions of 450 nm × 520 nm (horizontally × vertically) using Kirkpatrick–Baez mirrors. Coherent X-ray diffraction patterns, specifically under Au (111) Bragg conditions, were captured using a Timepix detector from Amsterdam Scientific Instruments. This detector had a pixel size of 55 $\mu$m and a resolution of 512 × 512 pixels, positioned 1 m away from the sample. The operando liquid cell was horizontally shifted until the detector captured a Bragg peak from an individual AuNP. The 3D coherent X-ray diffraction pattern near the Au (111) Bragg peak was conducted through a rocking curve scan within the ± 0.4° range, employing a step size of 0.02°, and allowing 5 to 10 s of exposure time for each step. We utilized a beamline-mounted optical microscope to localize the nanoparticles analyzed in the BCDI measurements (Supplementary Fig. 27). Subsequently, SEM analysis, guided by the optical images, was performed to precisely target the identical location. This approach enabled the measurement of the same nanoparticles using both BCDI and SEM techniques.

## BCDI analysis

The reconstruction process utilized error reduction (ER) and hybrid input-output (HIO) algorithms over a total of 620 iterations[69,70]. The phase-retrieval algorithm initiated with 20 ER iterations, followed by 180 iterations of the HIO algorithm with guided analysis. A Gaussian shrink-wrap function was employed as the support constraint. The resultant reconstruction data was visualized as isosurface contour maps representing image amplitude in both 2D and 3D cross-sections using Paraview. The reconstructed voxel size for the AuNP was determined to be 4.40 × 4.40 × 4.40 nm$^3$ or 5.18 × 5.18 × 5.18 nm$^3$ for each scan. This measurement was derived from the experimental setup (detector pixel size and distance) and the step size of the rocking scan. The spatial resolution was determined by differentiating the electron density amplitude across the object-substrate interface for the $x$, $y$, and $z$ directions and fitting a Gaussian to each of the profiles. The resolution was ≈ 8.0 nm to 19.9 nm, depending on the measurement conditions (Supplementary Table 3). Surface voxels are defined as those not completely surrounded by other voxels, while bulk voxels are those entirely enclosed by other voxels. Due to the voxel size, surface voxels hold average information not only from within the outermost atomic layer but also approximately from the 20 outermost atomic layers. The retrieved images at an isosurface of 30% showed the electron density distribution and the phase corresponding to displacement projected along $\mathbf{q}_{111}$ vector. The averaged lattice spacing ($\bar{d}_{111}$) was obtained through the length of the reciprocal lattice vector q$_{111}$ from the center of intensity mass of the 3D data. In the phase-retrieval algorithm, the obtained phase $\varphi(r)$ provides the lattice displacement projected along scattering vector $q$ throughout the entire crystal volume, by the following equation $\varphi(r) = q \cdot u(r)$, where u(r) represents the lattice displacement field. We calculated the corresponding strain field $\varepsilon_{111}$ using the definition $\partial u_{111}/\partial r_{111}$, where r$_{111}$ represents the spatial coordinate in the direction of $\mathbf{q}_{111}$[69,70].

## DFT calculation

We performed DFT calculations using the Vienna ab initio simulation package within the Perdew-Burke-Ernzerhof functional and the projector augmented wave method[71–73]. Our modeling of the Au(111) surface involved a periodic slab consisting of six Au layers, with a vacuum

spacing of more than 21 Å. Brillouin-zone integrations for the Au(111)-$(3 \times 3)$ unit cell were carried out with a $4 \times 4 \times 1$ $k$-point mesh. A plane-wave basis with a cutoff of 400 eV was used. Atom relaxation was conducted until residual force components were within 0.01 eV. The adsorption energy is defined as $E_{ad} = E_{OH/Au(111)} - E_{OH} - E_{Au(111)}$.

## Data availability

The data that support the findings of this study are available from the corresponding author upon request. The raw data generated at the Advanced Photon Source (34-ID-C beamline) are available from Zenodo[74]. Source data are provided with this paper.

## Code availability

All analysis codes for BCDI are available from Zenodo[74] and from the corresponding author.

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

## Acknowledgements

This research was supported by the National Research Foundation of Korea (NRF) grant funded by the Korea government (MSIT) (NRF-2023R1A2C2003128, NRF-2022R1A4A1030421). This research used resources of the Advanced Photon Source, a U.S. Department of Energy (DOE) Office of Science user facility operated for the DOE Office of Science by Argonne National Laboratory under Contract No. DE-AC02-06CH11357.

## Author contributions

S.H.P. performed the synthesis of Au/TiO$_2$, the overall experiments, and the data analysis. S.H.P., Sukyoung K., and W.C. carried out the BCDI imaging experiments at 34-ID-C beamline. S.H.P., W.C., and Sukyoung K. performed the BCDI data analysis. Seunghee K. performed the ROS generation experiments. J.W.P. carried out DFT simulations and analysis. S.H.P. wrote the original manuscript. J.L. was responsible for supervision of project administration and writing—review & editing.

## Competing interests

The authors declare no competing interests.
