## [Peer Review File · Nature Communications]

In-situ and wavelength-dependent photocatalytic strain evolution of a single Au nanoparticle on a TiO₂ filmREVIEWER COMMENTS

Reviewer #1 (Remarks to the Author):

In this work, by using a three-dimensional Bragg coherent X-ray diffraction methods, the authors revealed the relationship between the microstructure (stress changes) and activity of the photocatalyst at the atomic level, while Au/TiO₂ was chosen as an example photocatalyst to degrade methyl blue for assessing photocatalytic activity. The authors claimed that the stress changes of Au nanoparticles in Au/TiO₂ are caused by active oxygen species and are determined by wavelength-dependent charge transfer mechanisms. Furthermore, the study has established a correlation between strain and activity, with large stresses enhancing the adsorption of reactants, resulting in a significant increase in the number of reactive oxygen species (ROS) adsorbed onto the surface. The results are interesting. However, before publication, there are some crucial issues that need to be addressed:

- (1) The results obtained in this work are appealing, but their generality and applicability to other systems need to be further examined. For instance, it would be interesting to investigate whether the observed effects persist when Au nanoparticles are replaced with other noble (such as Pt or Rh) or non-noble (Ni) metal nanoparticles. Similarly, exploring the use of alternative substrates, such as strontium titanate or tantalum nitride, in place of the titanium dioxide substrate could provide valuable insights into the universality of the reported findings. Therefore, the authors are encouraged to extending their study to these and other relevant systems to enhance the breadth and impact of their research.
- (2) What is the guidance of this work for the design of photocatalysts?
- (3) The authors attributed the change of lattice distortion to "the adsorption of the generated ROS on the Au surface". This is an important assumption, while evidence from characterization like spectroscopy should be given out to confirm these adsorptions.
- (4) The types and content of ROS play a crucial role in determining the photocatalytic activity and stress changes. Therefore, it is essential to employ more advanced techniques, such as electron spin resonance (ESR) spectroscopy, to accurately determine the types and content of ROS.
- (5) In supplementary Figure 5, it is observed that, under ultraviolet light, only hydroxyl radicals participate in the degradation of reactive oxygen species. However, under green light, both hydroxyl and superoxide radicals are involved in the degradation process. How can these differences be distinguished? Direct evidence should be provided.
- (6) According to the text, ROS can affect the strain of Au nanoparticles, and high strain can lead to strong adsorption of active molecules. Is this, logically, a self-enhancing mechanism?
- (7) When both UV and green light are simultaneously irradiated, the manuscript should offer further explanation regarding the charge transfer pathway, the generation location of ROS, and the reasons for the increase in ROS quantity.
- (8) In the BCDI experiment, green light (20 mW/cm², $\lambda = 532$ nm), UV light (40 mW/cm², $\lambda = 368$ -365 nm), and green/UV light (70 mW/cm²) were used as light sources to examine stress variations with different wavelengths of light. In the photocatalytic degradation experiment, all three light sources had a power density of 30 mW/cm². Since the article is exploring the correlation between photocatalytic activity and stress alterations, why was the input light source density inconsistent?
- (9) For the phenomenon described in Figure 3a, the Gaussian distribution of Au nanoparticle amplitude shifted to the left after illumination, which means that ROS causes lattice distortion and reduces the crystallinity of Au. However, the derivation process and basis for this important conclusion are not clearly explained, making it difficult for readers to understand and accept this conclusion.

Reviewer #2 (Remarks to the Author):

The paper addresses the in situ imaging of strain evolution in Au nanoparticles supported on a TiO₂ film during photocatalysis. This marks the first instance of measuring the photocatalytic strain evolution of individual particles. I recommend publication after revision. However, I am not fully

convinced of the quality of the reconstructions, as the shape and size of the reconstructed particles exhibit significant variation. Evaluation of the reconstruction quality can be performed through the amplitude distributions presented in Fig. 3. A more peaked amplitude histogram indicates a better reconstruction, and the varying histograms suggest differing qualities of reconstructions. Given the diverse qualities of the reconstructions, this variability may introduce artifacts in the surface strain presented in the publication. The authors use the same amplitude threshold (30%) to illustrate the particle's surface strain, despite dissimilar amplitude histograms.

1. It would be interesting to include information about particle stability during Bragg coherent diffraction imaging measurements.
2. Please provide a scale bar in Fig. 2c.
3. Explain the observed differences in surface strain in Fig. 2h in the pristine state. The surface strain differs for various measurement durations (10 min, 20 min, and 30 min). Explain the large variations in the amplitude histogram in the pristine state (see Fig. 3a).
4. In Figs. 2g-h, the particle's shape and size differ. Add the contour of the initial pristine particle to guide the eye in Figs. 2 and 4, as well as in the Supplementary Figures. Are these differences attributed to poor reconstruction quality? The variability in the reconstructed particle's shape is puzzling.
5. The authors attribute irreversible changes to dissolution and defect dynamics but do not show defect formation during the photocatalytic reaction. Supplementary Figure 8 reveals that green excitation induces voids in the Au particle. Explain how such large voids can be elucidated and why, after turning off the light, the particle no longer exhibits voids (recovery from voids).
6. Provide information on how generated reactive oxygen species/radicals are evacuated between one excitation and another (UV/green).
7. Does the X-ray beam generate extra radicals or oxygen species?
8. Explain how you calculate compressive and strain areas in Supplementary Table 1. Provide the unit. If it is an area, it should be in square meters.
9. It would be interesting to obtain the full width at half maximum of the bulk and surface strain histograms (Fig. 3b and Supplementary Fig. 11) for comparison.
10. In BCDI, heterogeneous strain is observed. Why is the internal strain field not centered at zero under UV and green/UV illuminations? Was the diffraction pattern well-centered before phase retrieval?

There are a few misspellings. In the Methods section (Sample preparation), 'heterostructure' is not correctly written, and in the Methods section (Photocatalytic degradation of MB), replace 'A UV lamp were used' with 'A UV lamp was used.'

Reviewer #3 (Remarks to the Author):

see attached PDF

Review for NCOMMS-23-57475

Relevance: Relevant for wider audiences, due to the application of Bragg coherent X-ray diffraction imaging for in situ lattice displacement of nanostructures as a direct result of metal/adsorbate interactions. An insight into these displacements is important for understanding the behavior of nanostructures under reaction conditions and could also find applications in the fields of electrocatalysis, energy storage and nano fabrication.

Cited references: Diverse demographic background of the cited literature. All citations accessible to the broader public. Authors tend to cite review articles instead of original publications, especially when considering the used diffraction technique and the mechanism of hot electron injection from excited Au NPs into the TiO₂ film. In several sentences the cited literature does not fit the general message of the sentence.

Recommendation: Accept with major revisions

The manuscript "In situ photocatalytic strain evolution of a single Au nanoparticle in Au/TiO₂ heterostructures" investigates the deformation of an individual gold nanoparticle supported on TiO₂ under the influence of UV and green irradiation. The authors show that the photocatalytic generation of reactive oxygen species, triggered by irradiation causes a lattice displacement in the investigated gold nanoparticle. Initially the photocatalytic activity of the TiO₂ supported Au Nps under green, UV and a mix of green and UV light is compared with an Au Np, suspension and bare TiO₂. The results of both photocatalytic degradation of methylene blue and fluorescence of 7-OH-coumarin indicate an increased production of reactive oxygen species (ROS) on the heterosystem, with the highest activity under UV/green irradiation. The authors suggest an electron transfer from TiO₂ into the Au NP in the UV case and a hot electron injection from Au into the TiO₂ under green light. Therefore, OH radicals are generated on the electron depleted surface of the TiO₂ film while O₂⁻ radicals are generated at the negatively charged Au surface in the first case and vice versa in the latter. A similar behavior in OH and O₂⁻ radical production for both green and UV irradiation is suggested to verify the electron transfer between the Au and TiO₂ in both cases. In the second part the influence of this ROS generation on the structure of the gold nanoparticles is studied by in situ Bragg coherent X-ray diffraction imaging (BCDI). The authors show that the lattice strain of both the surface and the bulk of the Au nanoparticles increases with increased activity for ROS generation. In accordance with the decreasing Bragg diffraction maxima under different irradiation times, this indicates an adsorbent dependent restructuring of the Au NPs surface. The authors also show that the surface strain leads to a lattice displacement throughout the bulk of the nanoparticle. Lastly, density functional theorem was used to verify that the adsorption of OH could cause the restructuring of an Au surface.

The authors concluded that the high reactivity during UV/green irradiation is caused by an increased absorption efficiency of the Au/TiO₂ heterostructure, leading to higher negative surface charge density at the Au NPs, generating more active sites for ROS generation. This leads to an increased strain on the Au surface, causing further lattice displacement, which affects the Au adsorption energies, further altering catalytic activity. It is suggested that this indicates the Au NP as main source of active sites for the ROS generation. Lastly the irreversibility of the structural change under photocatalytic conditions is discussed and how it could hint on catalyst deactivations.

Major revisions:

The proposed mechanism for UV/green mixed light is difficult to understand, as the proposed mechanism for green light includes migration of hot electrons to the TiO₂ and the UV mechanism proposes photoelectrons migrating from TiO₂ to Gold. Following that logic, a mixed light should not increase efficiency as the two mechanisms are competing. Furthermore, the cited literature does only consider the visible spectrum [56] or uses a broadband spectrum that also exhibits a larger range in the UV part (280-400 vs. 200 to 1500 nm) which makes a direct comparison difficult [46]. This part needs further clarification. Additional DFT studies that investigate the implication of O₂- radical generation on lattice strain of Au nanoparticles need to be done to clarify if the Au NPs function as hole or electron carrier during UV/green irradiation.

Risk of overinterpretation due to the focus on one single Au nanoparticle. While the capabilities to understand the structural evolution of a single particle is indeed impressive, the proposed evolution should be demonstrated on at least three different particles to verify that the observed behavior is not an outlier.

Lack of investigation of surface roughness changes and particle radius and volume changes, under different irradiation conditions. A depiction of size evaluation is typically standard procedure for BDCI investigations.

The irreversibility in structural change during the photocatalytic process is not verified by a secondary method such as SEM. Post experimental SEM would also help to verify if the in situ BDCI experiments follow the same trends in regards of deformation as a blank experiment.

Availability of RAW data and used methods: The authors claim on line 416 that all relevant supporting data are available within main text or supplementary information. But neither manuscript contains raw data from the degradation experiments nor depicts the meta data for the used phase retrieval algorithm. This makes the evaluation of data reconstruction validity difficult as the errors are unknown.

Citation related revision:

On line 57 and 58 the cited literature does not discuss ROS generation in Au/TiO₂ heterostructures

On line 134 the reference 43 does not contain any of the cited information and discusses biological implications of coumarin on biological systems, no instances of gold or ROS appear in the article, also reference 44 does not discuss strain effects

On line 152, which is crucial for the explanation of the mechanism, the same two references 43 and 44 are cited, which still do not discuss relaxation and recombination effects.

Line 154 the reference 10 does only loosely refer to the mechanisms described in the sentence

Also 154 the cited article [11] is a review article that states in one single sentence that energetic holes retained on small Au nanoparticles have sufficient energy to drive the oxygen-evolution-half reaction, citing two other publications.

Minor revisions:

In Figures 3g, 4a and Supplementary figures 8 and 9 the orientation of the reconstructed iso surface is sometimes changed between the different conditions or irradiation times. This makes a comparison

difficult and the Figures confusing. To circumvent this an indication of the particle orientation would be helpful.

The difference in depicted strain structure between Figure 2 and Supplementary Fig.10 is not clear.

For Figure 3 binning is not kept constant over the different irradiation times, making a comparison difficult.

The data in Figure 3 a indicates a change in crystallinity of the pristine Au Np in water without irradiation over 30 minutes. This change is not discussed and might hint towards beam damage during X-ray experiments, especially as radiolysis is known to generate ROS in water

[<https://doi.org/10.3390/w3010235>]. Additionally, photo ionization of the Au Nps or the TiO₂ film, could alter green and UV ionization rates. From the given data synergistic interactions between X-ray irradiation and the observed photocatalytic activity cannot be Furthermore, no information of the photon flux is given, making the possibility of beam damage difficult to assess.

In the introduction on line 73 there is a sudden jump in reasoning from active sites to strain studies, without a further discussion of who these two parameters relate to each other.

Reviewer #4 (Remarks to the Author):

Reviewer #5 (Remarks to the Author):

The manuscript by Park et al. reports on the strain evolution of a TiO₂-supported Au nanoparticle under green, UV, and a combination of green and UV light irradiation conditions, using Bragg Coherent Diffraction X-ray Imaging. While I believe the results provided here are valuable and novel, significant revisions are required. The authors claim to explore structure-activity relationships and provide an understanding of wavelength-dependent structure strain, but the efforts in this regard could be significantly improved. A more extensive analysis, particularly focusing on surface strain in highly strained sites or regions, supported by DFT results, could be employed to better emphasise and spotlight the findings. I recommend resubmitting the paper after addressing these major revisions.

1. I have a few general comments on the figures. I recommend adding tripod axes to figures involving the reconstruction of the NP, whether in 3D views or cross-sections, as not the same views are consistently depicted across the figures. Additionally, I suggest the authors verify the strain magnitudes indicated in the figures and text. Strain values range from -0.04% to 0.04%, but 0.40% strain is mentioned in the text. The "%" symbol is omitted in some figures, which could lead to misinterpretation, suggesting 4% strain. Finally, the authors should provide both the voxel size and the spatial resolution achieved in this study.

2. As a non-expert of the photocatalysis field, I find the measurements of photocatalytic efficiency described, nicely illustrating the synergy gained from the Au/TiO₂ heterostructure. However, the details about the methylene blue (MB) solution are not clear. Specifically, the involvement of O₂ and OH⁻ in the process is mentioned, but details such as pH (and consequently OH⁻ concentration) or whether O₂ was bubbled in the solution are not addressed and could be indicated in the Methods section.

3. I acknowledge the achievement of measuring nanoparticles within a liquid under in situ conditions, considering the relatively small size of the nanoparticles, which makes it challenging for them to remain stable on the support. The authors were able to measure the same nanoparticle using SEM and BCDI. A description of their methodology to achieve this would be appreciated in the Methods section.

4. In the Results section, the mechanism describing the processes involved in the operation mode of the heterostructure is presented, but the reader is encouraged to refer to Supplementary Figure 5. Figure 1c provides more illustrative support for this mechanism.

5. Only the results obtained along the Au (111) direction are presented, not because it is sensitive to lattice distortions in Au (100) and Au (110), as stated in the text. I recommend removing the sentence that suggests lattice distortion along [100] and [110] directions is accessible through the 111 peak, as this might be confusing. Note that Au(hkl) is not a direction, but [hkl] is.

6. I highly appreciate the efforts invested in plotting the diffraction patterns which are often omitted from BCDI papers. I recommend not saturating the colours in the diffraction pattern representation. This might mask changes in the pattern features. As Phase Retrieval is a challenging task and, in some cases, its reliability can be limited, confidence in the reconstructed strain fluctuations can be achieved by examining variations in the diffraction pattern features. When looking closely at the diffraction patterns, they seem to have rotated slightly between the different conditions. Is there any explanation for this?

7. A reference on the phasing algorithm employed would be appreciated.

8. It is mentioned that the pristine AuNP displayed residual surface distortion due to non-equilibrium growth conditions. While equilibrium growth conditions might have led to a more relaxed nanoparticle, the morphology inherently imposes a strain signature, even in the absence of non-equilibrium growth conditions.

9. I suppose the pristine AuNP corresponds to the nanoparticle immersed in the MB solution and not in

air. But I did not clearly find this information.

10. The general correlation between the decrease in integrated intensity (I) and the distortion at the reconstruction surface is hazardous. Changes in integrated intensity can originate from various factors (beam, optics...) and may be difficult to interpret. They can indeed result from a reduction in crystallinity. Importantly, as I decreases, the reconstruction quality also decreases, directly correlating with the left shift of the peak of the reconstructed amplitude histogram, as illustrated in Fig. 3a. Plus, as the reconstruction quality decreases, the strain values are more likely to be large (positive or negative). In addition, given that the reconstruction is consistently observed at an isosurface of 30%, it is likely that equivalent surfaces are not always considered between reconstructions, making comparisons of surface strains challenging.

11. I am sceptical about drawing conclusions from the histogram of the normalised reconstructed amplitudes. Left shifts in the histogram and changes in histogram shape indicate a reduction in the quality of reconstruction, which are likely due to the decrease in integrated intensity. Moreover, it is usual to display the entire range [0; 1] of normalised amplitudes. These histograms should ideally serve to determine a relevant isosurface for each reconstruction.

12. I would encourage the authors to provide a methodology on the determination of the bulk and surface voxels.

13. Regardless of the strain region (bulk or surface), Supplementary Table 1 indicates that integrated strain over all tensile regions consistently exceeds integrated strain over all compressed regions. This appears contradictory, as the strain discussed in this paper is more precisely the heterogeneous strain – corresponding to lattice fluctuations around the average d-spacing of the scanned NP under a given condition. Consequently, this strain is always zero when averaged over all voxels. Thus, tensile strain cannot surpass compressive strain for both bulk and surface regions. The possibility of more tensile strain indicating an unremoved (positive) phase ramp in post-processing after phasing is acknowledged. In my view, the surface strain histogram asymmetry or full width at half maximum are more compelling, illustrating that specific surface sites are more strained than others. Additionally, the quality of reconstruction and (iso)surface determination strongly influences the histograms. Therefore, I recommend attaching more importance to strain distribution when reconstructions are considered sufficiently accurate.

14. Similar trends in surface vs bulk histograms are interpreted as indicative of a poor accommodation of the bulk region. I would rather propose the opposite perspective, but I might be overly concerned about this point, and respect the authors' interpretation. My intention is to ensure they accurately conveyed their intended meaning.

15. I discourage the use of displacement analysis in this context, unless specifically addressing defects, which is not the case here. In a BCDI experiment, displacement always involves an offset and cannot directly account for structural changes. Instead, focusing on how fast displacement changes across the NP (displacement variations) informs on structural changes, which aligns with strain and eliminates the offset issue. Additionally, the drawn cross sections in Fig. 4b is perpendicular to the [111] direction. This implies that the displacements shown correspond to atomic position variations out of the section plane, making interpretation challenging. Fig. 4c and 4d hold more meaningful information in this regard. I would also ask the authors to add the plane position since the dashed line box does not indicate where the section is taken.

16. I found Supplementary Figure 12c and d challenging to understand, particularly with the mention of "in-plane" distortion of the top slab layer while coloured arrows indicate the out-of-plane directions. Authors could clarify this. Employing DFT calculations is a highly valuable effort to rationalise strain observations under different reaction conditions. However, analysis could be enhanced by calculating the out-of-plane strain in the simulated slab to compare it with the strain captured experimentally on the top facet (also out-of-plane). Crystallographic axes in the figure would further improve clarity.

17. While the Results section clearly stipulates that O²⁻ and OH[°] should be generated at the AuNP surface under UV and green irradiation, respectively, the Discussion section introduces the idea of OH[°] formation at the AuNP surface under UV irradiation. However, I did not find any explanation for this point in the text. Given that OH[°] is said to cause a "higher tensile strain [on the NP] compared to that generated under green-light irradiation," I would appreciate further clarification on this part.

18. I am highly concerned about the interpretations presented in the Discussion section. The assertion

that the Au surface becomes negatively charged, particularly given that green-light irradiation generates a hole at the surface, is unclear to me. Assuming this is valid, the logical connection between the negative charge and the dilation of the Au lattice at the surface is not clearly established. Moreover, the subsequent interpretation appears to be reversed and biased. The authors suggest that the negatively charged surface dilates the Au surface lattice, shifting the d-band centre and subsequently increasing the adsorption of ROS species at the Au surface. Yet, no analytical evidence is provided to confirm this interpretation. The observed strain is more likely the result of adsorption phenomena, as demonstrated by numerous BCDI-catalysis papers, rather than from a negatively charged surface that would enhance ROS species adsorption. In my opinion, an in-depth analysis of the strain at the surface of the nanoparticle should be provided, linked explicitly to adsorption phenomena using DFT simulations as robust support. This would explicitly demonstrate that the TiO₂ supported NP is particularly active under UV/green-light irradiation.

REVIEWER COMMENTS

We thank the reviewers for their thoughtful review of the manuscript. We have carefully considered their comments when preparing our revision, which greatly improved the quality of the manuscript. The following text shows our responses to the reviewers' comments.

Reviewer #1 (Remarks to the Author):

In this work, by using a three-dimensional Bragg coherent X-ray diffraction methods, the authors revealed the relationship between the microstructure (stress changes) and activity of the photocatalyst at the atomic level, while Au/TiO₂ was chosen as an example photocatalyst to degrade methyl blue for assessing photocatalytic activity. The authors claimed that the stress changes of Au nanoparticles in Au/TiO₂ are caused by active oxygen species and are determined by wavelength-dependent charge transfer mechanisms. Furthermore, the study has established a correlation between strain and activity, with large stresses enhancing the adsorption of reactants, resulting in a significant increase in the number of reactive oxygen species (ROS) adsorbed onto the surface. The results are interesting. However, before publication, there are some crucial issues that need to be addressed:

1) The results obtained in this work are appealing, but their generality and applicability to other systems need to be further examined.

a. For instance, it would be interesting to investigate whether the observed effects persist when Au nanoparticles are replaced with other noble (such as Pt or Rh) or non-noble (Ni) metal nanoparticles. Similarly, exploring the use of alternative substrates, such as strontium titanate or tantalum nitride, in place of the titanium dioxide substrate could provide valuable insights into the universality of the reported findings. Therefore, the authors are encouraged to extending their study to these and other relevant systems to enhance the breadth and impact of their research.

Answer. We thank reviewer #1 and appreciate the positive feedback. We agree with your insight and have been tried to analyze a variety of heterogeneous catalysts consisting of various single nanoparticles and substrates using BCDI. For the experimental design of this manuscript, we conducted the feasibility of strain evolution of single AuNPs on amorphous, anatase, and rutile TiO₂ substrates (Additional Resource. 1). In this process, SEM images and XRD patterns were measured for all samples, and 3D reconstruction images were obtained through Bragg coherent patterns. In addition, we are also conducting observation of strain evolution of a single Ni nanocube on Si, TiO₂, and CeO₂ substrates during catalytic reactions such as CO₂ methanation or ammonia decomposition (Additional Resource. 2). Since the Advanced Photon Source (APS) is currently in the process of upgrading to a 4th generation high-energy light source, BCDI experiments via the 34-ID-C beamline are currently difficult. However, we believe

that after the upgrade, a beam with higher resolution will increase the potential for expansion of this research.

[Additional Resource. 1]

Additional Resource. 1. **a** SEM images of the Au/Amorphous TiO₂, Au/Anatase TiO₂, Au/Rutile TiO₂, respectively (Scale bar: 100 nm) and **b** their corresponding XRD patterns. **c** 3D reconstruction images and their 2D diffraction pattern at 111 Au Bragg peak.

[Additional Resource. 2]

Additional Resource. 2. **a** SEM images of the Ni/Si, Ni/TiO₂, Ni/CeO₂, respectively (Scale bar: 100 nm) and **(b)** their corresponding XRD patterns. **c** 3D reconstruction images and their 2D diffraction pattern at 111 Au Bragg peak.

2) What is the guidance of this work for the design of photocatalysts?

Answer. In this study, we demonstrated that the wavelength-dependent generation of ROS impacts the tensile strain of AuNPs. This strain influences the adsorption and desorption properties of the intermediate by upshifting the d-band center of the AuNPs. This shift plays a critical role in a rate-determining step, thereby affecting the photocatalytic activity. Therefore, investigating the correlation between strain evolution and photocatalytic performance of a single catalyst through the integration of 3D BCDI and an external illumination system could contribute to improving catalyst efficiency and stability in various chemical reactions.

3) The authors attributed the change of lattice distortion to "the adsorption of the generated ROS on the Au surface". This is an important assumption, while evidence from characterization like spectroscopy should be given out to confirm these adsorptions.

Answer. We thank the reviewer for the valuable comment. We have conducted new spectroscopic experiments including XPS, FT-IR, and Raman spectroscopy to confirm the adsorption of ROS on Au surface during the photocatalytic reaction. High-resolution XPS was performed on the Au samples before and after different wavelength irradiation. As shown in Fig. 2i, two peaks centered at 83.3 eV and 87.1 eV of XPS spectra in all samples represent the $4f_{7/2}$ and $4f_{5/2}$ signals of metallic Au (Au^0), respectively¹. Before light irradiation, the relatively narrow and symmetrical Au 4f peaks indicate that it mainly exists in the state of Au^0 . However, after different light irradiation, two shoulder peaks at 84.8 eV and 88.4 eV are observed, which are attributed to the oxide state of Au (Au^+)¹. Furthermore, the O 1s XPS spectra show two additional shoulder peaks at 529.0 eV and 535.0 eV, corresponding to chemisorbed oxygen species from ROS generated during photocatalytic reactions (Fig. 2j)². As shown in the Raman spectra in Supplementary Fig. 12a, a new peak at 480 cm^{-1} , which is attributed to the stretching vibrations of Au-O species, appears after different light irradiation³. The FT-IR spectra show bands at 3740 cm^{-1} and 1250 cm^{-1} , which can be regarded as the generation of new Au-O species on the Au surface (Supplementary Fig. 12b, c)^{4, 5}. The intensity of each peak tends to increase in the order of green, UV, and green/UV irradiation. This means that the amount of ROS generated is proportional to the amount of adsorption on the Au surface. We have updated the additional spectroscopic results and explanations about the adsorption of the generated ROS on the Au surface in the revised manuscript.

[Revised Supplementary Fig. 12]

Supplementary Fig. 12. a Raman spectra of Au-O for the Au/TiO₂ after green, UV, and green/UV irradiation, respectively. FT-IR spectra of b OH and c OOH bond for the Au/TiO₂ under green, UV, and green/UV irradiation, respectively.

[Revised Fig. 2]

Fig. 2. Characterization of a single AuNP deposited on TiO₂ film. SEM images of (a) top-view and (b) cross-view of the Au/TiO₂ heterostructure. c EDX mappings of Au, Ti, and O. Scale bar: 50 nm. Photocatalytic degradation of MB on (d) AuNP, (e) TiO₂, and (f) Au/TiO₂ heterostructure under green, UV, and green/UV irradiation for 60 min. ESR analysis of (g) •OH and (h) O₂^{•-} generation under green, UV, and green/UV irradiation after 1 h. DMPO was used as the spin-trapping agent. High-resolution XPS spectra of (i) Au 4f and (j) O 1s. k 3D reconstructed images of AuNP at a 30% amplitude threshold. The gray boundary illustrates the shape of the AuNP in the pristine condition at 10 min. l 3D strain images of AuNP associated with compressive (blue) and tensile (red) strains. The lattice strain along the [111] direction was projected onto the isosurfaces.

[Revised main manuscript, page 8]

“To confirm the adsorption of ROS on the Au surface, we conducted spectroscopic experiments including XPS, FT-IR, and Raman spectroscopy during the photocatalytic reaction. High-resolution XPS was performed on the Au samples before and after different wavelength irradiation. As shown in Fig. 2i, two peaks centered at 83.3 eV and 87.1 eV of XPS spectra in all samples represent the $4f_{7/2}$ and $4f_{5/2}$ signals of metallic Au (Au^0), respectively⁵⁰. Before light irradiation, the relatively narrow and symmetrical Au $4f$ peaks indicate that it mainly exists in the state of Au^0 . However, after different light irradiation, two shoulder peaks at 84.8 eV and 88.4 eV are observed, which are attributed to the oxide state of Au (Au^+)⁵⁰. Furthermore, the O $1s$ XPS spectra show two additional shoulder peaks at 529.0 eV and 535.0 eV, corresponding to chemisorbed oxygen species from ROS generated during photocatalytic reactions (Fig. 2j)⁵¹. As shown in the Raman spectra in Supplementary Fig. 12a, a new peak at 480 cm^{-1} , which is attributed to the stretching vibrations of Au-O species, appears after different light irradiation⁵². The FT-IR spectra show bands at 3740 cm^{-1} and 1250 cm^{-1} , which can be regarded as the generation of new Au-O species on the Au surface (Supplementary Fig. 12b, c)^{53, 54}. The intensity of each peak tends to increase in the order of green, UV, and green/UV irradiation. This means that the amount of ROS generated is proportional to the amount of adsorption on the Au surface.”

[Reference]

- [1] Tian B, Zhang J, Tong T, Chen F. Preparation of Au/TiO₂ catalysts from Au(I)–thiosulfate complex and study of their photocatalytic activity for the degradation of methyl orange. *Appl Catal B* **79**, 394-401 (2008).
- [2] Yeo BS, Klaus SL, Ross PN, Mathies RA, Bell AT. Identification of Hydroperoxy Species as Reaction Intermediates in the Electrochemical Evolution of Oxygen on Gold. *Chemphyschem* **11**, 1854-1857 (2010).
- [3] Pfisterer JHK, et al. Role of OH Intermediates during the Au Oxide Electro-Reduction at Low pH Elucidated by Electrochemical Surface-Enhanced Raman Spectroscopy and Implicit Solvent Density Functional Theory. *ACS Catal* **10**, 12716-12726 (2020).
- [4] Panda RN, Hsieh MF, Chung RJ, Chin TS. FTIR, XRD, SEM and solid state NMR investigations of carbonate-containing hydroxyapatite nano-particles synthesized by hydroxide-gel technique. *J Phys Chem Solids* **64**, 193-199 (2003).
- [5] Zhang R, et al. Insight into the Effective Aerobic Oxidative Cross-Esterification of Alcohols over Au/Porous Boron Nitride Catalyst. *ACS Appl Mater Interfaces* **11**, 46678-46687 (2019).

4) The types and content of ROS play a crucial role in determining the photocatalytic activity and stress changes. Therefore, it is essential to employ more advanced techniques, such as electron spin resonance (ESR) spectroscopy, to accurately determine the types and content of ROS.

Answer. We thank the reviewer for the comment on the type and content of ROS. We have conducted a new ESR analysis to accurately determine the types and content of ROS (Fig. 2g, h). These spectra show ESR signals of $\cdot\text{OH}$ and $\text{O}_2^{\cdot-}$ under all illumination conditions. Both radicals show that the ESR signal increases in the order of green, UV, and green/UV, which indicates an increase in the amount of ROS generated relative to the excitation wavelength. The result is consistent with the results of the ROS generation activity experiment using coumarin and DHE in Supplementary Figs. 7 and 8. We have updated the additional results and explanations about the type and content of ROS on the excitation wavelength in the revised manuscript.

[Revised Fig. 2]

Fig. 2. ESR analysis of **g** $\cdot\text{OH}$ and **h** $\text{O}_2^{\cdot-}$ generation under green, UV, and green/UV irradiation after 1 h. DMPO was used as the spin-trapping agent.

[Revised main manuscript, page 6]

“To verify the type and content of ROS generated, electron spin resonance (ESR) analysis was performed using DMPO as a spin trap agent for $\cdot\text{OH}$ and $\text{O}_2^{\cdot-}$ (Fig. 2g, h). After 1 h of reaction, the Au/TiO₂ heterostructure exhibited typical ESR signals of DMPO ($\cdot\text{OH}$) and DMPO ($\text{O}_2^{\cdot-}$) under all illumination conditions. The intensities of both ESR signals progressively increased in the order of green, UV, and green/UV irradiation, reflecting the same increasing trend for the excitation wavelength. These findings align with those from the ROS generation activity experiments using coumarin and DHE in Supplementary Figs. 7 and 8.”

5) In supplementary Figure 5, it is observed that, under ultraviolet light, only hydroxyl radicals participate in the degradation of reactive oxygen species. However, under green light, both hydroxyl and superoxide radicals are involved in the degradation process. How can these differences be distinguished? Direct evidence should be provided.

Answer. We thank the reviewer for the comment regarding the degradation process of MB. We apologize for any confusion about the ROS generation mechanism. The Au/TiO₂ heterostructure in this research varies in the location and type of ROS generation depending on UV and green light irradiation (Additional Resource. 3).

Under UV irradiation, electrons in the VB of TiO₂ can be excited to the CB, resulting in holes in the VB. Because the Fermi level of the Au is lower than that of the CB of TiO₂, the photoexcited electrons in the CB can migrate to the Au. ① This electron transfer generates O₂^{•-} on the Au surface through O₂ reduction, whereas ② the holes in TiO₂ react with OH⁻ to generate •OH.

Under green irradiation, photoinduced electron–hole pairs are generated through the plasmonic resonance effect of the AuNP. Because photoexcited electrons have a very short lifetime in the order of picoseconds, the electron–hole pairs on the AuNP cannot participate in the photocatalytic reaction. However, as AuNPs are attached to the TiO₂ surface, the photoexcited electrons from the AuNPs can effectively migrate to the CB of TiO₂, inhibiting the relaxation and recombination of electrons in the Au/TiO₂ heterostructure. ③ This electron-transfer process facilitates the accumulation of electrons on the TiO₂ surface and promotes the reduction of O₂ to O₂^{•-}. Furthermore, ④ the remaining holes in the AuNPs can react with OH⁻ to produce •OH.

During light irradiation, the generated O₂^{•-} radicals spontaneously react with H₂O to form additional •OH⁶. Therefore, both O₂^{•-} and •OH generated during UV and green irradiation can directly and indirectly participate in the photocatalytic MB degradation process.

We have updated the revised schematic illustration of the photocatalytic degradation mechanism of MB by the Au/TiO₂ heterostructure in the revised manuscript.

[Additional Resource. 3]

Additional Resource. 3. Schematic illustration of the photocatalytic degradation mechanism of MB by Au/TiO₂ heterostructure.

[Reference]

[6] Hirakawa T, Yawata K, Nosaka Y. Photocatalytic reactivity for O₂^{•-} and OH radical formation in anatase and rutile TiO₂ suspension as the effect of H₂O₂ addition. *Applied Catalysis A: General* **325**, 105-111 (2007).

6) According to the text, ROS can affect the strain of Au nanoparticles, and high strain can lead to strong adsorption of active molecules. Is this, logically, a self-enhancing mechanism?

Answer. We appreciate the reviewer for pointing out this important issue regarding the correlation between strain and adsorption. In response to the reviewer's comment, we have incorporated DFT calculations based on strain and OH coverages to deepen our understanding of strain and molecule reactions, as well as the self-enhancing mechanism. Additional Resource. 4a demonstrates the variation in adsorption energy of an OH molecule under nearly isolated conditions (with one OH molecule on a (3x3) surface) relative to the lattice constant of the Au surface. The adsorption energy of OH molecules decreases monotonically with the increasing lattice constant of the Au (111) surface, implying that the surface strain directly impacts the stability and reactivity of the isolated OH molecule. Additional Resource. 4b presents the variation in adsorption energy concerning coverage on both equilibrium (black line) and 1% surface-expanded conditions (red line). In both cases, the adsorption energy tends to stabilize up to 0.66 ML, beyond which it becomes unstable, indicating an optimal coverage of 0.66 ML. Notably, the strained surface consistently exhibits stabilization by approximately 0.1 eV compared to the unstrained surface. Considering the experimental evidence that the adsorption of OH molecules induces the surface strain, it is expected that the activity of OH molecules will increase with the surface strain until reaching optimal coverage.

[Additional Resource. 4]

Additional Resource. 4. a Adsorption energy of OH (0.11 ML) on the Au (111) surface as a function of strain. The negative and positive values indicate compressive and tensile strain of Au (111) surface, respectively. **b** Evolution of adsorption energy as a function of OH coverage at 0% strained and 1% strained Au (111) surface, respectively. In both calculations, the Au (111) surface consists of (3x3) unit cells.

7) When both UV and green light are simultaneously irradiated, the manuscript should offer further explanation regarding the charge transfer pathway, the generation location of ROS, and the reasons for the increase in ROS quantity.

Answer. We thank the reviewer for the comment on the ROS generation mechanism under UV/green irradiation. According to the literature, X-ray absorption spectroscopy shows that the absorption intensity of Ti increases differently with green and green/UV irradiation on Au/TiO₂ particles⁷. In their study, as depicted in Figure 7a of the paper, the researchers used green light with a wavelength of 545 nm as the visible light illumination, similar to the 532 nm wavelength we utilized in our study. Note that the electrons are not generated in TiO₂ under green irradiation due to the wide band gap of TiO₂. The X-ray absorption of Ti under green irradiation indicates that the electrons from only Au transfer to TiO₂. Additionally, green/UV irradiation shows a greater absorption intensity compared to green irradiation, indicating that a greater amount of electrons are accumulated in the CB of TiO₂ under green/UV irradiation. Therefore, efficient charge transfer through green/UV irradiation indicates an important cause for increasing the amount of ROS. Supplementary Fig. 11 in the revised manuscript shows the photoinduced charge transfer and ROS generation process in detail under green/UV irradiation. First, this figure shows the process of generating electrons from the AuNP through the plasmonic resonance effect caused by green irradiation and transferring these electrons to the CB of TiO₂. At the same time, UV irradiation generated electron-hole pairs. The excited

electrons of TiO₂ accumulated in the CB of TiO₂ with the electrons transferred from AuNP at the CB of TiO₂, which could promote the generation of O₂^{•-} in the CB of TiO₂. Additionally, the holes in the VB of both the AuNP and TiO₂ could promote the generation of •OH. We have updated the schematic illustration of the proposed photocatalytic mechanism of Au/TiO₂ heterostructure under green/UV irradiation and further explanations in the revised manuscript.

[Revised Supplementary Fig. 11]

Supplementary Fig. 11. Schematic illustration of the proposed photocatalytic mechanism of Au/TiO₂ heterostructure under green/UV irradiation.

[Revised main manuscript, page 7]

“Under combined green/UV irradiation, these mechanisms operate simultaneously, leading to efficient charge transfer and resulting in the highest ROS generation activities. As shown in Supplementary Fig. 11, the plasmonic resonance effect induced by green irradiation generates electrons in AuNPs, which are then transferred to the CB of TiO₂. At the same time, UV irradiation generates electron-hole pairs in TiO₂. The excited electrons of TiO₂ accumulated in the CB of TiO₂ with the electrons transferred from AuNP, which could promote the generation of O₂^{•-} in the CB of TiO₂. Additionally, the holes in the VB of both the AuNP and TiO₂ could promote the generation of •OH. This charge transfer mechanism demonstrates that the Au/TiO₂ heterostructure is capable of optimal ROS generation through its efficient wavelength absorption ability under green/UV irradiation.”

[Reference]

[7] Yang K-S, *et al.* Plasmon-Induced Visible-Light Photocatalytic Activity of Au Nanoparticle-Decorated Hollow Mesoporous TiO₂: A View by X-ray Spectroscopy. *J Phys Chem C* **122**, 6955-6962 (2018).

8) In the BCDI experiment, green light (20 mW/cm², $\lambda = 532$ nm), UV light (40 mW/cm², $\lambda = 365$ nm), and green/UV light (70 mW/cm²) were used as light sources to examine stress variations with different wavelengths of light. In the photocatalytic degradation experiment, all three light sources had a power density of 30 mW/cm². Since the article is exploring the correlation between photocatalytic activity and stress alterations, why was the input light source density inconsistent?

Answer. We appreciate the reviewer's insightful observation regarding the inconsistency in the input light source density in our experiment. As the reviewer pointed out, maintaining consistent input densities across both BCDI measurements and MB degradation experiments is critical for accurately comparing strain changes dependent on photocatalytic reactions. Unfortunately, due to technical limitations at the BCDI experiment setup, matching the light source densities used in the MB degradation experiments was not feasible. To address this, we conducted MB degradation experiments across a range of light source densities (see Additional Resource 5). These experiments demonstrated that the MB degradation trends were consistent even when the density was varied from 20 mW/cm² to 70 mW/cm². Furthermore, the degree of MB degradation was found to be minimally affected by changes in density under the same light conditions within our experimental framework.

[Additional Resource. 5]

Additional Resource. 5. Photocatalytic degradation of MB as a function of power density (20, 30, 50, and 70 mW/cm²) under green, UV, and green/UV irradiation for Au/TiO₂ heterostructures.

9) For the phenomenon described in Figure 3a, the Gaussian distribution of Au nanoparticle amplitude shifted to the left after illumination, which means that ROS causes lattice distortion and reduces the crystallinity of Au. However, the derivation process and basis for this important conclusion are not clearly explained, making it difficult for readers to understand and accept this conclusion.

Answer. We thank the reviewers for their comments on the correlation between Au lattice distortion and ROS generation. We have conducted spectroscopic experiments including XPS, FT-IR, and Raman spectroscopy to confirm the adsorption of ROS on the Au surface during photocatalytic reaction. High-resolution XPS was performed on the Au samples before and after different wavelength irradiation. It was confirmed that $O_2^{\cdot-}$ and $\cdot OH$ radicals generated during light irradiation are chemisorbed on the Au surface, forming Au-O species. The results from DFT calculations also suggest that the interactions of OH and O_2 with the Au surface can induce strain on the Au (111) surface, whereas the strain resulting from H_2O adsorption is negligible (Supplementary Fig. 26). Therefore, the formation of Au-O bonds causes Au lattice strain, which can consequently reduce the crystallinity of AuNP during photocatalytic reactions. This detailed information, including the experimental details and results demonstrating the lattice distortion, has been added to the revised manuscript.

[Revised main manuscript, page 8]

“To confirm the adsorption of ROS on the Au surface, we conducted spectroscopic experiments including XPS, FT-IR, and Raman spectroscopy during the photocatalytic reaction. High-resolution XPS was performed on the Au samples before and after different wavelength irradiation. As shown in Fig. 2i, two peaks centered at 83.3 eV and 87.1 eV of XPS spectra in all samples represent the $4f_{7/2}$ and $4f_{5/2}$ signals of metallic Au (Au^0), respectively⁵⁰. Before light irradiation, the relatively narrow and symmetrical Au 4f peaks indicate that it mainly exists in the state of Au^0 . However, after different light irradiation, two shoulder peaks at 84.8 eV and 88.4 eV are observed, which are attributed to the oxide state of Au (Au^+)⁵⁰. Furthermore, the O 1s XPS spectra show two additional shoulder peaks at 529.0 eV and 535.0 eV, corresponding to chemisorbed oxygen species from ROS generated during photocatalytic reactions (Fig. 2j)⁵¹. As shown in the Raman spectra in Supplementary Fig. 12a, a new peak at 480 cm^{-1} , which is attributed to the stretching vibrations of Au-O species, appears after different light irradiation⁵². The FT-IR spectra show bands at 3740 cm^{-1} and 1250 cm^{-1} , which can be regarded as the generation of new Au-O species on the Au surface (Supplementary Fig. 12b, c)^{53, 54}. The intensity of each peak tends to increase in the order of green, UV, and green/UV irradiation. This means that the amount of ROS generated is proportional to the amount of adsorption on the Au surface.”

Reviewer #2 (Remarks to the Author):

The paper addresses the in situ imaging of strain evolution in Au nanoparticles supported on a TiO₂ film during photocatalysis. This marks the first instance of measuring the photocatalytic strain evolution of individual particles. I recommend publication after revision. However, I am not fully convinced of the quality of the reconstructions, as the shape and size of the reconstructed particles exhibit significant variation. Evaluation of the reconstruction quality can be performed through the amplitude distributions presented in Fig. 3. A more peaked amplitude histogram indicates a better reconstruction, and the varying histograms suggest differing qualities of reconstructions. Given the diverse qualities of the reconstructions, this variability may introduce artifacts in the surface strain presented in the publication. The authors use the same amplitude threshold (30%) to illustrate the particle's surface strain, despite dissimilar amplitude histograms.

1) It would be interesting to include information about particle stability during Bragg coherent diffraction imaging measurements.

Answer. We thank the reviewer for the comment for the particle stability during Bragg coherent diffraction imaging measurements. As the reviewer points out, for accurate BCDI measurements, it is crucial to ensure that the sample remains stable on the substrate while illuminated by a steady X-ray beam. The particle stability was confirmed by repeatedly measuring the rocking curve of the single AuNP in air, indicating the NP was well fixed to the substrate (Supplementary Fig. 16). The stability was also confirmed in both the amplitude histogram and the particle shape remained constant for 40 min in air (Supplementary Fig. 17a). To show the particle stability in the water environment, BCDI data collected at 40 min were added to the pristine results of Fig. 3a (Supplementary Fig. 17b). A slight difference was observed in the amplitude histogram measured at 30 min in the water environment. However, it is important to note that the particle shape and amplitude histogram measured at 40 min returned to their original state. This suggests that the observed change was likely temporary, possibly due to momentary beam fluctuations or a slight misalignment of the AuNP. Therefore, even under long-term X-ray beam irradiation, the particle shape and amplitude histogram in the water environment remained nearly identical to those observed in air, indicating that the NP is stable during BCDI measurements. The detailed information, including reconstruction images and amplitude histograms demonstrating the stability of the AuNP, has been added in Supplementary Fig. 17 and Supplementary Note 1, as described in the revised manuscript.

[Revised Supplementary Fig. 16]

Supplementary Fig. 16. Bragg peak intensity as a function of rocking curve angle for successive scans of the AuNP in air condition.

[Revised Supplementary Fig. 17]

Supplementary Fig. 17. 3D reconstructed images and corresponding amplitude distribution of a single AuNP measured in **a** air and **b** water conditions. The gray boundary illustrates the shape of the AuNP + air at 10 min.

[Revised main manuscript, page 9]

“A measurement in air and water without irradiation was also performed to determine the beam stability and water radiolysis effect on AuNP (see Supplementary Note 1 and Supplementary Figs. 16-17)^{55, 56.}”

[Revised Supplementary Note 1]

“1) The change in crystallinity of the pristine AuNP: The particle stability was confirmed by repeatedly measuring the rocking curve of the single AuNP in air, indicating the NP was well fixed to the substrate (Supplementary Fig. 16). The stability was also confirmed as both the amplitude histogram and the particle shape remained constant for 40 min in air (Supplementary Fig. 17a). To show the particle stability in the water environment, BCDI data collected at 40 min were added to the pristine results of Fig. 3a (Supplementary Fig. 17b). A slight difference was observed in the amplitude histogram measured at 30 min in the water environment. However, it is important to note that the particle shape and amplitude histogram measured at 40 min returned to their original state. This suggests that the observed change was likely temporary, possibly due to momentary beam fluctuations or a slight misalignment of the AuNP. Therefore, even under long-term X-ray beam irradiation, the particle shape and amplitude histogram in the water environment remained nearly identical to those observed in air, indicating that the NP is stable during BCDI measurements.”

2) Please provide a scale bar in Fig. 2c.

Answer. Thanks for the reviewer’s attentive comments. Following the reviewer’s comment, we have added a scale bar in Figure 2c, as described in the revised manuscript.

[Revised Fig. 2c]

Fig. 2c. EDX mappings of Au, Ti, and O. Scale bar: 50 nm.

3-a) Explain the observed differences in surface strain in Fig. 2h in the pristine state. The surface strain differs for various measurement durations (10 min, 20 min, and 30 min).

Answer. Thanks for the reviewer's attentive comments. The slight strain change observed on the Au surface over time under pristine conditions can be attributed to the adsorption of ROS, which is generated through the water radiolysis, onto the Au surface. However, the amount of ROS generated near the Au surface during only X-ray irradiation is extremely small, and the majority is scavenged by dissolved oxygen or impurities⁵⁶. The constancy of the diffraction intensity and volume across each scan indicates that the very small amount of ROS generated through water radiolysis does not induce substantial changes in the nanoparticle. The detailed information has been added in Supplementary Note 1 as described in the revised manuscript.

[Revised main manuscript, page 9]

"A measurement in air and water without irradiation was also performed to determine the beam stability and water radiolysis effect on AuNP (see Supplementary Note 1 and Supplementary Figs. 16-17)^{55, 56}."

[Revised Supplementary Note 1]

"2) Water radiolysis and photo-ionization: As shown in Fig. 2l, the slight strain change observed on the Au surface over time under pristine conditions can be attributed to the adsorption of ROS, which is generated through water radiolysis, onto the Au surface. However, the amount of ROS generated near the Au surface during only X-ray irradiation is extremely small, and the majority is scavenged by dissolved oxygen or impurities⁵⁶. As shown in Supplementary Figs. 19 and 20, when irradiated with a green light in the water environment, the integrated intensity of the Bragg peak decreased to 85% of I_0 . In contrast, both under green light irradiation in air and in the absence of light in air, the integrated intensity shows no change, remaining at I_0 . The stability of the diffraction intensity and volume across each scan suggests that neither water radiolysis nor photo-ionization induces significant changes in the nanoparticle."

[Revised Reference]

[56] Le Caër S. Water Radiolysis: Influence of Oxide Surfaces on H₂ Production under Ionizing Radiation. *Water* **3**, 235-253 (2011).

3-b) Explain the large variations in the amplitude histogram in the pristine state (see Fig. 3a).

Answer. We appreciate the reviewer's comment regarding the variation in the amplitude histogram. To explain the variation, BCDI data collected at 40 min were added to the pristine results of Fig. 3a (Supplementary Fig. 17b). The particle shape and amplitude histogram of the AuNP measured at 40 min returned to their original state. Supplementary Fig. 17a shows both the amplitude histogram and the particle shape remained constant for 40 min in air. This suggests that the variation was likely temporary, possibly due to momentary beam fluctuations or a slight

misalignment of the AuNP. Therefore, even under long-term X-ray beam irradiation, the particle shape and amplitude histogram in the water environment remained nearly identical to those observed in air, indicating that the nanoparticle is stable during BCDI measurements. The detailed information, including reconstruction images and amplitude histograms demonstrating the stability of the AuNP, has been added in Supplementary Fig. 17 and Supplementary Note 1, as described in the revised manuscript.

[Revised Supplementary Fig. 17]

Supplementary Fig. 17. 3D reconstructed images and corresponding amplitude distribution of a single AuNP measured in **a** air and **b** water conditions. The gray boundary illustrates the shape of the AuNP + air at 10 min.

[Revised main manuscript, page 9]

“A measurement in air and water without irradiation was also performed to determine the beam stability and water radiolysis effect on AuNP (see Supplementary Note 1 and Supplementary Figs. 16-17)^{55, 56.}”

[Revised Supplementary Note 1]

“1) The change in crystallinity of the pristine AuNP: The particle stability was confirmed by repeatedly measuring the rocking curve of the single AuNP in air, indicating the NP was well fixed to the substrate (Supplementary Fig. 16). The stability was also confirmed as both the amplitude histogram and the particle shape remained constant for 40 min in air

(Supplementary Fig. 17a). To show the particle stability in the water environment, BCDI data collected at 40 min were added to the pristine results of Fig. 3a (Supplementary Fig. 17b). A slight difference was observed in the amplitude histogram measured at 30 min in the water environment. However, it is important to note that the particle shape and amplitude histogram measured at 40 min returned to their original state. This suggests that the observed change was likely temporary, possibly due to momentary beam fluctuations or a slight misalignment of the AuNP. Therefore, even under long-term X-ray beam irradiation, the particle shape and amplitude histogram in the water environment remained nearly identical to those observed in air, indicating that the NP is stable during BCDI measurements.”

4-a) In Figs. 2g-h, the particle's shape and size differ. Add the contour of the initial pristine particle to guide the eye in Figs. 2 and 4, as well as in the Supplementary Figures.

Answer. We thank the reviewer for the comment regarding the variability in the reconstructed shape. As the reviewer mentioned, we have updated the contour of the initial pristine particle to guide the eye in Figs. 2 and 4, as well as in the Supplementary Figures in the revised manuscript.

4-b) Are these differences attributed to poor reconstruction quality? The variability in the reconstructed particle's shape is puzzling.

Answer. We thank the reviewer for the comment regarding the reconstruction quality. We confirmed that the adsorption of ROS on Au atoms near the surface of AuNP forms Au-O bonds and introduces lattice distortion and a partial phase transformation, thus Bragg electron density at the transformed region decreases. We think this causes the intensity decrease as well as the left shift of the peak of the reconstructed amplitude histogram.

5) The authors attribute irreversible changes to dissolution and defect dynamics but do not show defect formation during the photocatalytic reaction. Supplementary Figure 8 reveals that green excitation induces voids in the Au particle. Explain how such large voids can be elucidated and why, after turning off the light, the particle no longer exhibits voids (recovery from voids).

Answer. We thank the reviewer for the comment regarding the irreversible change of AuNP. Given the challenges in observing defects due to the spatial resolution limitations of the BCDI experiment in this research, we revised the content and added further explanations about the recovery of voids in the revised manuscript.

[Revised main manuscript, page 10]

“The adsorption of ROS on the surface of AuNPs forms Au-O bonds and causes lattice distortions, including partial phase transformations. In regions where these distortions are significant, the AuNPs may lose crystallinity or exhibit a crystallographic orientation that differs

from the original state of the nanoparticle. These regions fail to contribute to the coherent X-ray diffraction pattern and could potentially result in large deformations, i.e., voids, in the shape of the reconstructed nanoparticles. Note that ROS exhibit high reactivity and possess a short lifetime, typically on the order of microseconds to nanoseconds⁵⁸. Terminating light irradiation, which ceases the generation of ROS and reduces adsorption to the AuNP surface, reduces lattice distortions, potentially eliminating voids and allowing the AuNPs to return to a state similar to their initial condition (Supplementary Fig. 19)^{59,60}. The single AuNP did not undergo significant changes in the air environment, in contrast to the water environment, irrespective of the wavelength of light (Supplementary Fig. 20). These results suggest that the structural deformation of the AuNP is due to the adsorption of the ROS, the products of photocatalytic reactions in water, onto the Au surface⁴⁶."

6) Provide information on how generated reactive oxygen species/radicals are evacuated between one excitation and another (UV/green).

Answer. We thank the reviewer for the comment on the ROS generation mechanism under UV/green irradiation. According to the literature, X-ray absorption spectroscopy shows that the absorption intensity of Ti increases differently with green and green/UV irradiation on Au/TiO₂ particles⁷. In their study, as depicted in Figure 7a of the paper, the researchers used green light with a wavelength of 545 nm as the visible light illumination, similar to the 532 nm wavelength we utilized in our study. Note that the electrons are not generated in TiO₂ under green irradiation due to the wide band gap of TiO₂. The X-ray absorption of Ti under green irradiation indicates that the electrons from only Au transfer to TiO₂. Additionally, green/UV irradiation shows a greater absorption intensity compared to green irradiation, indicating that a greater amount of electrons are accumulated in the CB of TiO₂ under green/UV irradiation. Therefore, efficient charge transfer through green/UV irradiation indicates an important cause for increasing the amount of ROS. Supplementary Fig. 11 in the revised manuscript shows the photoinduced charge transfer and ROS generation process in detail under green/UV irradiation. First, this figure shows the process of generating electrons from the AuNP through the plasmonic resonance effect caused by green irradiation and transferring these electrons to the CB of TiO₂. At the same time, UV irradiation generated electron-hole pairs. The excited electrons of TiO₂ accumulated in the CB of TiO₂ with the electrons transferred from AuNP at the CB of TiO₂, which could promote the generation of O₂^{•-} in the CB of TiO₂. Additionally, the holes in the VB of both the AuNP and TiO₂ could promote the generation of •OH. We have updated the schematic illustration of the proposed photocatalytic mechanism of Au/TiO₂ heterostructure under green/UV irradiation and further explanations in the revised manuscript.

[Revised Supplementary Fig. 11]

Supplementary Fig. 11. Schematic illustration of the proposed photocatalytic mechanism of Au/TiO₂ heterostructure under green/UV irradiation.

[Revised main manuscript, page 7]

“Under combined green/UV irradiation, these mechanisms operate simultaneously, leading to efficient charge transfer and resulting in the highest ROS generation activities. As shown in Supplementary Fig. 11, the plasmonic resonance effect induced by green irradiation generates electrons in AuNPs, which are then transferred to the CB of TiO₂. At the same time, UV irradiation generates electron-hole pairs in TiO₂. The excited electrons of TiO₂ accumulated in the CB of TiO₂ with the electrons transferred from AuNP, which could promote the generation of O₂^{•-} in the CB of TiO₂. Additionally, the holes in the VB of both the AuNP and TiO₂ could promote the generation of •OH. This charge transfer mechanism demonstrates that the Au/TiO₂ heterostructure is capable of optimal ROS generation through its efficient wavelength absorption ability under green/UV irradiation.”

[Reference]

[7] Yang K-S, et al. Plasmon-Induced Visible-Light Photocatalytic Activity of Au Nanoparticle-Decorated Hollow Mesoporous TiO₂: A View by X-ray Spectroscopy. *J Phys Chem C* **122**, 6955-6962 (2018).

7) Does the X-ray beam generate extra radicals or oxygen species?

Answer. We thank the reviewer for the comment for the ROS generation possibility by an X-ray beam. The amount of ROS generated near the Au surface during only X-ray irradiation is extremely small, and the majority is scavenged by dissolved oxygen or impurities⁵⁶. Therefore, even

under long-term X-ray beam irradiation in our experiments, the particle shape and amplitude histogram in the water environment remained nearly identical to those observed in air, indicating that the NP is stable during BCDI measurements. The detailed information, including reconstruction images and amplitude histograms demonstrating the stability of the AuNP, has been added in Supplementary Fig. 17 and Supplementary Note 1 as described in the revised manuscript.

[Revised Supplementary Fig. 17]

Supplementary Fig. 17. 3D reconstructed images and corresponding amplitude distribution of a single AuNP measured in **a** air and **b** water conditions. The gray boundary illustrates the shape of the AuNP + air at 10 min.

[Revised main manuscript, page 9]

“A measurement in air and water without irradiation was also performed to determine the beam stability and water radiolysis effect on AuNP (see Supplementary Note 1 and Supplementary Figs. 16-17)^{55, 56.}”

[Revised Supplementary Note 1]

“1) The change in crystallinity of the pristine AuNP: The particle stability was confirmed by repeatedly measuring the rocking curve of the single AuNP in air, indicating the NP was well fixed to the substrate (Supplementary Fig. 16). The stability was also confirmed as both the amplitude histogram and the particle shape remained constant for 40 min in air (Supplementary Fig. 17a). To show the particle stability in the water environment, BCDI data collected at 40 min were added to the pristine results of Fig. 3a (Supplementary Fig. 17b). A

slight difference was observed in the amplitude histogram measured at 30 min in the water environment. However, it is important to note that the particle shape and amplitude histogram measured at 40 min returned to their original state. This suggests that the observed change was likely temporary, possibly due to momentary beam fluctuations or a slight misalignment of the AuNP. Therefore, even under long-term X-ray beam irradiation, the particle shape and amplitude histogram in the water environment remained nearly identical to those observed in air, indicating that the NP is stable during BCDI measurements.

2) Water radiolysis and photo-ionization: As shown in Fig. 2l, the slight strain change observed on the Au surface over time under pristine conditions can be attributed to the adsorption of ROS, which is generated through water radiolysis, onto the Au surface. However, the amount of ROS generated near the Au surface during only X-ray irradiation is extremely small, and the majority is scavenged by dissolved oxygen or impurities⁵⁶. As shown in Supplementary Figs. 19 and 20, when irradiated with a green light in the water environment, the integrated intensity of the Bragg peak decreased to 85% of I_0 . In contrast, both under green light irradiation in air and in the absence of light in air, the integrated intensity shows no change, remaining at I_0 . The stability of the diffraction intensity and volume across each scan suggests that neither water radiolysis nor photo-ionization induces significant changes in the nanoparticle.”

[Revised Reference]

[56] Le Caër S. Water Radiolysis: Influence of Oxide Surfaces on H₂ Production under Ionizing Radiation. *Water* **3**, 235-253 (2011).

8) Explain how you calculate compressive and strain areas in Supplementary Table 1. Provide the unit. If it is an area, it should be in square meters.

Answer. We thank the reviewer for the comment on the strain area. As in reviewer's comment # 9, we think it is more appropriate to obtain the full width at half maximum of the bulk and surface strain histograms than Supplementary Table 1. Therefore, Supplementary Table 1 was replaced with FWHM values in the revised manuscript.

9) It would be interesting to obtain the full width at half maximum of the bulk and surface strain histograms (Fig. 3b and Supplementary Fig. 11) for comparison.

Answer. We appreciate the reviewer's insightful comments regarding the full width at half maximum (FWHM) of the strain histograms. As the reviewer's comment, we calculated the FWHM value of surface and bulk strain histograms depending on the irradiation conditions (Supplementary Table 1). Under pristine conditions, the FWHM values of surface and bulk strain distributions

show similar values, with an average of 0.018% and 0.017%, respectively. However, during the photocatalytic reactions, the average FWHM value of the surface strain distribution showed an increase in the sequence of green (0.022%), UV (0.023%), and green/UV (0.026%) light irradiation. This trend indicates that the surface strain intensified in proportion to the generated ROS. In addition, the average FWHM value of the bulk strain distribution under photoreaction conditions also increased compared to the value of pristine conditions but showed almost the same value (0.021%) regardless of the wavelength. This indicates that the strain originates at the AuNP surface and is accommodated across the entire bulk region of the AuNP. We have updated the additional results and explanations about the FWHM of the strain distribution in the revised manuscript.

[Revised Supplementary Table 1]

FWHM (%)	Pristine		Green		UV		Green/UV	
	Surface	Bulk	Surface	Bulk	Surface	Bulk	Surface	Bulk
10 min	0.018	0.015	0.023	0.021	0.021	0.020	0.026	0.022
20 min	0.017	0.016	0.022	0.022	0.025	0.022	0.028	0.023
30 min	0.019	0.019	0.023	0.020	0.024	0.021	0.025	0.021
Average	0.018	0.017	0.022	0.021	0.023	0.021	0.026	0.022

Supplementary Table 1. Full width at half maximum value of surface versus bulk strain distributions as a function of the irradiated wavelength.

[Revised main manuscript, page 11]

“Supplementary Table 1 presents the full width at half maximum (FWHM) values of surface versus bulk strain distributions as a function of the irradiated wavelength. Under pristine conditions, the FWHM values of surface and bulk strain distributions show similar values, with an average of 0.018% and 0.017%, respectively. However, during the photocatalytic reactions, the average FWHM value of the surface strain distribution showed an increase in the sequence of green (0.022%), UV (0.023%), and green/UV (0.026%) light irradiation. This trend indicates that the surface strain intensified in proportion to the generated ROS. In addition, the average FWHM value of the bulk strain distribution under photoreaction conditions also increased compared to the value of pristine conditions but showed almost the same value (0.021%) regardless of the wavelength. This suggests that the strain originates at the AuNP surface and is accommodated across the entire bulk region of the AuNP.”

10) In BCDI, heterogeneous strain is observed. Why is the internal strain field not centered at zero under UV and green/UV illuminations? Was the diffraction pattern well-centered before phase retrieval?

Answer. We appreciate the reviewer's insightful comments regarding the internal strain fields observed during UV and green/UV irradiation. To enable accurate comparisons, we increased the number of bins for each experimental condition to prevent distortion in graphical interpretation due to bin size. After adjusting the binning, the internal strain field was observed to be nearly zero-centered under UV and green/UV irradiation. Additionally, analysis of the diffraction pattern confirmed that it was well-centered prior to the phase retrieval process. Consequently, we determined that there were no significant issues with the internal strain field data obtained through the phase retrieval process. This detailed information has been added in Fig. 3 as described in the revised manuscript.

[Revised Fig. 3]

Fig. 3. Statistical distribution of the amplitude and strain of a single AuNP on TiO_2 film from Fig. 2. The amplitude distributions between 0.0 and 1.0 are plotted in (a) with corresponding (b) bulk strain distributions. The lines on the strain plots indicate the threshold values (-0.2% and 0.2%).

11) There are a few misspellings. In the Methods section (Sample preparation), 'heterostructure' is not correctly written, and in the Methods section (Photocatalytic degradation of MB), replace 'A UV lamp were used' with 'A UV lamp was used.'

Answer. We appreciate the concern of the reviewer. The manuscript was revised to reflect your comment.

[Revised main manuscript, page 15]

"The preparation of Au/TiO₂ heterostructure was conducted following the procedure."

[Revised main manuscript, page 16]

"The morphology, size, and elemental composition of the Au/TiO₂ heterostructures were characterized using extreme high resolution scanning electron microscope (FE-SEM, Gemini 3, Zeiss) at the Center for Polymers and Composite Materials, Hanyang University, Korea."

[Revised main manuscript, page 16]

"A UV lamp (Asone, LUV-4) was used for UV irradiation."

Reviewer #3 (Remarks to the Author):

Relevance: Relevant for wider audiences, due to the application of Bragg coherent X-ray diffraction imaging for in situ lattice displacement of nanostructures as a direct result of metal/adsorbate interactions. An insight into these displacements is important for understanding the behavior of nanostructures under reaction conditions and could also find applications in the fields of electrocatalysis, energy storage and nano fabrication.

Cited references: Diverse demographic background of the cited literature. All citations accessible to the broader public. Authors tend to cite review articles instead of original publications, especially when considering the used diffraction technique and the mechanism of hot electron injection from excited Au NPs into the TiO₂ film. In several sentences the cited literature does not fit the general message of the sentence.

Recommendation: Accept with major revisions.

The manuscript "In situ photocatalytic strain evolution of a single Au nanoparticle in Au/TiO₂ heterostructures" investigates the deformation of an individual gold nanoparticle supported on TiO₂ under the influence of UV and green irradiation. The authors show that the photocatalytic generation of reactive oxygen species, triggered by irradiation causes a lattice displacement in the investigated gold nanoparticle. Initially the photocatalytic activity of the TiO₂ supported Au Nps under green, UV and a mix of green and UV light is compared with an Au Np, suspension and bare TiO₂. The results of both photocatalytic degradation of methylene blue and fluorescence of 7-OH-coumarin indicate an increased production of reactive oxygen species (ROS) on the heterosystem, with the highest activity under UV/green irradiation. The authors suggest an electron transfer from TiO₂ into the Au NP in the UV case and a hot electron injection from Au into the TiO₂ under green light. Therefore, OH radicals are generated on the electron depleted surface of the TiO₂ film while O₂⁻ radicals are generated at the negatively charged Au surface in the first case and vice versa in the latter. A similar behavior in OH and O₂⁻ radical production for both green and UV irradiation is suggested to verify the electron transfer between the Au and TiO₂ in both cases. In the second part the influence of this ROS generation on the structure of the gold nanoparticles is studied by in situ Bragg coherent X-ray diffraction imaging (BCDI). The authors show that the lattice strain of both the surface and the bulk of the Au nanoparticles increases with increased activity for ROS generation. In accordance with the decreasing Bragg diffraction maxima under different irradiation times, this indicates an adsorbent dependent restructuring of the Au NPs surface. The authors also show that the surface strain leads to a lattice displacement throughout the bulk of the nanoparticle. Lastly, density functional theorem was used to verify that the adsorption of OH could cause the restructuring of an Au surface.

The authors concluded that the high reactivity during UV/green irradiation is caused by an increased absorption efficiency of the Au/TiO₂ heterostructure, leading to higher negative surface charge density at the Au NPs, generating more active sites for ROS generation. This leads to an increased strain on

the Au surface, causing further lattice displacement, which affects the Au adsorption energies, further altering catalytic activity. It is suggested that this indicates the Au NP as main source of active sites for the ROS generation. Lastly the irreversibility of the structural change under photocatalytic conditions is discussed and how it could hint on catalyst deactivations.

[Major revisions]

1) The proposed mechanism for UV/green mixed light is difficult to understand, as the proposed mechanism for green light includes migration of hot electrons to the TiO₂ and the UV mechanism proposes photoelectrons migrating from TiO₂ to Gold.

a. Following that logic, a mixed light should not increase efficiency as the two mechanisms are competing. Furthermore, the cited literature does only consider the visible spectrum [56] or uses a broadband spectrum that also exhibits a larger range in the UV part (280-400 vs. 200 to 1500 nm) which makes a direct comparison difficult [46]. This part needs further clarification.

Answer. We thank the reviewer for the comment on the ROS generation mechanism under UV/green irradiation. According to the literature, X-ray absorption spectroscopy shows that the absorption intensity of Ti increases differently with green and green/UV irradiation on Au/TiO₂ particles⁷. In their study, as depicted in Figure 7a of the paper, the researchers used green light with a wavelength of 545 nm as the visible light illumination, similar to the 532 nm wavelength we utilized in our study. Note that the electrons are not generated in TiO₂ under green irradiation due to the wide band gap of TiO₂. The X-ray absorption of Ti under green irradiation indicates that the electrons from only Au transfer to TiO₂. Additionally, green/UV irradiation shows a greater absorption intensity compared to green irradiation, indicating that a greater number of electrons are accumulated in the CB of TiO₂ under green/UV irradiation. Therefore, efficient charge transfer through green/UV irradiation indicates an important cause for the increasing amount of ROS. Supplementary Fig. 11 in the revised manuscript shows the photoinduced charge transfer and ROS generation process in detail under green/UV irradiation. First, this figure shows the process of generating electrons from the AuNP through the plasmon resonance effect caused by green irradiation and transferring these electrons to the CB of TiO₂. At the same time, UV irradiation generated electron-hole pairs. The excited electrons of TiO₂ accumulated in the CB of TiO₂ with the electrons transferred from AuNP at the CB of TiO₂, which could promote the generation of O₂^{•-} in the CB of TiO₂. Additionally, the holes in the VB of both the AuNP and TiO₂ could promote the generation of •OH. We have updated the schematic illustration of the proposed photocatalytic mechanism of Au/TiO₂ heterostructure under green/UV irradiation and further explanations in the revised manuscript.

[Revised Supplementary Fig. 11]

Supplementary Fig. 11. Schematic illustration of the proposed photocatalytic mechanism of Au/TiO₂ heterostructure under green/UV irradiation.

[Revised main manuscript, page 7]

“Under combined green/UV irradiation, these mechanisms operate simultaneously, leading to efficient charge transfer and resulting in the highest ROS generation activities. As shown in Supplementary Fig. 11, the plasmonic resonance effect induced by green irradiation generates electrons in AuNPs, which are then transferred to the CB of TiO₂. At the same time, UV irradiation generates electron-hole pairs in TiO₂. The excited electrons of TiO₂ accumulated in the CB of TiO₂ with the electrons transferred from AuNP, which could promote the generation of O₂^{-•} in the CB of TiO₂. Additionally, the holes in the VB of both the AuNP and TiO₂ could promote the generation of •OH. This charge transfer mechanism demonstrates that the Au/TiO₂ heterostructure is capable of optimal ROS generation through its efficient wavelength absorption ability under green/UV irradiation.”

[Reference]

[7] Yang K-S, et al. Plasmon-Induced Visible-Light Photocatalytic Activity of Au Nanoparticle-Decorated Hollow Mesoporous TiO₂: A View by X-ray Spectroscopy. *J Phys Chem C* **122**, 6955-6962 (2018).

b. Additional DFT studies that investigate the implication of O₂^{-•} radical generation on lattice strain of Au nanoparticles need to be done to clarify if the Au NPs function as hole or electron carrier during UV/green irradiation.

Answer. We thank the reviewer for the comment regarding the lattice strain of AuNP induced by O₂ radicals. Based on the reviewer's comments, we performed additional DFT calculations of O₂ adsorption on the Au (111) surface. As shown in Supplementary Fig. 26b, O₂ was adsorbed on the Au bridge site with an adsorption energy of -0.653 eV. Furthermore, the adsorbed Au atoms expanded by up to 2.33 pm, which is 13.7 times larger than the 0.17 pm expansion induced by H₂O adsorption. The results from DFT calculations suggest that the interactions of OH and O₂ with the Au surface can induce strain on the Au (111) surface, whereas the strain resulting from H₂O adsorption is negligible. We have updated the additional DFT calculation about O₂ adsorption on the Au (111) surface in the revised manuscript.

[Revised Supplementary Fig. 26]

Supplementary Fig. 26. Adsorption configurations of OH, O₂, and H₂O on AuNP and their associated displacement response. **a-c** Optimized adsorption configurations of OH, O₂, and H₂O on the Au (111) surface: (left) top view and (right) side view of the most stable structure. **d-f** In-plane distortion of the top-layer Au atoms for the adsorption of OH, O₂, and H₂O. The in-plane distortion of the top-layer Au surface is determined from the top 9 Au atoms. The red and blue colors represent the out-of-plane directions along the z-axis, with red indicating upward movement in the +z direction away from the center of mass, and blue indicating downward movement in the -z direction toward the surface.

[Revised main manuscript, page 13]

“We considered OH, O₂, and H₂O as potential adsorbates on the Au (111) surface^{62, 63}.”

[Revised main manuscript, page 13]

“The O₂ molecule also exhibited stability at the Au bridge site, with an adsorption energy of -0.653 eV, causing the Au atoms to expand by 2.33 pm from their original positions (Supplementary Fig. 26b, e). In contrast, H₂O adsorbed nearly parallel to the top site of the Au (111) surface via oxygen atoms with an adsorption energy of -0.112 eV, exhibiting the lowest adsorption energy among the considered molecules (Supplementary Fig. 26c). The surface expansion of the H₂O adsorbed Au surface was only 0.170 pm (Supplementary Fig. 26f), which is 29.4 and 13.7 times smaller than those observed for OH and O₂ adsorption, respectively. These results suggest that the interactions of OH and O₂ with the Au surface can induce strain on the Au (111) surface, whereas the strain due to H₂O adsorption is negligible.”

2) Risk of over interpretation due to the focus on one single Au nanoparticle. While the capabilities to understand the structural evolution of a single particle is indeed impressive, the proposed evolution should be demonstrated on at least three different particles to verify that the observed behavior is not an outlier.

Answer. We appreciate the reviewer's comments regarding the risk of over-interpretation of structural evolution. In response to the reviewer's comments, we have added strain distribution, average lattice spacing variation ($\Delta\bar{d}$), and average strain field data for ellipsoidal AuNP, which is different in shape from the AuNP depicted in Fig. 2. As shown in Supplementary Figs. 23 and 24, in the case of pristine conditions, both surface and bulk strain distributions remained below critical values (-0.4% and +0.4%). However, after light irradiation, the high strain area significantly increased due to ROS adsorption on the Au surface. Additionally, both the surface and bulk strain distributions showed a tendency for the high tensile region to increase more significantly than the high compressive region, indicating that the strain initiated at the surface propagated to the interior of the AuNP. Supplementary Fig. 25 shows that $\Delta\bar{d}$ and average strain increase in the order of green, UV, and green/UV irradiation, indicating that the lattice distortion is proportional to the amount of ROS adsorption on the Au surface. These results are consistent with the structural evolution results obtained from the AuNP in Fig. 2. As suggested by the reviewer, we attempted to avoid the risk of over-interpretation by performing BCDI measurements on multiple nanoparticles; however, we could not proceed further due to limited beam time. Unfortunately, since the Advanced Photon Source (APS) is currently in the process of upgrading to a 4th generation high-energy light source, BCDI experiments via the 34-ID-C beamline are currently difficult. Nevertheless, we believe that the observation of identical structural evolution trends in AuNPs with different shapes may enhance our understanding of the structure-activity relationship in AuNPs during photocatalytic reactions.

This detailed information has been added to the revised manuscript.

[Revised Supplementary Fig. 23]

Supplementary Fig. 23. Surface strain distribution of a single ellipsoidal AuNP. The lines on the strain plots indicate the threshold values (-0.4% and 0.4%).

[Revised Supplementary Fig. 24]

Supplementary Fig. 24. Bulk strain distribution of a single ellipsoidal AuNP. The lines on the strain plots indicate the threshold values (-0.4% and 0.4%).

[Revised Supplementary Fig. 25]

Supplementary Fig. 25. **a** 3D reconstructed images at a 30% amplitude threshold with lattice displacement along the [111] direction projected on the isosurfaces. The gray boundary depicts the pristine cross-section in (a). **b** The change in $\Delta \bar{d}$ is calculated using the formula $\Delta \bar{d} = \bar{d}_{111} - \bar{d}_{111}^*$, where the \bar{d}_{111} is the average lattice spacing measured across the AuNP and \bar{d}_{111}^* is average the lattice spacing of the first measured pristine AuNP. **c** Cross-sectional views of the internal strain field at the dashed line box in Supplementary Fig. 25a. **d** Average strain field for the internal plane in Supplementary Fig. 25c under green, UV, and green/UV irradiation for 30 min. The gray boundary illustrates the shape of the AuNP in pristine condition at 10 min.

[Revised main manuscript, page 13]

“To increase the reliability of the results through BCDI measurement, we also analyzed a single ellipsoidal AuNP, which differs in shape from the AuNP shown in Fig. 2. As shown in Supplementary Figs. 23-25, we observed consistent trends in structural evolution across the AuNPs of different types.

3) Lack or investigation of surface roughness changes and particle radius and volume changes, under different irradiation conditions. A depiction of size evaluation is typically standard procedure for BCDI investigations.

Answer. We thank the reviewers for their comments on the morphological changes under different irradiation conditions. We calculated the volume change of AuNPs to confirm the morphological changes resulting from the photocatalytic reactions. As shown in Supplementary Fig. 18, under pristine conditions, the AuNPs exhibited no significant volume change over time. However, when the light source was irradiated for 30 minutes, the volume of AuNPs gradually decreased. Specifically, the volume of the AuNPs decreased significantly in sequence with green, UV, and green/UV light irradiation, indicating that lattice distortion in Au is proportional to the quantity of Au-O bonds generated. Consequently, the Au-O bonds formed on the surface of the AuNPs are likely to contribute to a reduction in the crystallinity or a variation in the crystallographic orientation of the Au surface. The reduction or variation is believed to have influenced the observed changes in surface roughness (i.e., voids) and volume loss as detected by BCDI. We have updated the volume change of AuNP under different irradiation conditions and further explanations in the revised manuscript.

[Revised Supplementary Fig. 18]

Supplementary Fig. 18. Volume changes of the AuNP as a function of irradiation time under green, UV, and green/UV irradiation. The volume of the AuNP is evaluated from the 3D reconstructed images.

[Revised main manuscript, page 10]

“Furthermore, the AuNP exhibited morphological changes such as volume reduction and surface roughening due to the photocatalytic reaction. As shown in Supplementary Fig. 18, under pristine conditions, the AuNPs did not exhibit significant volume changes over time. However, when the light source was irradiated for 30 minutes, the volume of AuNPs gradually decreased. The volume of the AuNPs decreased significantly in sequence with green (95%),

UV (93%), and green/UV (90%) irradiation, indicating that lattice distortion in Au is proportional to the quantity of Au-O bonds generated.”

4) The irreversibility in structural change during the photocatalytic process is not verified by a secondary method such as SEM. Post experimental SEM would also help to verify if the in situ BDCI experiments follow the same trends in regards of deformation as a blank experiment.

Answer. We thank the reviewer for the valuable comment. As shown in Supplementary Fig. 19, during the photoreaction, the intensity of the AuNP decreased by up to 85%, but it recovered to 98% after turning off the light. Even after a considerable amount of time, the intensity did not fully return to 100%, which was interpreted as an irreversible change. However, the subtle irreversible changes on the surface under our experimental conditions were difficult to observe using SEM due to the limitations of SEM’s analytical capabilities (Additional Resource. 6). According to the literature, researchers were unable to analyze the morphological changes of the acid-treated nanoparticles by SEM after BDCI measurements, and only a slight change could be observed using TEM⁸ (Supplementary Fig. 9, *Nat Commun* **10**, 703 (2019)). Because it is a heterogeneous catalyst composed of the AuNP on the TiO₂ substrate in our experiments, the same AuNP after the BDCI measurement was not suitable for preparing a sample for TEM analysis. Therefore, we have conducted new spectroscopic experiments, including XPS, FT-IR, and Raman spectroscopy (Fig. 2i, j and Supplementary Fig. 12). It was confirmed that O₂^{•-} and •OH radicals generated during light irradiation were chemically adsorbed on the Au surface to form Au-O species. The results from DFT calculations support that generation of the Au-O species on the Au surface can induce strain on the surface. Therefore, we can confirm that the adsorption of ROS on Au atoms near the surface of AuNP forms Au-O species and introduces structural change.

[Additional Resource. 6]

Additional Resource. 6. SEM image of AuNPs on the TiO₂ substrate (a) in air and (b) in the water environment obtained before and after light irradiation.

[Revised Fig. 2]

Fig. 2. Characterization of a single AuNP deposited on TiO₂ film. SEM images of (a) top-view and (b) cross-view of the Au/TiO₂ heterostructure. (c) EDX mappings of Au, Ti, and O. Scale bar: 50 nm. Photocatalytic degradation of MB on (d) AuNP, (e) TiO₂, and (f) Au/TiO₂ heterostructure under green, UV, and green/UV irradiation for 60 min. ESR analysis of (g) •OH and (h) O₂^{•-} generation under green, UV, and green/UV irradiation after 1 h. DMPO was used as the spin-trapping agent. High-resolution XPS spectra of (i) Au 4f and (j) O 1s. (k) 3D reconstructed images of AuNP at a 30% amplitude threshold. The gray boundary illustrates

the shape of the AuNP in the pristine condition at 10 min. I 3D strain images of AuNP associated with compressive (blue) and tensile (red) strains. The lattice strain along the [111] direction was projected onto the isosurfaces.

[Revised main manuscript, page 8]

“To confirm the adsorption of ROS on the Au surface, we conducted spectroscopic experiments including XPS, FT-IR, and Raman spectroscopy during the photocatalytic reaction. High-resolution XPS was performed on the Au samples before and after different wavelength irradiation. As shown in Fig. 2i, two peaks centered at 83.3 eV and 87.1 eV of XPS spectra in all samples represent the $4f_{7/2}$ and $4f_{5/2}$ signals of metallic Au (Au^0), respectively⁵⁰. Before light irradiation, the relatively narrow and symmetrical Au 4f peaks indicate that it mainly exists in the state of Au^0 . However, after different light irradiation, two shoulder peaks at 84.8 eV and 88.4 eV are observed, which are attributed to the oxide state of Au (Au^+)⁵⁰. Furthermore, the O 1s XPS spectra show two additional shoulder peaks at 529.0 eV and 535.0 eV, corresponding to chemisorbed oxygen species from ROS generated during photocatalytic reactions (Fig. 2j)⁵¹. As shown in the Raman spectra in Supplementary Fig. 12a, a new peak at 480 cm^{-1} , which is attributed to the stretching vibrations of Au-O species, appears after different light irradiation⁵². The FT-IR spectra show bands at 3740 cm^{-1} and 1250 cm^{-1} , which can be regarded as the generation of new Au-O species on the Au surface (Supplementary Fig. 12b, c)^{53, 54}. The intensity of each peak tends to increase in the order of green, UV, and green/UV irradiation. This means that the amount of ROS generated is proportional to the amount of adsorption on the Au surface.”

[Reference]

[8] Yuan K, *et al.* Oxidation induced strain and defects in magnetite crystals. *Nat Commun* **10**, 703 (2019).

5) Availability of RAW data and used methods: The authors claim on line 416 that all relevant supporting data are available within main text or supplementary information. But neither manuscript contains raw data from the degradation experiments nor depicts the meta data for the used phase retrieval algorithm. This makes the evaluation of data reconstruction validity difficult as the errors are unknown.

Answer. We thank the reviewer for the valuable comment. In response to the reviewer’s valuable comment, we have added raw data from the degradation experiments and meta data for the phase retrieval algorithm. The manuscript was revised to reflect your comment.

[Revised Supplementary Fig. 4]

Supplementary Fig. 4. Photocatalytic degradation of MB on AuNP under green, UV, and green/UV irradiation for 60 min.

[Revised Supplementary Fig. 5]

Supplementary Fig. 5. Photocatalytic degradation of MB on TiO₂ under green, UV, and green/UV irradiation for 60 min.

[Revised Supplementary Fig. 6]

MB degradation

Supplementary Fig. 6. Photocatalytic degradation of MB on Au/TiO₂ under green, UV, and green/UV irradiation for 60 min.

[Revised Supplementary Fig. 7]

Coumarin

Supplementary Fig. 7. Time-dependent fluorescence intensity of 7-OH-coumarin for detecting the •OH generation of the Au/TiO₂ heterostructure irradiated by green, UV, and green/UV.

[Revised Supplementary Fig. 8]

Supplementary Fig. 8. Time-dependent fluorescence intensity of DHE for detecting the $O_2^{\cdot -}$ generation of the Au/TiO₂ heterostructure irradiated by green, UV, and green/UV.

[Revised main manuscript, page 5]

“The absorbance of MB at 664 nm was measured to calculate its concentration (Supplementary Figs. 4-6).”

[Revised main manuscript, page 6]

“The generation activities of $\cdot OH$ and $O_2^{\cdot -}$ were quantified by measuring the emission intensities of 7-OH-coumarin (455 nm) and dihydroethidium (DHE) (420 nm), respectively (Supplementary Figs. 7 and 8)^{32, 44}.”

[Revised Supplementary Fig. 15]

Supplementary Fig. 15. Schematic diagram of phase retrieval algorithm combined with Error reduction (ER) and Hybrid input-output (HIO) using coherent diffraction pattern.

[Revised main manuscript, page 9]

“A phase-retrieval algorithm was employed to determine the 3D electron density and local strain field of a single AuNP (Supplementary Fig. 15).”

[Citation related revision]

1) On line 57 and 58 the cited literature does not discuss ROS generation in Au/TiO₂ heterostructures

Answer. We thank the reviewer for their comment on the citation for ROS generation in Au/TiO₂ heterostructures. In response to the reviewer's comment, we have updated the reference regarding ROS generation in Au/TiO₂ heterostructures in the revised manuscript. We are thankful for the opportunity to correct these errors to ensure the accuracy of our manuscript.

[Revised main manuscript, page 3]

“In particular, AuNPs in Au/TiO₂ heterostructures exhibit a distinctive reactive oxygen species (ROS) generation pathway depending on the excitation wavelength^{15, 16}.”

[Revised reference]

[15] Tan TH, Scott J, Ng YH, Taylor RA, Aguey-Zinsou K-F, Amal R. Understanding Plasmon and Band Gap Photoexcitation Effects on the Thermal-Catalytic Oxidation of Ethanol by TiO₂-Supported Gold. *ACS Catal* **6**, 1870-1879 (2016).

[16] Wang Y, Yang C, Chen A, Pu W, Gong J. Influence of yolk-shell Au@TiO₂ structure induced photocatalytic activity towards gaseous pollutant degradation under visible light. *Appl Catal B* **251**, 57-65 (2019).

2) On line 134 the reference 43 does not contain any of the cited information and discusses biological implications of coumarin on biological systems, no instances of gold or ROS appear in the article, also reference 44 does not discuss strain effects

Answer. We thank the reviewer for the comment on the inadequacy of references 43 and 44. Based on the reviewer's comments, we have updated the references to accurately support the ROS generation and strain effects in the revised manuscript.

[Revised main manuscript, page 6]

“We further investigated the ROS generation activity (ROS = •OH and O₂^{-•}) of the Au/TiO₂ heterostructure to study the impact of the ROS on the AuNP structure^{16, 43}.”

[Revised reference]

[16] Wang Y, Yang C, Chen A, Pu W, Gong J. Influence of yolk-shell Au@TiO₂ structure induced photocatalytic activity towards gaseous pollutant degradation under visible light. *Appl Catal B* **251**, 57-65 (2019).

[43] Choi S, *et al.* In Situ Strain Evolution on Pt Nanoparticles during Hydrogen Peroxide Decomposition. *Nano Lett* **20**, 8541-8548 (2020).

3) On line 152, which is crucial for the explanation of the mechanism, the same two references 43 and 44 are cited, which still do not discuss relaxation and recombination effects.

Answer. We appreciate the reviewer's comments on references 43 and 44 regarding ROS generation mechanisms. We have updated the references in the revised manuscript to accurately reflect the discussion of relaxation and recombination effects.

[Revised main manuscript, page 7]

"However, as AuNPs are attached to the TiO₂ surface, the photoexcited electrons from the AuNPs can effectively migrate to the conduction band (CB) of TiO₂, inhibiting the relaxation and recombination of electrons in the Au/TiO₂ heterostructure^{40, 47, 48.}"

[Revised reference]

[40] Du L, Furube A, Yamamoto K, Hara K, Katoh R, Tachiya M. Plasmon-Induced Charge Separation and Recombination Dynamics in Gold-TiO₂ Nanoparticle Systems: Dependence on TiO₂ Particle Size. *J Phys Chem C* **113**, 6454-6462 (2009).

[47] Bian Z, Tachikawa T, Zhang P, Fujitsuka M, Majima T. Au/TiO₂ Superstructure-Based Plasmonic Photocatalysts Exhibiting Efficient Charge Separation and Unprecedented Activity. *J Am Chem Soc* **136**, 458-465 (2014).

[48] Li X, *et al.* Efficient hole abstraction for highly selective oxidative coupling of methane by Au-sputtered TiO₂ photocatalysts. *Nat Energy* **8**, 1013-1022 (2023).

4) Line 154 the reference 10 does only loosely refer to the mechanisms described in the sentence

Answer. We thank the reviewer for this comment. We have updated the reference to directly support the mechanism of ROS generation.

[Revised main manuscript, page 7]

"This electron-transfer process facilitates the accumulation of electrons on the TiO₂ surface and promotes the reduction of O₂ to O₂^{-•}^{10, 48.}"

[Revised reference]

[48] Li X, *et al.* Efficient hole abstraction for highly selective oxidative coupling of methane by Au-sputtered TiO₂ photocatalysts. *Nat Energy* **8**, 1013-1022 (2023).

5) Also 154 the cited article [11] is a review article that states in one single sentence that energetic holes retained on small Au nanoparticles have sufficient energy to drive the oxygen-evolution-half reaction, citing two other publications.

Answer. We thank the reviewer for the valuable comment. According to the reviewer's comment, we have updated the reference regarding the OH generation through holes in AuNPs in the revised manuscript.

[Revised main manuscript, page 7]

“Furthermore, the remaining holes in the AuNPs can react with OH⁻ to produce •OH^{39, 48, 49}.”

[Revised reference]

[48] Li X, *et al.* Efficient hole abstraction for highly selective oxidative coupling of methane by Au-sputtered TiO₂ photocatalysts. *Nat Energy* **8**, 1013-1022 (2023).

[Minor revisions]

1) In Figures 3g, 4a and Supplementary figures 8 and 9 the orientation of the reconstructed iso surface is sometimes changed between the different conditions or irradiation times. This makes a comparison difficult and the Figures confusing. To circumvent this an indication of the particle orientation would be helpful.

Answer. We are grateful for the reviewer's attentive comment. For better understanding, we have added the tripod axes to all figures in the revised manuscript.

2) The difference in depicted strain structure between Figure 2 and Supplementary Fig.10 is not clear.

Answer. We thank the reviewer's comment regarding the difference of strain structure between Fig. 2 and Supplementary Fig.10. Following the reviewer's comment, we have added more detail on the Supplementary Fig. 21 in the revised manuscript.

[Revised Supplementary Fig. 21]

Supplementary Fig. 21. 3D strain images depicting the highly compressive (blue, strain < -0.2% shown in Fig. 3b) and tensile (red, strain > 0.2% shown in Fig. 3b) strain regions during the photocatalytic reaction. The particle shape is shown as a semi-transparent grey isosurface. The illumination conditions are indicated in the figures. The gray boundary illustrates the shape of the AuNP in pristine condition at 10 min.

3) For Figure 3 binning is not kept constant over the different irradiation times, making a comparison difficult.

Answer. We are grateful for the reviewer's attentive comments. For better understanding, Fig. 3 has been modified to keep the number of bins. The manuscript was revised to reflect your comment.

[Revised Fig. 3]

Fig. 3. Statistical distribution of the amplitude and strain of a single AuNP on TiO₂ film from Fig. 2. The amplitude distributions between 0.0 and 1.0 are plotted in (a) with corresponding (b) bulk strain distributions. The lines on the strain plots indicate the threshold values (-0.2% and 0.2%).

4-a) The data in Figure 3 a indicates a change in crystallinity of the pristine Au NP in water without irradiation over 30 minutes. This change is not discussed and might hint towards beam damage during X-ray experiments, especially as radiolysis is known to generate ROS in water [<https://doi.org/10.3390/w3010235>]. Additionally, photo ionization of the Au Nps or the TiO₂ film, could alter green and UV ionization rates. From the given data synergistic interactions between X-ray irradiation and the observed photocatalytic activity cannot be.

Answer. We thank the reviewer for the comment for the particle stability during Bragg coherent diffraction imaging measurements.

1) The change in crystallinity of the pristine AuNP: As the reviewer points out, for accurate BCDI measurements, it is crucial to ensure that the sample remains stable on the substrate while illuminated by a steady X-ray beam. The particle stability was confirmed by repeatedly measuring the rocking curve of the single AuNP in air, indicating the NP was well fixed to the substrate (Supplementary Fig. 16). The stability was also confirmed as both the amplitude histogram and the particle shape remained constant for 40 min in air (Supplementary Fig. 17a). To show the particle stability in the water environment, BCDI data collected at 40 min were added to the pristine results of Fig. 3a (Supplementary Fig. 17b). A slight difference was observed in the amplitude histogram measured at 30 min in the water environment. However, it is important to note that the particle shape and amplitude histogram measured at 40 min returned to their original state. This suggests that the observed change was likely temporary, possibly due to momentary beam fluctuations or a slight misalignment of the AuNP. Therefore, even under long-term X-ray beam irradiation, the particle shape and amplitude histogram in the water environment remained nearly identical to those observed in air, indicating that the NP is stable during BCDI measurements.

2) Water radiolysis and photo-ionization: We are confident that your insights on water radiolysis and photoionization will significantly contribute to the comprehensive discussion in this research. As shown in Fig. 2l, the slight strain change observed on the Au surface over time under pristine conditions can be attributed to the adsorption of ROS, which is generated through water radiolysis, onto the Au surface. However, the amount of ROS generated near the Au surface during only X-ray irradiation is extremely small, and the majority is scavenged by dissolved oxygen or impurities⁵⁶. As shown in Supplementary Figs. 19 and 20, when irradiated with a green light in the water environment, the integrated intensity of the Bragg peak decreased to 85% of I_0 . In contrast, both under green light irradiation in air and in the absence of light in air, the integrated intensity shows no change, remaining at I_0 . The stability of the diffraction intensity and volume across each scan suggests that neither water radiolysis nor photo-ionization induces significant changes in the nanoparticle. The detailed information, including reconstruction images and amplitude histograms demonstrating the stability of AuNP, has been added in Supplementary Fig. 17 as described in the revised manuscript.

[Revised Supplementary Fig. 16]

Supplementary Fig. 16. Bragg peak intensity as a function of rocking curve angle for successive scans of the AuNP in air condition.

[Revised Supplementary Fig. 17]

Supplementary Fig. 17. 3D reconstructed images and corresponding amplitude distribution of a single AuNP measured in **a** air and **b** water conditions. The gray boundary illustrates the shape of the AuNP + air at 10 min.

[Revised main manuscript, page 9]

“A measurement in air and water without irradiation was also performed to determine the beam stability and water radiolysis effect on AuNP (see Supplementary Note 1 and Supplementary Figs. 16-17)^{55, 56.}”

[Revised Supplementary Note 1]

“1) The change in crystallinity of the pristine AuNP: The particle stability was confirmed by repeatedly measuring the rocking curve of the single AuNP in air, indicating the NP was well fixed to the substrate (Supplementary Fig. 16). The stability was also confirmed as both the amplitude histogram and the particle shape remained constant for 40 min in air (Supplementary Fig. 17a). To show the particle stability in the water environment, BCDI data collected at 40 min were added to the pristine results of Fig. 3a (Supplementary Fig. 17b). A slight difference was observed in the amplitude histogram measured at 30 min in the water environment. However, it is important to note that the particle shape and amplitude histogram measured at 40 min returned to their original state. This suggests that the observed change was likely temporary, possibly due to momentary beam fluctuations or a slight misalignment of the AuNP. Therefore, even under long-term X-ray beam irradiation, the particle shape and amplitude histogram in the water environment remained nearly identical to those observed in air, indicating that the NP is stable during BCDI measurements.

2) Water radiolysis and photo-ionization: As shown in Fig. 2l, the slight strain change observed on the Au surface over time under pristine conditions can be attributed to the adsorption of ROS, which is generated through water radiolysis, onto the Au surface. However, the amount of ROS generated near the Au surface during only X-ray irradiation is extremely small, and the majority is scavenged by dissolved oxygen or impurities⁵⁶. As shown in Supplementary Figs. 19 and 20, when irradiated with a green light in the water environment, the integrated intensity of the Bragg peak decreased to 85% of I_0 . In contrast, both under green light irradiation in air and in the absence of light in air, the integrated intensity shows no change, remaining at I_0 . The stability of the diffraction intensity and volume across each scan suggests that neither water radiolysis nor photo-ionization induces significant changes in the nanoparticle.”

[Revised Reference]

[56] Le Caër S. Water Radiolysis: Influence of Oxide Surfaces on H₂ Production under Ionizing Radiation. *Water* **3**, 235-253 (2011).

4-b) Furthermore, no information of the photon flux is given, making the possibility of beam damage difficult to assess.

Answer. We appreciate the reviewer's comment about photon flux, considering the possibility of beam damage. Following the Reviewer's comment, we have added more detail on the photon flux in the revised manuscript.

[Revised main manuscript, page 18]

“A coherent X-ray beam, with an energy of 10 keV and a flux of 5×10^9 photons/s, was

precisely focused to dimensions of 450 nm × 520 nm (horizontally × vertically) using Kirkpatrick-Baez mirrors.”

5) In the introduction on line 73 there is a sudden jump in reasoning from active sites to strain studies, without a further discussion of how these two parameters relate to each other.

Answer. We appreciate the reviewer's comments regarding the relevance of active sites to strain studies. Following the reviewer's comment, the manuscript was revised to reflect your comment.

[Revised main manuscript, page 4]

“Nevertheless, a comprehensive understanding of the structural deformation and the catalytic activity of individual nanoparticles is not clearly understood.”

[Revised main manuscript, page 4]

“Bragg coherent X-ray diffraction imaging (BCDI) has proven to be a powerful technique for investigating the structural deformation of individual nanocrystals during catalytic reactions^{26, 27.}”

Reviewer #4 (Remarks to the Author):

Reviewer #5 (Remarks to the Author):

The manuscript by Park et al. reports on the strain evolution of a TiO₂-supported Au nanoparticle under green, UV, and a combination of green and UV light irradiation conditions, using Bragg Coherent Diffraction X-ray Imaging. While I believe the results provided here are valuable and novel, significant revisions are required. The authors claim to explore structure-activity relationships and provide an understanding of wavelength-dependent structure strain, but the efforts in this regard could be significantly improved. A more extensive analysis, particularly focusing on surface strain in highly strained sites or regions, supported by DFT results, could be employed to better emphasise and spotlight the findings. I recommend resubmitting the paper after addressing these major revisions.

1) I have a few general comments on the figures.

a. I recommend adding tripod axes to figures involving the reconstruction of the NP, whether in 3D views or cross-sections, as not the same views are consistently depicted across the figures.

Answer. We are grateful for the reviewer's attentive comment. For better understanding, we have added the tripod axes to figures involving the reconstruction of the AuNP in the revised manuscript.

b. Additionally, I suggest the authors verify the strain magnitudes indicated in the figures and text. Strain values range from -0.04% to 0.04%, but 0.40% strain is mentioned in the text. The "%" symbol is omitted in some figures, which could lead to misinterpretation, suggesting 4% strain.

Answer. We appreciate the reviewer's invaluable suggestions. We have updated the strain magnitudes indicated in the figures and text in the revised manuscript.

c. Finally, the authors should provide both the voxel size and the spatial resolution achieved in this study.

Answer. We thank the reviewer for the comment regarding the voxel size and the spatial resolution. In response to the reviewer's valuable comment, we have updated the additional explanations about the voxel size and the spatial resolution in the revised manuscript.

[Revised Supplementary Table 2]

Sample	Spatial resolution				
	Voxel size (nm ³)	x direction (nm)	y direction (nm)	z direction (nm)	
AuNP (shown in Fig 2)	Pristine (10 min)	4.40 × 4.40 × 4.40	11.0	8.9	15.6
	Pristine (20 min)	4.40 × 4.40 × 4.40	9.2	8.6	14.3
	Pristine (30 min)	4.40 × 4.40 × 4.40	11.5	9.4	17.4
	Pristine (40 min)	4.40 × 4.40 × 4.40	10.1	8.0	16.4
	Green (10 min)	4.40 × 4.40 × 4.40	9.5	9.7	13.0
	Green (20 min)	4.40 × 4.40 × 4.40	13.3	12.2	16.0
	Green (30 min)	4.40 × 4.40 × 4.40	12.2	10.0	17.3
	UV (10 min)	4.40 × 4.40 × 4.40	9.9	8.0	16.6
	UV (20 min)	4.40 × 4.40 × 4.40	11.3	8.2	17.8
	UV (30 min)	4.40 × 4.40 × 4.40	11.4	9.0	15.6
	green/UV (10 min)	4.40 × 4.40 × 4.40	9.7	8.1	13.3
	green/UV (20 min)	4.40 × 4.40 × 4.40	10.5	9.4	10.9
	green/UV (30 min)	4.40 × 4.40 × 4.40	18.9	13.4	17.4
	Ellipsoidal AuNP (shown in Supplementary Figs. 23-25)	Pristine (10 min)	5.18 × 5.18 × 5.18	11.8	10.2
Pristine (20 min)		5.18 × 5.18 × 5.18	12.6	10.5	17.1
Pristine (30 min)		5.18 × 5.18 × 5.18	11.3	9.3	15.5
Green (10 min)		5.18 × 5.18 × 5.18	12.2	9.9	18.6
Green (20 min)		5.18 × 5.18 × 5.18	12.8	10.0	19.9
Green (30 min)		5.18 × 5.18 × 5.18	14.2	11.8	15.1
UV (10 min)		5.18 × 5.18 × 5.18	11.6	9.5	15.3
UV (20 min)		5.18 × 5.18 × 5.18	11.6	9.9	17.4
UV (30 min)		5.18 × 5.18 × 5.18	13.2	11.5	13.4
green/UV (10 min)		5.18 × 5.18 × 5.18	13.1	10.8	17.5
green/UV (20 min)		5.18 × 5.18 × 5.18	11.4	11.2	14.7
green/UV (30 min)		5.18 × 5.18 × 5.18	13.8	10.8	16.2

Supplementary Table 2. Spatial resolution of AuNPs in the x, y, and z directions for each measurement condition.

[Revised main manuscript, page 19]

“The reconstructed voxel size for the AuNP was determined to be $4.40 \times 4.40 \times 4.40 \text{ nm}^3$ or $5.18 \times 5.18 \times 5.18 \text{ nm}^3$ for each scan. This measurement was derived from the experimental setup (detector pixel size and distance) and the step size of the rocking scan. The spatial resolution was determined by differentiating the electron density amplitude across the object-substrate interface for the X, Y, and Z directions and fitting a Gaussian to each of the profiles. The resolution was approximately 8.0 ~ 19.9 nm, depending on the measurement conditions (Supplementary Table 2).”

2) As a non-expert of the photocatalysis field, I find the measurements of photocatalytic efficiency described, nicely illustrating the synergy gained from the Au/TiO₂ heterostructure. However, the details about the methylene blue (MB) solution are not clear. Specifically, the involvement of O₂ and OH⁻ in the process is mentioned, but details such as pH (and consequently OH⁻ concentration) or whether O₂ was bubbled in the solution are not addressed and could be indicated in the Methods section.

Answer. We thank the reviewer for the comment on the details about the methylene blue (MB) solution. Prior to the photocatalytic reaction, MB aqueous solution (1 mM) was freshly prepared using deionized water (pH 6.4) without O₂ bubbling prior to the photocatalytic reaction. We have conducted a new ESR (Electron Spin Resonance) analysis to accurately determine the types and concentrations of ROS generated under light irradiation (Fig. 2g, h). These spectra show ESR signals of •OH and O₂^{-•} under all illumination conditions while there is no signal in an aqueous solution. Note that •OH and O₂^{-•} exhibit high reactivity and possess short lifetimes on the order of microseconds to nanoseconds⁹. The MB solution contains no radical species. Both radicals show that the ESR signal increases in the order of green, UV, and green/UV, which indicates an increase in the amount of ROS generated relative to the excitation wavelength. The result is consistent with the results of the ROS generation activity experiment using coumarin and DHE in Supplementary Figs. 7 and 8. We have updated the additional results and explanations in the revised manuscript.

[Revised Fig. 2]

Fig. 2. ESR analysis of **g** •OH and **h** O₂^{-•} generation under green, UV, and green/UV irradiation for 1 h. DMPO was used as the spin-trapping agent.

[Revised main manuscript, page 6]

“To verify the type and content of ROS generated, electron spin resonance (ESR) analysis was performed using DMPO as a spin trap agent for •OH and O₂^{-•} (Fig. 2g, h). After 1 h of reaction, the Au/TiO₂ heterostructure exhibited typical ESR signals of DMPO (•OH) and DMPO

($O_2^{\cdot -}$) under all illumination conditions. The intensities of both ESR signals progressively increased in the order of green, UV, and green/UV irradiation, reflecting the same increasing trend for the excitation wavelength. These findings align with those from the ROS generation activity experiments using coumarin and DHE in Supplementary Figs. 7 and 8.”

[Revised main manuscript, page 16]

“MB aqueous solution (1 mM, pH 6.4, 25 °C) was freshly prepared using deionized water without O_2 bubbling prior to the photocatalytic reaction.”

[Reference]

[9] SIES H. Strategies of antioxidant defense. *Eur J Biochem* **215**, 213-219 (1993).

3) I acknowledge the achievement of measuring nanoparticles within a liquid under in situ conditions, considering the relatively small size of the nanoparticles, which makes it challenging for them to remain stable on the support. The authors were able to measure the same nanoparticle using SEM and BCDI. A description of their methodology to achieve this would be appreciated in the Methods section.

Answer. We thank the reviewer for the positive evaluation. We utilized a beamline-mounted optical microscope to localize the nanoparticles analyzed in the BCDI measurements (Supplementary Fig. 27). Subsequently, SEM analysis, guided by the optical images, was performed to precisely target the identical location. This approach enabled the measurement of the same nanoparticles using both BCDI and SEM techniques.

[Revised Supplementary Fig. 27]

Supplementary Fig. 27. **a** Optical images of Au/TiO₂ heterostructures taken from the optical microscope mounted on the beamline. **b** SEM images of the squared region in optical images.

[Revised main manuscript, page 18]

“We utilized a beamline-mounted optical microscope to localize the nanoparticles analyzed in the BCDI measurements (Supplementary Fig. 27). Subsequently, SEM analysis, guided by the optical images, was performed to precisely target the identical location. This approach enabled the measurement of the same nanoparticles using both BCDI and SEM techniques.”

4) In the Results section, the mechanism describing the processes involved in the operation mode of the heterostructure is presented, but the reader is encouraged to refer to Supplementary Figure 5. Figure 1c provides more illustrative support for this mechanism.

Answer. We appreciate the reviewer’s invaluable suggestions. Following the reviewer’s comment, we have repositioned Supplementary Fig. 5 and Fig. 1c to provide more detailed descriptive support for this mechanism.

[Revised Fig. 1]

Fig. 1. Schematic illustration of the in situ photocatalytic BCDI experiment. **a** Au/TiO₂ heterostructure was placed in an aqueous liquid cell and the excitation wavelength was controlled using a xenon lamp. This setup allowed us to systematically irradiate the green (532 nm), UV (365 nm), and green/UV (532 nm and 365 nm) to the Au/TiO₂ heterostructure during

BCDI measurements. The incident focused X-ray beam interacts with the single AuNP inside the reaction liquid cell. Diffraction patterns from the single AuNP were collected at the off-specular (111) Bragg angles. **b** Photograph of the in situ photocatalytic BCDI setup at the 34-ID-C at APS. **c** Schematic illustration of the photocatalytic degradation mechanism of MB by Au/TiO₂ heterostructure under green and UV light irradiation.

5) Only the results obtained along the Au (111) direction are presented, not because it is sensitive to lattice distortions in Au (100) and Au (110), as stated in the text. I recommend removing the sentence that suggests lattice distortion along [100] and [110] directions is accessible through the 111 peak, as this might be confusing. Note that Au(hkl) is not a direction, but [hkl] is.

Answer. Thanks for the reviewer's valuable comment. Based on your comments, we have deleted the sentence and corrected the error in the revised manuscript.

[Revised main manuscript, page 8]

~~"Only the results obtained along the Au (111) direction are presented, because it is also sensitive to lattice distortions in Au (100) and Au (110)."~~

6) I highly appreciate the efforts invested in plotting the diffraction patterns which are often omitted from BCDI papers.

a. I recommend not saturating the colours in the diffraction pattern representation. This might mask changes in the pattern features. As Phase Retrieval is a challenging task and, in some cases, its reliability can be limited, confidence in the reconstructed strain fluctuations can be achieved by examining variations in the diffraction pattern features.

Answer. We deeply appreciate the reviewer for the invaluable suggestion. We agree that the saturation of diffraction patterns can obscure changes in pattern features and decrease the reliability of strain variation measurements. Based on the reviewer's suggestion, we modified the diffraction pattern to avoid saturation in the revised manuscript.

[Revised Supplementary Fig. 14]

Supplementary Fig. 14. Coherent x-ray diffraction measurements of the AuNP at the (111) Bragg peak as a function of x, y, and z coordinates of the scattering vector Q_{111} . The intensity has been normalized by its maximum value. The illumination conditions under which the measurements have been performed are indicated in the figures. As light irradiation continued, leading to the generation of ROS, slight changes were observed in the center of the Bragg peak, and the fringe pattern developed a pronounced asymmetry, whereas the pristine condition exhibited minimal change over time.

b. When looking closely at the diffraction patterns, they seem to have rotated slightly between the different conditions. Is there any explanation for this?

Answer. We appreciate the reviewer for the comment regarding the diffraction patterns. Following the reviewer's comment, we examined whether the diffraction pattern rotated with changes in light conditions. It appeared that there was no significant rotation. However, as light irradiation continued, leading to the generation of ROS, we observed slight changes in the center of the Bragg peak, and the fringe pattern developed a pronounced asymmetry, while the pristine condition exhibited minimal change over time.

[Revised Supplementary Fig. 14]

“**Supplementary Fig. 14.** Coherent x-ray diffraction measurements of the AuNP at the (111) Bragg peak as a function of x, y, and z coordinates of the scattering vector Q_{111} . The intensity has been normalized by its maximum value. The illumination conditions under which the measurements have been performed are indicated in the figures. As light irradiation continued, leading to the generation of ROS, slight changes were observed in the center of the Bragg peak, and the fringe pattern developed a pronounced asymmetry, whereas the pristine condition exhibited minimal change over time.”

7) A reference on the phasing algorithm employed would be appreciated.

Answer. Thanks for the reviewer’s attentive comments. We have added references and metadata to the phasing algorithm as per your suggestion. This detailed information has been added in Supplementary Fig. 15 as described in the revised manuscript.

[Revised Supplementary Fig. 15]

Supplementary Fig. 15. Schematic diagram of phase retrieval algorithm combined with Error reduction (ER) and Hybrid input-output (HIO) using coherent diffraction pattern.

[Revised main manuscript, page 9]

“A phase-retrieval algorithm was employed to determine the 3D electron density and local strain field of a single AuNP (Supplementary Fig. 15).”

[Revised main manuscript, page 19]

“The reconstruction process utilized Error Reduction (ER) and Hybrid Input-Output (HIO) algorithms over a total of 620 iterations^{71, 72}.”

[Revised reference]

[71] Gerchberg RW. A practical algorithm for the determination of plane from image and diffraction pictures. *Optik* **35**, 237-246 (1972).

[72] Fienup JR. Phase retrieval algorithms: a comparison. *Appl Opt* **21**, 2758-2769 (1982).

8) It is mentioned that the pristine AuNP displayed residual surface distortion due to non-equilibrium growth conditions. While equilibrium growth conditions might have led to a more relaxed nanoparticle, the morphology inherently imposes a strain signature, even in the absence of non-equilibrium growth conditions.

Answer. We thank the reviewer for the comment regarding the residual surface distortion of pristine AuNP. The manuscript was revised to reflect the reviewer's comments.

[Revised main manuscript, page 9]

"Additionally, the pristine AuNP exhibited residual distortion of its surface due to the inherent strain signature of its morphology, and these distortions varied with the applied wavelength (Fig. 2)⁵⁷."

9) I suppose the pristine AuNP corresponds to the nanoparticle immersed in the MB solution and not in air. But I did not clearly find this information.

Answer. We are grateful for the reviewer's attentive comment. We conducted a series of chemical analyses to substantiate the results obtained from BCDI measurements. The solution composition for each experiment was prepared for a specific purpose. Detailed descriptions of the experimental methods have been added in the revised manuscript to enhance the clarity and reproducibility of the research.

1) To observe the in situ strain evolution of the single NP by ROS generated during the photocatalytic reactions: The DI water (250 μ L) without O₂ bubbling was dropped onto the sample mounted in a liquid cell for BCDI measurements. Additionally, the sample was measured in air to test the stability.

2) To determine both the type and quantity of ROS generated by the photocatalytic reaction: Coumarin and dihydroethidium (DHE) were employed as detection probes for hydroxyl radicals (\cdot OH) and superoxide radicals (O₂⁻), respectively. These probes were diluted to a concentration of 10 μ M in the deionized water without O₂ bubbling and subsequently introduced to the sample. Furthermore, for ESR analysis, 5,5-dimethyl-1-pyrroline N-oxide (DMPO) at a concentration of 50 mM was introduced into the deionized water and a methanol solution, respectively, without O₂ bubbling. The Au/TiO₂ was then subjected to analysis under

light irradiation.

3) To confirm the interaction between the surface of the NP and ROS: The Au/TiO₂ samples were immersed in the deionized water without O₂ bubbling. The samples were analyzed using FT-IR, XPS, and Raman spectroscopy.

4) To investigate the impact of ROS generated by the NP on the degradation of MB during photocatalytic reactions: A MB aqueous solution (1 mM) was freshly prepared using deionized water without O₂ bubbling prior to the photocatalytic reaction.

[Revised main manuscript, page 16]

“The samples were analyzed using FT-IR (Nicolet iS10, Thermo Fisher Scientific Co., USA), XPS (Theta, Thermo Fisher Scientific Co., Waltham, MA, USA), and Raman spectroscopy (DXR3xi, Thermo Fisher Scientific Co., Waltham, MA, USA).”

[Revised main manuscript, page 16]

“MB aqueous solution (1 mM, pH 6.4, 25 °C) was freshly prepared using deionized water without O₂ bubbling prior to the photocatalytic reaction.”

[Revised main manuscript, page 17]

“To determine both the type and quantity of ROS generated by the photocatalytic reaction: Coumarin and dihydroethidium (DHE) were employed as detection probes for hydroxyl radicals ($\cdot\text{OH}$) and superoxide radicals ($\text{O}_2^{\cdot-}$), respectively. These probes were diluted to a concentration of 10 μM in the deionized water without O₂ bubbling and subsequently introduced to a 24-well plate with a 1 cm x 1 cm film of Au/TiO₂ heterostructure. Furthermore, for ESR analysis, 5,5-dimethyl-1-pyrroline N-oxide (DMPO) at a concentration of 50 mM was introduced into the deionized water and a methanol solution, respectively, without O₂ bubbling.”

[Revised main manuscript, page 17]

“The DI water (250 μL) without O₂ bubbling was dropped onto the sample mounted in a liquid cell for BCDI measurements. Additionally, the sample was measured in air to test the stability.”

10) The general correlation between the decrease in integrated intensity (I) and the distortion at the reconstruction surface is hazardous.

a. Changes in integrated intensity can originate from various factors (beam, optics...) and may be difficult to interpret. They can indeed result from a reduction in crystallinity. Importantly, as I decreases, the reconstruction quality also decreases, directly correlating with the left shift of the peak of the reconstructed amplitude histogram, as illustrated in Fig. 3a.

Answer. We thank the reviewer for the comment regarding the interpretation based on the intensity

decrease. As the reviewer mentioned, the intensity decrease can be caused by various factors as well as by a reduction in crystallinity. In this regard, we confirmed the particle stability by repeatedly measuring the rocking curve of the single AuNP in air, indicating the NP was well fixed to the substrate (Supplementary Fig. 16). Furthermore, as shown in Supplementary Fig. 17, the amplitude histogram and the particle shape in the water environment remained nearly identical to those observed in air, confirming that the nanoparticle is stable during BCDI measurements. In the case of our research, we confirmed that the adsorption of ROS on Au atoms near the surface of AuNP forms Au-O species and introduces a partial phase transformation, thus decreasing the Bragg electron density at the transformed region (Supplementary Fig. 18). We believe that this is the primary reason for the intensity decrease as well as the left shift of the peak of the reconstructed amplitude histogram.

b. Plus, as the reconstruction quality decreases, the strain values are more likely to be large (positive or negative). In addition, given that the reconstruction is consistently observed at an isosurface of 30%, it is likely that equivalent surfaces are not always considered between reconstructions, making comparisons of surface strains challenging.

Answer. We thank the reviewer for the comment regarding the determination of an isosurface to visualize a 3D image. We understand what the reviewer is concerning. Determining an isosurface level based on a reconstructed amplitude histogram could be one measure¹⁰ [see Fig. 11d in the reference]. Several researchers have analyzed data from specific isosurfaces to observe the relative changes in strain under different conditions^{11, 12, 13} (i.e., an isosurface of 23% in *Nat Commun* **12**, 5385 (2021), an isosurface of 25% in *Nano Lett* **19**, 5044-5052 (2019), an isosurface of 65% in *Nat Mater* **22**, 754-761 (2023)). In our observation, we confirmed structural deformation of an AuNP as a function of light illumination. The adsorption of ROS to Au atoms causes phase transformation, attributed to the formation of Au-O species, resulting in a decrease in Bragg electron density. The histogram of reconstructed amplitude also shows shifts to the left side. Because of this reason, we selected an isosurface of 30%, a value that can facilitate the assessment of relative changes across all cases (Additional Resource. 7), to visualize a 3D image in order not to lose information at the transformed region.

[Additional Resource. 7]

Additional Resource. 7. Amplitude histogram of a single AuNP from Fig. 2 and criterion for isosurface level determination.

[Reference]

[10] Carnis J, *et al.* Towards a quantitative determination of strain in Bragg Coherent X-ray Diffraction Imaging: artefacts and sign convention in reconstructions. *Sci Rep* **9**, 17357 (2019).

[11] Carnis J, *et al.* Twin boundary migration in an individual platinum nanocrystal during catalytic CO oxidation. *Nat Commun* **12**, 5385 (2021).

[12] Kim D, *et al.* Defect Dynamics at a Single Pt Nanoparticle during Catalytic Oxidation. *Nano Lett* **19**, 5044-5052 (2019).

[13] Atlan C, *et al.* Imaging the strain evolution of a platinum nanoparticle under electrochemical control. *Nat Mater* **22**, 754-761 (2023).

11) I am sceptical about drawing conclusions from the histogram of the normalised reconstructed amplitudes. Left shifts in the histogram and changes in histogram shape indicate a reduction in the quality of reconstruction, which are likely due to the decrease in integrated intensity. Moreover, it is usual to display the entire range [0; 1] of normalised amplitudes. These histograms should ideally serve to determine a relevant isosurface for each reconstruction.

Answer. We thank the reviewer for the comment regarding the amplitude histogram. We understand what the reviewer is concerning. Please refer to the answer to question 10b. Additionally, to display the entire range of normalized amplitudes from 0 to 1, we have updated the amplitude histogram in the revised manuscript.

[Revised Fig. 3]

Fig. 3. Statistical distribution of the amplitude and strain of a single AuNP on TiO_2 film from Fig. 2. The amplitude distributions between 0.0 and 1.0 are plotted in (a) with corresponding (b) bulk strain distributions. The lines on the strain plots indicate the threshold values (-0.2% and 0.2%).

12) I would encourage the authors to provide a methodology on the determination of the bulk and surface voxels.

Answer. We thank the reviewer for the valuable comment. In response to the reviewer's valuable comment, we have added the methodology on the determination of the bulk and surface voxels in the revised manuscript.

[Revised main manuscript, page 19]

“Surface voxels are defined as those not completely surrounded by other voxels, while bulk voxels are those entirely enclosed by other voxels. Due to the voxel size, surface voxels hold average information not only from within the outermost atomic layer but also approximately from the 20 outermost atomic layers.”

13) Regardless of the strain region (bulk or surface), Supplementary Table 1 indicates that integrated strain over all tensile regions consistently exceeds integrated strain over all compressed regions.

a. This appears contradictory, as the strain discussed in this paper is more precisely the heterogeneous strain — corresponding to lattice fluctuations around the average d-spacing of the scanned NP under a given condition. Consequently, this strain is always zero when averaged over all voxels. Thus, tensile strain cannot surpass compressive strain for both bulk and surface regions. The possibility of more tensile strain indicating an unremoved (positive) phase ramp in post-processing after phasing is acknowledged.

Answer. We thank the reviewer for the comment on the strain area. As in reviewer's comment # 13b, we think it is more appropriate to obtain the full width at half maximum of the bulk and surface strain histograms than Supplementary Table 1. Therefore, Supplementary Table 1 was replaced with FWHM values in the revised manuscript.

b. In my view, the surface strain histogram asymmetry or full width at half maximum are more compelling, illustrating that specific surface sites are more strained than others. Additionally, the quality of reconstruction and (iso)surface determination strongly influences the histograms. Therefore, I recommend attaching more importance to strain distribution when reconstructions are considered sufficiently accurate.

Answer. We appreciate the reviewer's insightful comments regarding the full width at half maximum (FWHM) of the strain histograms. As the reviewer's comment, we calculated the FWHM value of the surface and bulk strain histograms depending on the irradiation conditions (Supplementary Table 1). Under pristine conditions, the FWHM values of the surface and bulk strain distributions show similar values, with an average of 0.018% and 0.017%, respectively. However, during the photocatalytic reactions, the average FWHM value of the surface strain distribution showed an increase in the sequence of green (0.022%), UV (0.023%), and green/UV (0.026%) light irradiation. This trend indicates that the surface strain intensified in proportion to the generated ROS. In addition, the average FWHM value of the bulk strain distribution under photoreaction conditions also increased compared to the value of pristine conditions but showed almost the same value (0.021%) regardless of the wavelength. This indicates that the strain originates at the AuNP surface and is accommodated across the entire bulk region of the AuNP. We have updated the additional results and explanations about the FWHM of the strain distribution in the revised manuscript.

[Revised Supplementary Table 1]

FWHM (%)	Pristine		Green		UV		Green/UV	
	Surface	Bulk	Surface	Bulk	Surface	Bulk	Surface	Bulk
10 min	0.018	0.015	0.023	0.021	0.021	0.020	0.026	0.022
20 min	0.017	0.016	0.022	0.022	0.025	0.022	0.028	0.023
30 min	0.019	0.019	0.023	0.020	0.024	0.021	0.025	0.021
Average	0.018	0.017	0.022	0.021	0.023	0.021	0.026	0.022

Supplementary Table 1. Full width at half maximum value of surface versus bulk strain distributions as a function of the irradiated wavelength.

[Revised main manuscript, page 11]

“Supplementary Table 1 presents the full width at half maximum (FWHM) values of surface versus bulk strain distributions as a function of the irradiated wavelength. Under pristine conditions, the FWHM values of surface and bulk strain distributions show similar values, with an average of 0.018% and 0.017%, respectively. However, during the photocatalytic reactions, the average FWHM value of the surface strain distribution showed an increase in the sequence of green (0.022%), UV (0.023%), and green/UV (0.026%) light irradiation. This trend indicates that the surface strain intensified in proportion to the generated ROS. In addition, the average FWHM value of the bulk strain distribution under photoreaction conditions also increased compared to the value of pristine conditions but showed almost the same value (0.021%) regardless of the wavelength. This suggests that the strain originates at the AuNP surface and is accommodated across the entire bulk region of the AuNP.”

14) Similar trends in surface vs bulk histograms are interpreted as indicative of a poor accommodation of the bulk region. I would rather propose the opposite perspective, but I might be overly concerned about this point, and respect the author's interpretation. My intention is to ensure they accurately conveyed their intended meaning.

Answer. We deeply appreciate the reviewer for the invaluable suggestion, which improves the quality of our manuscript. We completely understand and agree with your opinion. This part has been revised in the manuscript as follows.

[Revised main manuscript, page 12]

“This suggests that the strain originates at the AuNP surface and is accommodated across the entire bulk region of the AuNP.”

15) I discourage the use of displacement analysis in this context, unless specifically addressing defects, which is not the case here.

a. In a BCDI experiment, displacement always involves an offset and cannot directly account for structural changes. Instead, focusing on how fast displacement changes across the NP (displacement variations) informs on structural changes, which aligns with strain and eliminates the offset issue. Additionally, the drawn cross sections in Fig. 4b is perpendicular to the [111] direction. This implies that the displacements shown correspond to atomic position variations out of the section plane, making interpretation challenging.

Answer. We thank the reviewer for the comment regarding the displacement variations of AuNP. We agree that displacement is subject to offset issues, complicating its direct application in elucidating structural changes. As recommended by the reviewer, we investigated the structural change induced by ROS by monitoring the variation in the average lattice spacing ($\Delta\bar{d}$) of the entire nanoparticle under varying light irradiation conditions. As shown in Fig. 4b, the average lattice spacing increased sequentially from green (0.0017 Å to 0.0019 Å) to UV (0.0027 Å to 0.0032 Å) to green/UV (0.0038 Å to 0.0043 Å) under different light irradiation conditions, with the increase being proportional to the generation of ROS. Furthermore, the observed average displacement variation, which ranged from 0.18% to 0.41%, exhibited a similar increasing trend as the strain changes associated with different light irradiation conditions, thus confirming that the structural changes in the AuNPs are induced by the generation of ROS.

b. Fig. 4c and 4d hold more meaningful information in this regard. I would also ask the authors to add the plane position since the dashed line box does not indicate where the section is taken.

Answer. We thank the reviewer for the valuable comment. According to the reviewer's comment, we have updated the plane position of the AuNP in the revised manuscript.

[Revised Fig. 4]

Fig. 4. Average strain field of a single AuNP. **a** 3D reconstructed images at a 30% amplitude threshold with lattice displacement along the [111] direction projected on the isosurfaces. The gray boundary depicts the pristine cross-section in (a). **b** Variation in the average lattice spacing ($\Delta\bar{d}$) under green, UV, and green/UV irradiation for 30 min. The change in $\Delta\bar{d}$ is calculated using the formula $\Delta\bar{d} = \bar{d}_{111} - \bar{d}_{111}^*$, where the \bar{d}_{111} is the average lattice spacing measured across the AuNP and \bar{d}_{111}^* is the lattice spacing of the first measured pristine AuNP. **c** Cross-sectional views of the internal strain field at the dashed line box in Fig. 4a. The gray boundary illustrates the shape of the AuNP in pristine condition at 10 min. **d** Average strain field for the internal plane in Fig. 4c under green, UV, and green/UV irradiation for 30 min.

[Revised main manuscript, page 12]

“To investigate the propagation of surface changes inward to the bulk region of the AuNP, we calculated the absolute variation in average lattice spacing ($\Delta\bar{d} = \bar{d}_{111} - \bar{d}_{111}^*$), where \bar{d}_{111}^* represents to the initially measured pristine AuNP (Fig. 4a, b)^{30,61}. Under the pristine condition, no significant changes in $\Delta\bar{d}$ were observed during the BCDI measurement. However, subsequent to various light irradiations, $\Delta\bar{d}$ sequentially increased in the order of green (0.0017 Å ~ 0.0019 Å), UV (0.0027 Å ~ 0.0032 Å), and green/UV (0.0038 Å ~ 0.0043 Å). This increase in lattice spacing $\Delta\bar{d}$ correlates with the generation of ROS, which is attributed to ROS adsorption on the Au surface.”

16-a) I found Supplementary Figure 12c and d challenging to understand, particularly with the mention of "in-plane" distortion of the top slab layer while coloured arrows indicate the out-of-plane directions. Authors could clarify this.

Answer. We thank the reviewer for the comment regarding the in-plane distortion of the top slab layer. To enhance understanding of the displacement directions of Au atoms, tripod axes have been incorporated for comparison. As shown in Supplementary Fig. 26d-f, the top nine Au atoms undergo movement for structural stabilization upon adsorption of OH, O₂, and H₂O molecules. Specifically, the Au atoms with adsorbed OH and O₂ migrate upward along the +z axis, as illustrated by the red arrows, moving away from the center of mass. In contrast, the remaining Au atoms that do not have adsorbates move downward towards the surface, along the -z axis. It is noteworthy that the Au atoms with OH and O₂ adsorptions expand by 29.4 times and 13.7 times more along the z-axis compared to those with H₂O. This substantial displacement, driven by the adsorption of ROS during photocatalytic reactions, contributes to the deformation of the AuNPs. We have updated the DFT results about the displacement of the top-layer Au atoms in the revised manuscript.

[Revised Supplementary Fig. 26]

Supplementary Fig. 26. Adsorption configurations of OH, O₂, and H₂O on AuNP and their

associated displacement response. **a-c** Optimized adsorption configurations of OH, O₂, and H₂O on the Au (111) surface: (left) top view and (right) side view of the most stable structure. **d-f** In-plane distortion of the top-layer Au atoms for the adsorption of OH, O₂, and H₂O. The in-plane distortion of the top-layer Au surface determined from the top 9 Au atoms. The red and blue colors represent the out-of-plane directions along the z-axis, with red indicating upward movement in the +z direction away from the center of mass, and blue indicating downward movement in the -z direction toward the surface.

16-b) Employing DFT calculations is a highly valuable effort to rationalise strain observations under different reaction conditions. However, analysis could be enhanced by calculating the out-of-plane strain in the simulated slab to compare it with the strain captured experimentally on the top facet (also out-of-plane). Crystallographic axes in the figure would further improve clarity.

Answer. We thank the reviewer for the insightful comment. Additional analysis on the strain has been addressed in response 18-c. In accordance with the reviewer's suggestion, we have added crystallographic axes to the figure to improve clarity.

17) While the Results section clearly stipulates that O₂^{•-} and OH[•] should be generated at the AuNP surface under UV and green irradiation, respectively, the Discussion section introduces the idea of OH[•] formation at the AuNP surface under UV irradiation. However, I did not find any explanation for this point in the text. Given that OH[•] is said to cause a "higher tensile strain [on the NP] compared to that generated under green-light irradiation," I would appreciate further clarification on this part.

Answer. We thank the reviewer for the comment regarding the •OH formation at the AuNP surface under UV irradiation. We apologize for any confusion about the ROS generation mechanism. The Au/TiO₂ heterostructure in this research varies in the location and type of ROS generation depending on UV and green light irradiation (Additional Resource. 3).

Under UV irradiation, electrons in the VB of TiO₂ can be excited to the CB, resulting in holes in the VB. Because the Fermi level of the Au is lower than that of the CB of TiO₂, the photoexcited electrons in the CB can migrate to the Au. ① This electron transfer generates O₂^{•-} on the Au surface through O₂ reduction, whereas ② the holes in TiO₂ react with OH⁻ to generate •OH.

Under green irradiation, photoinduced electron-hole pairs are generated through the plasmonic resonance effect of the AuNP. Because photoexcited electrons have a very short lifetime in the order of picoseconds, the electron-hole pairs on the AuNP cannot participate in the photocatalytic reaction. However, as AuNPs are attached to the TiO₂ surface, the photoexcited electrons from the AuNPs can effectively migrate to the CB of TiO₂, inhibiting the

relaxation and recombination of electrons in the Au/TiO₂ heterostructure. ③ This electron-transfer process facilitates the accumulation of electrons on the TiO₂ surface and promotes the reduction of O₂ to O₂^{•-}. Furthermore, ④ the remaining holes in the AuNPs can react with OH⁻ to produce •OH.

During photocatalytic reactions, the generated O₂^{•-} radicals spontaneously react with H₂O to form additional •OH⁶. This means that both O₂^{•-} and •OH generated during photocatalytic reactions can directly and indirectly participate in the structural deformation of the AuNP. As shown in Fig. 2g, h, and Supplementary Figs. 6-9, UV irradiation produces a greater quantity of ROS than green irradiation. Results from various spectroscopy analyses also indicate that, under UV, a greater quantity of Au-O species forms on the Au surface compared to under green irradiation (Fig. 2i, j, and Supplementary Fig. 12). These results support that greater tensile strain can be generated in the AuNP. To avoid misleading, we have updated the revised schematic illustration and the ROS generation mechanism of Au/TiO₂ heterostructure in the revised manuscript.

[Additional Resource. 3]

Additional Resource. 3. Schematic illustration of the photocatalytic degradation mechanism of MB by Au/TiO₂ heterostructure.

[Revised main manuscript, page 7]

“Under combined green/UV irradiation, these mechanisms operate simultaneously, leading to efficient charge transfer and resulting in the highest ROS generation activities. As shown in Supplementary Fig. 11, the plasmonic resonance effect induced by green irradiation generates electrons in AuNPs, which are then transferred to the CB of TiO₂. At the same time, UV irradiation generates electron-hole pairs in TiO₂. The excited electrons of TiO₂ accumulated in the CB of TiO₂ with the electrons transferred from AuNP, which could promote the generation of O₂^{•-} in the CB of TiO₂. Additionally, the holes in the VB of both the AuNP and TiO₂ could promote the generation of •OH. This charge transfer mechanism demonstrates that the

Au/TiO₂ heterostructure is capable of optimal ROS generation through its efficient wavelength absorption ability under green/UV irradiation.”

[Revised main manuscript, page 14]

“These holes accumulate near the Au surface, serving as additional active sites for the generation of •OH. The resultant ROS induces approximately 0.2% tensile strain in the Au lattice^{66, 67}. However, under UV irradiation, the AuNP acts as an electron sink, and these accumulated electrons on the surface trigger the generation of O₂^{•-} and •OH species^{33, 43, 45}. During the photocatalytic reaction, the interaction between the Au surface and O₂^{•-} and •OH species leads to the formation of Au-O species on the Au surface. Considering that UV irradiation produces more ROS than green irradiation (Fig. 2f), it results in a larger tensile strain (approximately 0.3%) in the AuNP. Moreover, under green/UV irradiation, the Au/TiO₂ heterostructure efficiently absorbs both green and UV wavelengths, maximizing ROS generation through enhanced charge transfer, which is reflected in the increased formation of Au-O on the Au surface^{45, 68}.”

[Reference]

[6] Hirakawa T, Yawata K, Nosaka Y. Photocatalytic reactivity for O₂^{•-} and OH radical formation in anatase and rutile TiO₂ suspension as the effect of H₂O₂ addition. *Applied Catalysis A: General* **325**, 105-111 (2007).

18-a) I am highly concerned about the interpretations presented in the Discussion section. The assertion that the Au surface becomes negatively charged, particularly given that green-light irradiation generates a hole at the surface, is unclear to me.

Answer. We thank the reviewer for the comment on the charge transfer pathway of AuNP. We apologize for any confusion about the charge transfer mechanism under green/UV irradiation. According to the literature, X-ray absorption spectroscopy shows that the absorption intensity of Ti increases differently with green and green/UV irradiation on Au/TiO₂ particles⁷. In their study, as depicted in Figure 7a of the paper, the researchers used green light with a wavelength of 545 nm as the visible light illumination, similar to the 532 nm wavelength we utilized in our study. Note that the electrons are not generated in TiO₂ under green irradiation due to the wide band gap of TiO₂. The X-ray absorption of Ti under green irradiation indicates that the electrons from only Au transfer to TiO₂. Additionally, green/UV irradiation shows a greater absorption intensity compared to green irradiation, indicating that a greater amount of electrons are accumulated in the CB of TiO₂ under green/UV irradiation. Therefore, efficient charge transfer through green/UV irradiation indicates an important cause for increasing the amount of ROS. Supplementary Fig. 11 in the revised manuscript shows the photoinduced charge transfer and ROS generation process in detail under green/UV irradiation. First, this figure shows the

process of generating electrons from the AuNP through the plasmonic resonance effect caused by green irradiation and transferring these electrons to the CB of TiO₂. At the same time, UV irradiation generated electron-hole pairs. The excited electrons of TiO₂ accumulated in the CB of TiO₂ with the electrons transferred from AuNP at the CB of TiO₂, which could promote the generation of O₂^{•-} in the CB of TiO₂. Additionally, the holes in the VB of both the AuNP and TiO₂ could promote the generation of •OH. We have updated the schematic illustration of the proposed photocatalytic mechanism of Au/TiO₂ heterostructure under green/UV irradiation and further explanations in the revised manuscript.

[Revised Supplementary Fig. 11]

Supplementary Fig. 11. Schematic illustration of the proposed photocatalytic mechanism of Au/TiO₂ heterostructure under green/UV irradiation.

[Revised main manuscript, page 7]

“Under combined green/UV irradiation, these mechanisms operate simultaneously, leading to efficient charge transfer and resulting in the highest ROS generation activities. As shown in Supplementary Fig. 11, the plasmonic resonance effect induced by green irradiation generates electrons in AuNPs, which are then transferred to the CB of TiO₂. At the same time, UV irradiation generates electron-hole pairs in TiO₂. The excited electrons of TiO₂ accumulated in the CB of TiO₂ with the electrons transferred from AuNP, which could promote the generation of O₂^{•-} in the CB of TiO₂. Additionally, the holes in the VB of both the AuNP and TiO₂ could promote the generation of •OH. This charge transfer mechanism demonstrates that the Au/TiO₂ heterostructure is capable of optimal ROS generation through its efficient wavelength absorption ability under green/UV irradiation.”

[Revised main manuscript, page 14]

“These holes accumulate near the Au surface, serving as additional active sites for the

generation of •OH. The resultant ROS induces approximately 0.2% tensile strain in the Au lattice^{66, 67}. However, under UV irradiation, the AuNP acts as an electron sink, and these accumulated electrons on the surface trigger the generation of O₂^{•-} and •OH species^{33, 43, 45}. During the photocatalytic reaction, the interaction between the Au surface and O₂^{•-} and •OH species leads to the formation of Au-O species on the Au surface. Considering that UV irradiation produces more ROS than green irradiation (Fig. 2f), it results in a larger tensile strain (approximately 0.3%) in the AuNP. Moreover, under green/UV irradiation, the Au/TiO₂ heterostructure efficiently absorbs both green and UV wavelengths, maximizing ROS generation through enhanced charge transfer, which is reflected in the increased formation of Au-O on the Au surface^{45, 68}. Consequently, the increased availability of electrons and holes under combined irradiation conditions provides more active sites for the generation of ROS compared to those available under either green or UV irradiation alone, leading to a significant dilation (approximately 0.4%) of the lattice due to the increased production of ROS. This distortion is likely the cause of tensile strain in AuNPs that affects the d-band, as well as the adsorption and dissociation energies^{69, 70}.”

[Reference]

[7] Yang K-S, *et al.* Plasmon-Induced Visible-Light Photocatalytic Activity of Au Nanoparticle-Decorated Hollow Mesoporous TiO₂: A View by X-ray Spectroscopy. *J Phys Chem C* **122**, 6955-6962 (2018).

18-b) Assuming this is valid, the logical connection between the negative charge and the dilation of the Au lattice at the surface is not clearly established. Moreover, the subsequent interpretation appears to be reversed and biased. The authors suggest that the negatively charged surface dilates the Au surface lattice, shifting the d-band centre and subsequently increasing the adsorption of ROS species at the Au surface. Yet, no analytical evidence is provided to confirm this interpretation. The observed strain is more likely the result of adsorption phenomena, as demonstrated by numerous BCDI-catalysis papers, rather than from a negatively charged surface that would enhance ROS species adsorption.

Answer. We agree that our explanation of the strain evolution process of AuNP was not clear in the previously submitted manuscript. As noted by the reviewer, our findings in this research suggest that the adsorption of ROS, a product of the photocatalytic reaction, induces strain on the surface of the AuNP. To confirm the adsorption of ROS on the Au surface, we conducted spectroscopic experiments including XPS, FT-IR, and Raman spectroscopy during photocatalytic reaction (Fig. 2i, j and Supplementary Fig. 12). The results show the generation of new Au-O species on the Au surface and the amount of ROS generated is proportional to the amount of adsorption on the Au surface.

On the other hand, we think the sentence on page 12, line 16 of the original manuscript could confuse readers. The photo-induced electron-hole pair and the negatively charged surface can

efficiently generate ROS, which induces the strain. Therefore, the negatively charged surface does not directly influence the strain in nanoparticles. We have revised the sentence for clarity in the revised manuscript.

[Revised main manuscript, page 14]

“Moreover, under green/UV irradiation, the Au/TiO₂ heterostructure efficiently absorbs both green and UV wavelengths, maximizing ROS generation through enhanced charge transfer, which is reflected in the increased formation of Au-O on the Au surface^{45, 68}. Consequently, the increased availability of electrons and holes under combined irradiation conditions provides more active sites for the generation of ROS compared to those available under either green or UV irradiation alone, leading to a significant dilation (approximately 0.4%) of the lattice due to the increased production of ROS. This distortion is likely the cause of tensile strain in AuNPs that affects the d-band, as well as the adsorption and dissociation energies^{69, 70}.”

[Revised Fig. 2]

Fig. 2. Characterization of a single AuNP deposited on TiO₂ film. SEM images of (a) top-view and (b) cross-view of the Au/TiO₂ heterostructure. (c) EDX mappings of Au, Ti, and O. Scale bar: 50 nm. Photocatalytic degradation of MB on (d) AuNP, (e) TiO₂, and (f) Au/TiO₂ heterostructure under green, UV, and green/UV irradiation for 60 min. ESR analysis of (g) •OH and (h) O₂^{•-} generation under green, UV, and green/UV irradiation after 1 h. DMPO was used as the spin-trapping agent. High-resolution XPS spectra of (i) Au 4f and (j) O 1s. (k) 3D reconstructed images of AuNP at a 30% amplitude threshold. The gray boundary illustrates the shape of the AuNP in the pristine condition at 10 min. (l) 3D strain images of AuNP associated with compressive (blue) and tensile (red) strains. The lattice strain along the [111] direction was projected onto the isosurfaces.

[Revised Supplementary Fig. 12]

Supplementary Fig. 12. **a** Raman spectra of Au-O for the Au/TiO₂ after green, UV, and green/UV irradiation, respectively. FT-IR spectra of **b** OH and **c** OOH bond for the Au/TiO₂ under green, UV, and green/UV irradiation, respectively.

18-c) In my opinion, an in-depth analysis of the strain at the surface of the nanoparticle should be provided, linked explicitly to adsorption phenomena using DFT simulations as robust support. This would explicitly demonstrate that the TiO₂ supported NP is particularly active under UV/green-light irradiation.

Answer. We thank the reviewer for the insightful comment regarding the strain analysis of nanoparticle surfaces. We agree that DFT calculations are crucial for a comprehensive understanding of adsorption phenomena along crystal planes and the associated strain changes on nanoparticle surfaces. Indeed, recent studies have reported in-depth analyses of strain changes in Pt nanoparticles by combining robust DFT calculations with BCDI analysis¹³. However, in our revision, we faced challenges in supporting the strain evolution through additional DFT calculations. This difficulty could arise due to the short lifetime of ROS generated under different wavelengths of illumination, the presence of intermediates, and the presence of multiple types of ROS. It is challenging to simultaneously consider the types of ROS and correlate the strain changes on the Au surface, as measured by BCDI, with DFT calculations for the entire nanoparticle. We believe it is necessary to address this issue in future research through collaborations focusing on DFT calculations. Despite this limitation, we are confident that our study offers significant contributions to the fields of photocatalysis and BCDI analysis. Specifically, we have established a BCDI-based photocatalyst system for the first time and have observed the structure-activity relationship. Additionally, through this revision, we have identified the types of ROS generated during the photocatalytic reaction and provided direct experimental evidence that these ROS form new Au-O bonds on the nanoparticle surface.

1) Electron spin resonance (ESR) analysis of $\bullet\text{OH}$ and $\text{O}_2\text{-}\bullet$ species to accurately determine the types and content of ROS (Fig. 2g, h).

2) XPS and Raman spectroscopy measurements to distinguish the Au-O species formed on the Au surface (Fig. 2i, j and Supplementary Fig. 12a).

3) FT-IR analysis to detect the surface-adsorbed species on the Au surface (Supplementary Fig. 12b, c).

These experimental findings strongly support the in situ BCDI measurement results obtained during the photocatalytic reaction.

[Reference]

[13] Atlan C, *et al.* Imaging the strain evolution of a platinum nanoparticle under electrochemical control. *Nat Mater* **22**, 754-761 (2023).

REVIEWER COMMENTS

Reviewer #1 (Remarks to the Author):

Most of the comments proposed have been responded. I recommend its publication.

Reviewer #2 (Remarks to the Author):

The answers from the authors are fine for me.

Reviewer #3 (Remarks to the Author):

I read both the paper and the reviewer's comments and authors rebuttal.

I think the authors did a very good job answering the questions and comments of the two reviewers and with these explanations and changes in the paper the paper is now acceptable for publication.

Reviewer #4 (Remarks to the Author):

Reviewer #5 (Remarks to the Author):

The revised version of the paper titled In situ photocatalytic strain evolution of a single Au nanoparticle in Au/TiO₂ heterostructures has significantly improved in quality by addressing most of the previous comments. The additional investigations on beam-induced radiolysis, and the differentiation of changes in air versus water, are highly appreciated. Moreover, providing additional analysis on a second particle enhances the reliability of their findings. The article is nearly ready for publication, but two major points still need to be addressed:

1. In L. 336 (This distortion is likely the cause of tensile strain in AuNPs that affects the d-band, as well as the adsorption and dissociation energies): the distortion does not cause tensile strain; rather, the strain is a measure of the distortion. Also, I am not fully convinced of the necessity of establishing a relationship between strain and d-band centre/adsorption energy in this context, considering specifically the AuNP under reaction conditions. The tensile strain is induced from the adsorption and according to the authors' reasoning, this would imply that adsorption affects the d-band, the dissociation energies and the adsorption itself... Is this the intended meaning?

2. I am slightly confused about the conclusions drawn at the end of the discussion. Irreversible morphological changes are mentioned, but not clearly evidenced. Shall we infer from Supplementary Fig 19 that morphology cannot be recovered after the catalytic reaction, even if the initial intensity I_0 is restored? What about the other structural quantities such as strain/dspacing? Does the AuNP still exhibit large positive values after the reaction? If so, the relationship with the d-band theory and the reduction in photocatalytic performance would be more comprehensible in this context (i.e. after not during the reaction). However, this must be balanced with the statements made in L 239–242 (Terminating light irradiation, which ceases the generation of ROS and reduces adsorption to the AuNP surface, reduces lattice distortions, potentially eliminating voids and allowing the AuNPs to return to a state similar to their initial condition (Supplementary Fig. 19)) which, to me, seem to indicate the opposite conclusion.

Finally, here are some minor concerns and suggestions the authors can also address :

1. While discussing the absolute d-spacing evolution is a good point, I wonder whether the d-spacing returns to its pristine value. If it does not, could this indicate that some adsorbates remain bonded to the surface upon returning to water? (see comment below)

2. P 12: Fig. 4c shows the corresponding strain field of a single AuNP under different irradiation conditions. The term "corresponding" is no longer relevant, given the edits made just above in the text.

3. As I mentioned in my previous round of comments, including strain instead of displacement in the DFT results would provide clear evidence that the strain response follows the same trends as observed experimentally. In comment 16-b, I did not ask for new DFT simulations, but merely for strain calculations using the already relaxed slabs. Alternatively, the authors could argue for the use of displacement analysis by emphasising that the "bulk" atoms in the DFT slabs were fixed to the gold lattice parameter during the DFT simulation. Thus, a qualitative interpretation could extend the displacement analysis to that of strain.

REVIEWER COMMENTS_NCOMMS-23-57475A

Reviewer #5:

The revised version of the paper titled In situ photocatalytic strain evolution of a single Au nanoparticle in Au/TiO₂ heterostructures has significantly improved in quality by addressing most of the previous comments. The additional investigations on beam-induced radiolysis, and the differentiation of changes in air versus water, are highly appreciated. Moreover, providing additional analysis on a second particle enhances the reliability of their findings. The article is nearly ready for publication, but two major points still need to be addressed:

1-a. In L. 336 (This distortion is likely the cause of tensile strain in AuNPs that affects the d-band, as well as the adsorption and dissociation energies): the distortion does not cause tensile strain; rather, the strain is a measure of the distortion. Also, I am not fully convinced of the necessity of establishing a relationship between strain and d-band centre/adsorption energy in this context, considering specifically the AuNP under reaction conditions. The tensile strain is induced from the adsorption and according to the authors' reasoning, this would imply that adsorption affects the d-band, the dissociation energies and the adsorption itself... Is this the intended meaning?

Answer. We thank the reviewer for a thorough and logical review of the manuscript. We have removed the sentence that could cause confusion.

2. I am slightly confused about the conclusions drawn at the end of the discussion. Irreversible morphological changes are mentioned, but not clearly evidenced. Shall we infer from Supplementary Fig 19 that morphology cannot be recovered after the catalytic reaction, even if the initial intensity I_0 is restored? What about the other structural quantities such as strain/d-spacing? Does the AuNP still exhibit large positive values after the reaction? If so, the relationship with the d-band theory and the reduction in photocatalytic performance would be more comprehensible in this context (i.e. after not during the reaction). However, this must be balanced with the statements made in L 239–242 (Terminating light irradiation, which ceases the generation of ROS and reduces adsorption to the AuNP surface, reduces lattice distortions, potentially eliminating voids and allowing the AuNPs to return to a state similar to their initial condition (Supplementary Fig. 19)) which, to me, seem to indicate the opposite conclusion.

Answer. We appreciate the reviewer's comments regarding the irreversible changes. We acknowledge and understand the concerns raised. As mentioned by the reviewer, we have calculated the strain and d-spacing changes in the AuNPs as shown in Supplementary Figure 19. After terminating light irradiation, the fact that they do not fully revert to the original state implies irreversible structural changes (Supplementary Fig. 19 and Supplementary Table 1). Accumulative changes during repeated and long-term photocatalytic reactions could lead to significant irreversible structural changes. We have updated the additional results in the revised manuscript.

[Revised Supplementary Table 1]

	Water	Green	Water	
	40min	40min	10min	20min
(a) Average Strain (%)	0.05	0.23	0.12	0.11
(b) $\Delta\bar{d}$ ($\times 10^{-3}\text{\AA}$)	0.01	1.92	0.15	0.18

Supplementary Table 1. a Average strain field of the AuNP in Supplementary Fig 19. **b** Variation in the average lattice spacing under water and green condition. The change in $\Delta\bar{d}$ is calculated using the formula $\Delta\bar{d} = \bar{d}_{111} - \bar{d}_{111}^*$, where the \bar{d}_{111} is the average lattice spacing measured across the AuNP and \bar{d}_{111}^* is the lattice spacing of the first measured pristine AuNP.

[Revised main manuscript, page 10]

“Terminating light irradiation ceases the generation of ROS and reduces adsorption to the AuNP surface, which reduces lattice distortions. This could potentially reduce voids, allowing the AuNPs to approach a state similar to their initial condition^{50, 60}. However, the fact that they do not fully revert to their original state implies irreversible structural changes (Supplementary Fig. 19 and Supplementary Table 1). Accumulative changes during repeated and long-term photocatalytic reactions could lead to significant irreversible structural changes.”

[Revised main manuscript, page 14]

“Even after the photocatalytic reaction, the reconstructed AuNPs did not completely revert to their initial state. These findings suggest that irreversible structural deformations at the single-particle level can contribute to macroscopic irreversibility including reductions in photocatalytic performance and catalyst stability, particularly during repeated and long-term reactions at a bulk scale.”

Finally, here are some minor concerns and suggestions the authors can also address :

1. While discussing the absolute d-spacing evolution is a good point, I wonder whether the d-spacing returns to its pristine value. If it does not, could this indicate that some adsorbates remain bonded to the surface upon returning to water? (see comment below)

Answer. We thank the reviewer for the insightful comment regarding the d-spacing. As mentioned by the reviewer, we have calculated the strain and d-spacing changes in the AuNPs as shown in Supplementary Figure 19 (Supplementary Table 1). After the photocatalytic reaction, the average lattice spacing of the AuNP increased by 0.00018 Å compared to their state in water, indicating that some adsorbates remained bound to the Au surface even after the reaction. Consequently, the Au-O bonds formed by the adsorption of ROS, which persisted on the surface post-reaction, may contribute to the observed increase in Au lattice spacing.

2. P 12: Fig. 4c shows the corresponding strain field of a single AuNP under different irradiation conditions. The term "corresponding" is no longer relevant, given the edits made just above in the text.

Answer. We thank the reviewer for the insightful comment. Following the Reviewer's comment, we

have updated the revised manuscript.

[Revised main manuscript, page 12]

“Fig. 4c shows the strain field of a single AuNP under different irradiation conditions.”

3. As I mentioned in my previous round of comments, including strain instead of displacement in the DFT results would provide clear evidence that the strain response follows the same trends as observed experimentally. In comment 16-b, I did not ask for new DFT simulations, but merely for strain calculations using the already relaxed slabs. Alternatively, the authors could argue for the use of displacement analysis by emphasizing that the “bulk” atoms in the DFT slabs were fixed to the gold lattice parameter during the DFT simulation. Thus, a qualitative interpretation could extend the displacement analysis to that of strain.

Answer. We acknowledge the insufficiency of our previous response and concur with the reviewer’s suggestion regarding the importance of analyzing the out-of-plane relaxation in DFT for a more comprehensive understanding of the surface strain. Our current DFT slab model employs a supercell in the nearly isolated limit to mitigate complex molecule interactions, focusing on surface-molecule interactions while utilizing of a six-layer configuration for computational feasibility. However, due to the limited number of layers in our slab model, the convergence of out-of-plane relaxation may not be adequate to precisely characterize surface strain. Consequently, we opted to analyze external pressure from DFT calculations to assess surface strain after OH adsorption. Additional Resource 1a shows the external pressure of the OH (0.11 ML)/Au (111) surface as a function of the lattice constant. OH adsorption induces residual pressure of 2.12 kB at equilibrium lattice, indicating tensile strain on the surface. This strain is alleviated through lattice constant expansion, with full relaxation achieved at approximately 0.3%. Additional Resource 1b depicts external pressure dependence on OH coverage, revealing an increase in surface strain with OH coverage, suggesting direct induction of surface strain by OH adsorption. Thus, we believe that our current analysis can elucidates the relation between OH adsorption and surface strain.

[Additional Resource 1]

Additional Resource 1. Theoretical in-plane external pressure of the OH/Au(111) surface. **a** External pressure as a function of lattice constant for the OH (0.11 ML)/Au(111) surface. **b**

External pressure as a function of OH coverage. The zero value corresponds to the external pressure of the pristine Au(111) surface. 1 monolayer (ML) refers to one OH molecule per topmost Au atom.

REVIEWERS' COMMENTS

Reviewer #5 (Remarks to the Author):

The comments addressed in the last version are satisfying. I recommend publication.